# The Mechanism of Prediction Head in Non-contrastive Self-supervised Learning

**Zixin Wen**
Machine Learning Department
Carnegie Mellon University
zixinw@andrew.cmu.edu

**Yuanzhi Li**
Machine Learning Department
Carnegie Mellon University
yuanzhil@andrew.cmu.edu

## Abstract

The surprising discovery of the BYOL method shows the negative samples can be replaced by adding a prediction head to the neural network. It is mysterious why even when there exist trivial collapsed global optimal solutions, neural networks trained by (stochastic) gradient descent can still learn competitive representations. In this work, we present our empirical and theoretical discoveries on non-contrastive self-supervised learning. Empirically, we find that when the prediction head is initialized as an identity matrix with only its off-diagonal entries being trainable, the network can learn competitive representations even though the trivial optima still exist in the training objective. Theoretically, we characterized the substitution effect and acceleration effect of the trainable, but identity-initialized prediction head. The substitution effect happens when learning the stronger features in some neurons can substitute for learning these features in other neurons through updating the prediction head. And the acceleration effect happens when the substituted features can accelerate the learning of other weaker features to prevent them from being ignored. These two effects enable the neural networks to learn diversified features rather than focus only on learning the strongest features, which is likely the cause of the dimensional collapse phenomenon. To the best of our knowledge, this is also the first end-to-end optimization guarantee for non-contrastive methods using nonlinear neural networks with a trainable prediction head and normalization.

## 1 Introduction

Self-supervised learning is about learning representations of real-world vision or language data without human supervision, and contrastive learning [62, 43, 41, 24, 20, 34] is one of the most successful self-supervised learning approaches. It has been known that the behavior of contrastive learning depends critically on the minimization of the *negative term*, which corresponds to contrasting the representations of *negative pairs*, i.e., pairs of different data points. However, the surprising finding of the *Bootstrap Your Own Latent* (BYOL) method by Grill et al. [37] initiated the research of *non-contrastive self-supervised learning*, which refers to contrastive learning methods without using the negative pairs. BYOL achieved state-of-the-art results in various computer vision benchmarks and there are plenty of follow-up works [39, 26, 21, 17, 33, 87, 44, 61] in this direction.

On a high level, in non-contrastive self-supervised learning, one wishes to learn a network $\phi$ such that $\phi(x)$ aligns in direction with $\phi(x')$, where $x$ and $x'$ are called the *positive pair*, generated by random augmentations from the same sample. Without the negative samples, collapsed global optima exist in the training objectives. The *complete collapse* is when $\phi(\cdot)$ is a constant vector whose variance is zero. Another trivial solutions called **dimensional collapse** by [44] is when all the coordinates $\phi_i(\cdot)$ are exactly aligned. Nevertheless, adding a trainable prediction head on top of (one branch of) $\phi(x)$ magically avoids learning such solutions, **even though the prediction head *can* possibly learn the identity mapping and render itself useless**. A more formal introduction will be given in Section 2.

Since the proposition of BYOL, there have been lots of empirical studies trying to understand non-contrastive learning. The SimSiam method by Chen and He [26] shows the exponential moving average (EMA) is not necessary for avoiding collapsed solutions while **stop-gradient** is necessary. [68] empirically disproved using batch normalization (BN) is the reason why BYOL can avoid collapse. [21, 88] further explored other similar approaches. If one wishes to work without both asymmetry and the negative pairs, one must add extra diversity-enforcing structures as in *Barlow Twins* [87] or [33, 44, 17]. Although these previous papers provided some empirical insights, in theory, the question of how the prediction head helps in learning those diverse features is still unanswered.

Despite the great empirical progress, there is very little theoretical progress towards explaining them. Most of existing theories focus on contrastive learning, especially from the statistical learning perspective [79, 81, 14, 80, 40, 82, 13, 15, 47, 45, 59]. However, due to the existence of trivial collapsed *global optimal* solutions (even with the prediction head) of the non-contrastive methods, to the best of our knowledge, *there is no well-established statistical framework for those methods yet*. To explain the non-contrastive learning, it is inevitable to study how the solutions are chosen during the optimization. Therefore, our research questions are:

---

**Our theoretical questions: the role of prediction head**

Why do most non-contrastive self-supervised methods learn collapsed solutions when the so-called prediction head is absent in the network architecture? How does the *trainable* prediction head help **optimizing** the neural network to learn more diversified representations in non-contrastive self-supervised learning?

---

Due to the existence of trivial collapsed optimal solutions of the non-contrastive learning objective, we need to understand the **implicit bias in optimization** posed by the prediction head. However, to the best of our knowledge, all of the previous implicit biases theories focus only on the supervised learning tasks, and thus cannot be applied to our question. On a high level, the results in this paper are summarized as follows:

**Our empirical contributions.** In non-contrastive self-supervised learning, we obtain the following experimental results:

- We discover empirically that even when the prediction head is **linear** and initialized as an identity matrix with only off-diagonal entries being trainable, the performance of learned representation is comparable to using the usual non-linear two-layer MLP or randomly initialized (trainable) linear prediction head. See Figure 1.

- We empirically verified that even when the prediction head is an identity-initialized matrix (with fixed diagonal entries), its off-diagonal entries display a rise-and-fall pattern, and it does not always converge to a symmetric matrix. See Figure 3.

**Our theoretical contributions.** We based our theory on a very simple setting, where the data consist of two features: the strong feature and the weak feature. Intuitively, the strong features in a dataset are the ones that show up more frequently or with large magnitude, and weak features as those that show up rarely or with small magnitude. We consider learning with a **two-layer non-linear neural network with output normalization** using (stochastic) gradient descent. Under this setting:

- We prove that without a prediction head, even with BN on the output to avoid complete collapse, the networks will still converge to dimensional collapsed solutions, which provides a theoretical explanation to the dimensional collapse phenomenon observed in [44].

- We prove that the trainable prediction head, combined with suitable output normalization and stop-gradient operation, can learn diversified features to avoid the dimensional collapse problem. We characterize two effects leveraged by the prediction head: the **substitution effect** and the **acceleration effect**, as intuitively described below:

---

**The effects of the trainable prediction head**

In our setting, we prove that the trainable prediction head can help to learn diversed features by leveraging two effects: the **substitution effect** and the **acceleration effect**. The substitution effect happens when by learning the prediction head, the learned stronger features in some neurons can substitute for learning the same features in other neurons. The acceleration effect happens when the substituted features from the prediction head further accelerate learning the weaker features in those substituted neurons.

---

Besides the above effects, we also explain in our setting, how the two common components in non-contrastive learning: *stop-gradient* operation and *output normalization*, can assist the prediction

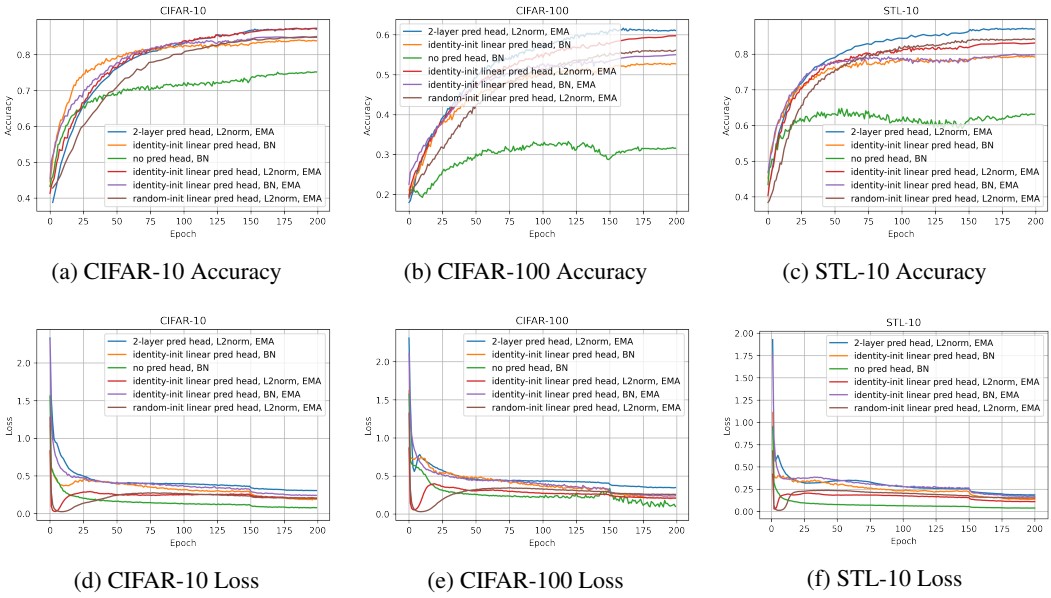

Figure 1: Performances of using different prediction heads. Here in CIFAR-10, CIFAR-100 and STL-10, identity-initialized linear prediction head can achieve good accuracies comparable to commonly used two-layer non-linear MLP or randomly-initialized linear head. All the prediction heads are trainable, while for identity-init prediction head only the off-diagonals are trainable. Here BN or L2norm represents the output normalization, and EMA represents using exponential moving average to update the target network as in BYOL [39].

head in creating those effects during training, which will be further discussed in Section 5.3. There are already some theoretical papers [78, 83, 63] that try to address similar questions. Our results provide a completely different perspective compared to them: **We explain why *training* the prediction head can encourage the network to learn diversified features and avoid dimensional collapses**, even when the trivial collapsed optima still exist in the training objective, which is not covered by the prior works, as shall be discussed below.

## 1.1 Comparison to Similar Studies

In this section, we will clarify the differences between our results and some similar studies. We point out that all the claims below are derived **only in our theoretical setting** and are partially verified in experiments over datasets such as CIFAR-10, CIFAR-100, and STL-10.

**Can eigenspace alignment explain the effects of training the prediction head?**  The paper [78] presented a theoretical statement that (symmetric) linear prediction head will *converge* to a matrix that commutes with the covariance matrix of linear representations *at the end of training*, and they provided experiments to support their theory. However, our theory suggests that *the intermediate stage of training the prediction head matters more to the feature learning of the encoder network than the convergence*. Indeed, as shown in Figure 3, in many cases, the trainable projection head will **converge back to identity** after training, which commutes with any covariance matrix. Moreover, the experiments in Figure 3a shows *the training trajectory of the prediction head displays a clear two-stage separation*, which demonstrates that the convergence result (e.g., the eigenspace alignment result in [78]) is not sufficient to understand the trainable prediction head.

**Can the symmetric prediction head explain the trainable prediction head?**  In the paper [78], experiments over the STL-10 dataset showed that the linear prediction head converges to a symmetric matrix during training. And the follow-up paper [83] established a theory under the symmetric prediction head (which is not trained but manually set at each iteration). Specifically, under their linear network setting, where $W$ is the weight matrix of the base encoder, they manually set the prediction head $W_p$ at iteration $t$ to be $W_p^{(t)} \leftarrow W^{(t)} \mathbb{E}_{x_1} x_1 x_1^\top (W^{(t)})^\top$ and the outputs of both online and target network are not normalized. Under this manual update rule of the prediction head, they proved a subspace learning result over spherical gaussian data. Nevertheless, our experiments in

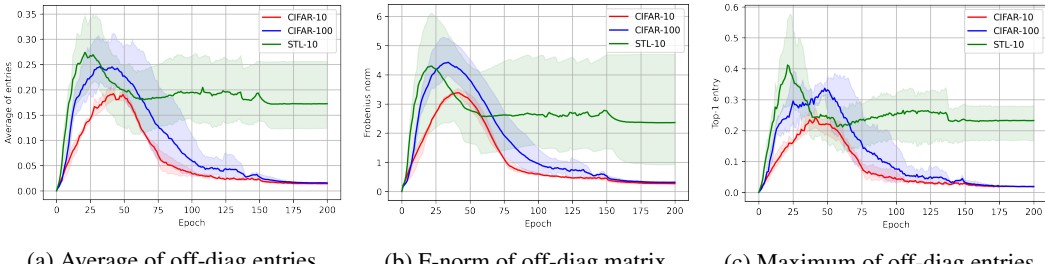

(a) Average of off-diag entries     (b) F-norm of off-diag matrix     (c) Maximum of off-diag entries

Figure 2: Trajectories of the identity-initialized prediction head with a $(\min, \max)$ confidence band, average over 3 runs. In all three datasets, we observe a consistent rise and fall trajectory pattern.

Figure 1 and Figure 3b show that even if we initialize the prediction head using a symmetric matrix (identity), **the trainable prediction head can be very asymmetric at the early training stage when the encoder network learn most of its features**. Actually, in the presence of feature imbalance (e.g., $\mathbb{E}_{x_1} x_1 x_1^\top$ has huge eigen-gap), the symmetric prediction head is also likely to learn a rank-one matrix where $W$ focus on learning the largest eigenvector of $\mathbb{E}_{x_1} x_1 x_1^\top$.

**The role of stop-gradient and output-normalization.** It is discussed in the theory of Tian et al. [78] that without the stop-gradient, the linear network will learn the zero (constant) solution. [83] also incorporated the stop-gradient into their theory, but did not explain why it is necessary for their setting. As a comparison, we proved in our setting that the stop-gradient and output-normalization together can turn the features substituted via the prediction head into a factor in the gradient of the slower learning neurons, thereby creating the acceleration effect. In contrast, analyses in [78, 83] did not incorporate the output normalization, even though their experiments have used certain forms of normalizations. To the best of our knowledge, our paper is the first to explain the effects of output-normalization in optimizing nonlinear neural networks in self-supervised learning.

## 2 Preliminaries on Non-contrastive Learning

In this section, we formally define what is non-contrastive self-supervised learning. To do this, we first introduce contrastive learning following [24, 85] as background. We use $[N]$ as a shorthand for the index set $\{1, \ldots, N\}$.

**Background on contrastive learning.** Letting $\phi_W(\cdot)$ be the neural networks, contrastive learning aims to learn good representations $\phi_W$ via contrasting representations of similar data samples to those of dissimilar ones. Usually we are given a batch of data points $\{X_i\}_{i \in [N]}$, and we construct for each $i \in [N]$ a positive pair $(X_i^{(1)}, X_i^{(2)})$ by applying random data augmentations to $X_i$, and collect negative pairs $(X_i^{(1)}, X_j^{(2)})$ for $i \neq j \in [N]$. Now given $z_i = \phi_W(X_i^{(1)})$, $z_i' = \phi_W(X_i^{(2)})$, $i \in [N]$, we train the network $\phi_W$ to minimize the contrastive loss:

$$L_{\text{contrastive}}(\phi_W) := \frac{1}{N} \sum_{i \in [N]} \underbrace{-\mathbf{sim}(z_i, z_i')/\tau}_{\text{positive term}} + \underbrace{\log \left[ \sum_{j \in [N]} \exp\left(\mathbf{sim}(z_i, z_j')/\tau\right) \right]}_{\text{negative term}} \qquad (2.1)$$

where $\mathbf{sim}(\cdot, \cdot)$ is the similarity metric, often defined as the cosine similarity, and $\tau$ is the so-called temperature hyper-parameter. Intuitively, minimizing the contrastive loss can be roughly viewed as trying to classify the representation $z_i$ as $z_i'$ instead of $z_j', j \neq i$. It is a common belief that in order for the network $\phi_W$ to be able to "distinguish" data points $X_i$ from $X_j, j \neq i$, merely minimizing the positive term of contrastive loss is not sufficient.

**Non-contrastive self-supervised learning.** We choose the `SimSiam` method [26] as our primary framework, whose differerence with `BYOL` is a EMA component that is proven inessential in [26]. Following the same notations as above, except that $z_i' = \mathsf{StopGrad}[\phi_W(X_i^{(2)})]$ is detached from

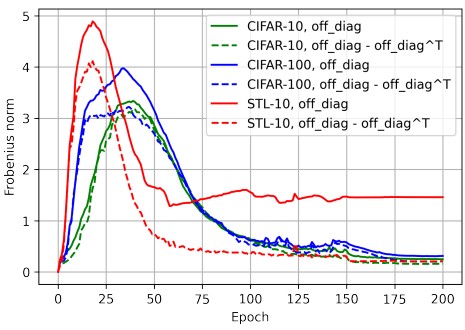 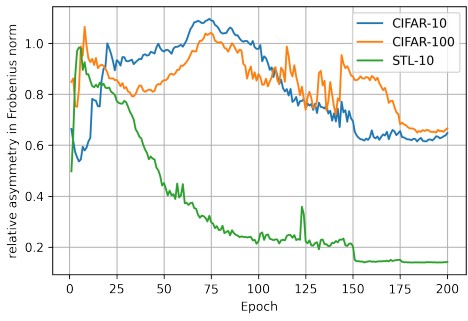

(a) $\|\text{off-diag}(E^{(t)})\|_F$ and $\|E^{(t)} - (E^{(t)})^\top\|_F$

(b) $\|E^{(t)} - (E^{(t)})^\top\|_F / \|\text{off-diag}(E^{(t)})\|_F$

Figure 3: Trajectories of the identity-initialized prediction head. off-diag$(E)$ is obtained by setting the diagonal of $E$ to be zero. In (a), we discover that the Frobenius norm of our identity-initialized prediction head's off-diagonal matrix clearly display a two stage separation, more precisely, a rise and fall pattern; In (b), The off-diagonal matrix of the prediction head is not symmetric in CIFAR-10 and CIFAR-100.

gradient computation, the loss objective become: (the symmetric network version)

$$L'_{\texttt{SimSiam}}(\phi_W) = \frac{1}{N} \sum_{i \in [N]} -\mathbf{sim}(z_i, z'_i) \tag{2.2}$$

which is just the positive term in contrastive loss (2.1) (not divided by $\tau$). Clearly there exist plenty trivial **global optimal** solutions for this objective. For example, the *complete collapse* refers to when $\phi_W(\cdot)$ learns some constant vector. Another solution called **dimensional collapse** [44] is when all the coordinates $[\phi_W(\cdot)]_i$ has correlation $\pm 1$. The dimensional collapsed solution can minimize the objective (2.2) even when $\phi_W(\cdot)$ is BN-normalized to avoid learning a constant vector [44, 88].

However, by adding a *trainable prediction head* on top of $z_i$, the training miraculously succeeds and outputs a state-of-the-art feature extractor. Let $g(\cdot)$ be a shallow feed-forward network (often one or two-layer, or even simply linear), we train $g$ and $\phi_W$ simultaneously on the following objective:

$$L_{\texttt{SimSiam}}(\phi_W, g) = \frac{1}{N} \sum_{i \in [N]} -\mathbf{sim}(g(z_i), z'_i) \tag{2.3}$$

where $z'_i$ is still detached from gradient computation. The $g(z_i) = g \circ \phi_W(X_i^{(1)})$ and the detached part $z'_i = \mathsf{StopGrad}[\phi_W(X_i^{(2)})]$ are often called the *online* network and the *target* network respectively following [39], known as two branches of non-contrastive learning. Note that the trainable prediction head can represent identity function, so the objective (2.3) still has the collapsed optima.

## 3 Problem Setup

In this section, we present the setting of our theoretical results. We first define the data distribution.

**Notations.** We use $O, \Omega, \Theta$ notations to hide universal constants with respect to $d$ and $\widetilde{O}, \widetilde{\Omega}, \widetilde{\Theta}$ notations to hide polynomial factors of $\log d$. We denote $a = o(1)$ if $a \to 0$ when $d \to \infty$. We use the notations $\mathsf{poly}(d)$, $\mathsf{polylog}(d)$ to represent large constant degree polynomials of $d$ or $\log d$. We use $\mathcal{N}(\mu, \Sigma)$ to denote standard normal distribution in with mean $\mu$ and covariance matrix $\Sigma$. We use the bracket $\langle \cdot, \cdot \rangle$ to denote the inner product and $\| \cdot \|_2$ the $\ell_2$-norm in Euclidean space. And for a subspace $V \subset \mathbb{R}^d$, we denote $V^\perp$ as its orthogonal complement. We use $\mathbb{1}_B$ to denote the indicator function of event $B$. We use $I_m$ to denote the $m \times m$ identity matrix.

Following the standard structure of image datasets, we consider data divided into patches, where each patch can contain either features or noises.

**Definition 3.1** (data distribution and features). Let $X \sim \mathcal{D}$ be $X = (X_1, \dots, X_P) \in \mathbb{R}^{d \times P}$ where each $X_i \in \mathbb{R}^d$ is a patch. We assume that there are two feature vectors $v_1, v_2$ such that $\|v_\ell\|_2 = 1, \ell = 1, 2$ and are orthogonal to each other. To generate a sample $X$, we uniformly sampled $\ell \in [2]$ and generate for each $p \in [P]$:

$$X_p = z_p(X)v_\ell + \xi_p \mathbb{1}_{z_p=0}, \quad \mathbb{E}_{X \sim \mathcal{D}}[z_p(X)] = 0, \quad \forall p \in [P]$$

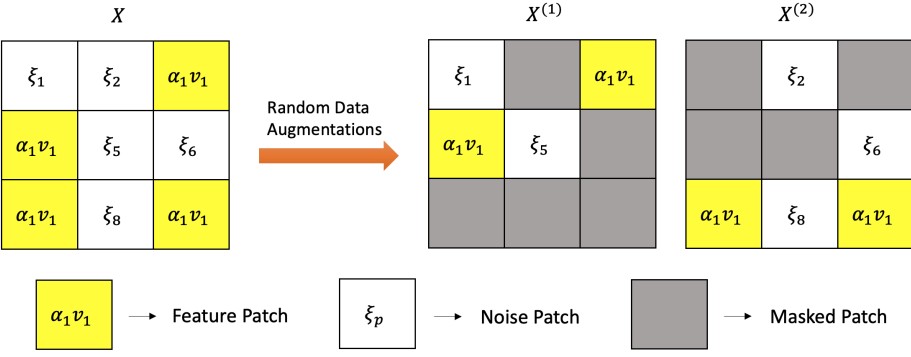

Figure 4: Illustration of the data distribution and data augmentations. Each data is equipped with a feature, either $v_1$ or $v_2$, and contains a lot of noise patches. After the data augmentations, the positive pair $(X^{(1)}, X^{(2)})$ is constructed by randomly masking out half of non-overlapping patches for each positive sample. The reason for constructing positive pair with non-overlapping patches is because of the strong noise assumption we made in Assumption 3.2 and the *feature decoupling* principle in [85].

where $z_p(X)$ is the latent vector of $X$, $\xi_p$ is the noise vector of patch $p \in [P]$ whose assumption will be given in Assumption 3.2. We denote $\mathcal{S}(X) = \{p : z_p(X) \neq 0\} \subseteq [P]$ as the set of feature patches, where $z_p(X) = z_{p'}(X) \in \{0, \pm\alpha_\ell\}, \forall p, p' \in [P]$, where $\alpha_\ell$ will be picked afterwards. We assume $P = \mathsf{polylog}(d)$, $\mathcal{S}(X) \equiv P_0 = \Theta(\log d)$ for every $X$. A figurative illustration is given in Figure 4.

**Strong and weak features.** We pick $\alpha_1 = 2^{\mathsf{polyloglog}(d)}$ and $\alpha_2 = \alpha_1/\mathsf{polylog}(d)$. Hence $v_1$ is the *strong feature* and $v_2$ is the *weak feature*, and we want the learner network to learn both $v_1, v_2$ (but by different neurons) as their learning goal. This is a simplification of the real scenario. Intuitively, we can think of the strong features in a dataset are the ones that show up more frequently or with larger magnitude, and weak features as those that show up rarely or with smaller magnitude.

**Assumption 3.2** (noise). Denoting $V = \mathrm{span}(v_1, v_2)$, we assume $\xi_p \in V^{\perp}$ is independent for each $p \in [P] \setminus S(X)$, where $X = (X_p)_{p \in [P]} \sim \mathcal{D}$, and:

(a) For any unit vector $u \in V^{\perp}$, $\mathbb{E}[\langle \xi_p, u \rangle] = 0$, and $\mathbb{E}[\langle \xi_p, u \rangle^6] = \sigma^6$ for some $\sigma = \Theta(1)$;

(b) It holds for some $\varrho \in [0, \frac{1}{d^{\Omega(1)}}]$ it holds $|\mathbb{E}[\langle u_1, \xi_p \rangle^3 \langle u_2, \xi_p \rangle^3]| \leq \varrho$ and $|\mathbb{E}[\langle u_1, \xi_p \rangle^5 \langle u_2, \xi_p \rangle]| \leq \varrho$ for any two vectors $u_1, u_2 \in V^{\perp}$ that are orthogonal to each other.

*Remark* 3.3. A simple example of our noise $\xi_p$ is the spherical Gaussian noise in $V^{\perp}$. Assumption 3.2b ensures that the prediction head cannot be used to cancel the noise correlation between different neurons. We point out that the features in our data can be learned via clustering, but we emphasize that we do not intend to compare our algorithm with any clustering method in this setting since our goal is to study how the prediction head helps in learning the features.

### 3.1 Learner Network

Following the `SimSiam` framework, the online and target network share the same encoder network in our setting, as explained in Section 2. We consider the base encoder network $f$ as a simple convolutional neural network: Let $W = (w_1, \ldots, w_m) \in \mathbb{R}^{d \times m}$ be the weight matrix, where $w_i \in \mathbb{R}^d$, the **encoder network** $f$ is defined by

$$f_j(X) := \sum_{p \in [P]} \sigma(\langle w_j, X_p \rangle), \qquad \forall j \in [m]$$

Here we use the cubic activation function $\sigma(z) = z^3$, as polynomial activations are standard in literatures of deep learning theory [9, 35, 50, 2, 52, 23] and also has comparable performance in practice [2]. The (identity initialized) prediction head is defined as a matrix $E = [E_{i,j}]_{(i,j) \in [m]^2}$ with $E_{i,i} \equiv 1, i \in [m]$, where only the the off-diagonals $E_{i,j}, i \neq j$ are trainable parameters. The **online**

**network** $\widetilde{F}$ is defined by: given $j \in [m]$, we let $F_j(X) := f_j(X) + \sum_{r \neq j} E_{j,r} f_r(X)$, and

$$\widetilde{F}_j(X) := \mathsf{BN}\left(F_j(X)\right) = \mathsf{BN}\left[\sum_{p \in [P]}\left(\sigma(\langle w_j, X_p\rangle) + \sum_{r \neq j} E_{j,r}\sigma(\langle w_r, X_p\rangle)\right)\right]$$

where the batch normalization $\mathsf{BN}$[1] here is defined as follows: Given a batch of inputs $\{z_i\}_{i \in [N]}$,

$$\mathsf{BN}(z_i) := \frac{z_i - \frac{1}{N}\sum_{i \in [N]} z_i}{\sqrt{\frac{1}{N}\sum_{i \in [N]} z_i^2 - \left(\frac{1}{N}\sum_{i \in [N]} z_i\right)^2}} \tag{3.1}$$

And the we define the **target network** $G$ as $\widetilde{G}_j(X) := \mathsf{BN}\left(G_j(X)\right) = \mathsf{BN}\left(f_j(X)\right), \forall j \in [m]$.

## 3.2 Training Algorithm

**Data augmentation.** We use a very simple data augmentation: for each data $X = (X_p)_{p \in [P]}$, we randomly and uniformly sample half of the patches $\mathcal{P} \subseteq [P]$ to generate the *positive pair*:

$$X^{(1)} = (X_p \mathbb{1}_{p \in \mathcal{P}})_{p \in [P]}, \quad X^{(2)} = (X_p \mathbb{1}_{p \notin \mathcal{P}})_{p \in [P]} \tag{3.2}$$

Our data augmentation is similar to the common random cropping used in contrastive learning [22, 76]. It is also analogous to the data augmentations studied in theoretical literatures [85, 47, 58].

**Non-contrastive loss function.** Now we define the loss function as follows: we sample $N$ data points $\{X_i\}_{i \in [N]}, X_i \overset{\text{i.i.d.}}{\sim} \mathcal{D}$ and apply our data augmentation (3.2) to obtain $\mathcal{S} = \{X^{(i,1)}, X^{(i,2)}\}_{i \in [N]}$. Now we define

$$L_{\mathcal{S}}(W, E) := \frac{1}{N}\sum_{i \in [N]}\left\|\widetilde{F}(X^{(i,1)}) - \mathsf{StopGrad}[\widetilde{G}(X^{(i,2)})]\right\|_2^2 \tag{3.3}$$

where the $\mathsf{StopGrad}$ operator detach gradient computation of the target network $\widetilde{G}(\cdot)$. This form of objective (3.3) is first defined in [37] and is equivalent to (2.3) in Chen and He [26] when $\widetilde{F}$ and $\widetilde{G}$ share the same encoder network $f(\cdot)$ and their outputs are normalized.

**Intuitions of the data augmentation and collapse.** In Definition 3.1, the features $v_1, v_2$ appear in multiple patches, but the noises are independent across different patches (see Figure 4). As our data augmentation produces positive pairs with non-overlapping patches, learning to emphasize noises cannot align the representations of the positive pair, but learning **either one of** the features $\phi(X) = \sum_p \sigma(\langle v_1, X_p\rangle)$ or $\phi(X) = \sum_p \sigma(\langle v_2, X_p\rangle)$ is sufficient. **We consider learning the same feature $v_i$ in *all the neurons* $f_j$ in the encoder network $f$ as the dimensional collapsed solution.**

**Initialization and hyper-parameters.** At $t = 0$, we initialize $W$ and $E$ as $W_{i,j}^{(0)} \sim \mathcal{N}(0, \frac{1}{d})$ and $E^{(0)} = I_m$ and we only train the off-diagonal entries of $E^{(t)}$. For the simplicity of analysis, **we let $m = 2$, which suffices to illustrate our main message.** For the learning rates, we let $\eta \in (0, \frac{1}{\mathsf{poly}(d)}]$ be sufficiently small and $\eta_E \in [\eta/\alpha_1^{O(1)}, \eta/\mathsf{polylog}(d)]$, which is smaller than $\eta^2$.

**Optimization algorithm** Given the data augmentation and the loss function, we perform (stochastic) gradient descent on the training objective (3.3) as follows: at each iteration $t = 0, \ldots, T-1$, we sample a new batch of augmented data $\mathcal{S}_t = \{X^{(t,i,1)}, X^{(t,i,2)}\}_{i \in [N]}$ and update

$$W^{(t+1)} = W^{(t)} - \eta\nabla_W L_{\mathcal{S}_t}(W^{(t)}, E^{(t)}), \quad E_{i,j}^{(t+1)} = E_{i,j}^{(t)} - \eta_E\nabla_{E_{i,j}} L_{\mathcal{S}_t}(W^{(t)}, E^{(t)}), \quad \forall i \neq j.$$

If we do not train the prediction head, we just simply keep $E^{(t)} \equiv I_m$.

---

[1] We use batch normalization as a output-normalization method, rather than for the supposed implicit negative term effects as disproved in Richemond et al. [68].

[2] We conjecture that by modifying certain assumptions for the noise (especially by allowing the noise to span the feature subspace $V$), one can prove a similar result for the case $\eta_E = \eta$.

# 4 Statements of Main Results

In this section, we shall present our main theoretical results on the mechanism of learning the prediction head in non-contrastive learning. To measure the correlation between neurons, we introduce the following notion: letting $\mathbf{Var}(\psi(X)) := \mathbb{E}_{X \sim \mathcal{D}}[(\psi(X) - \mathbb{E}[\psi(X)])^2]$ be the variance of any function $\psi$ of $X \sim \mathcal{D}$, we denote the correlation $\mathbf{Corr}(\psi(X), \psi'(X))$ of any two function $\psi, \psi'$ over $\mathcal{D}$ as

$$\mathbf{Corr}(\psi(X), \psi'(X)) := \frac{\mathbb{E}[(\psi(X) - \mathbb{E}[\psi(X)])(\psi'(X) - \mathbb{E}[\psi'(X)])]}{\sqrt{\mathbf{Var}(\psi(X))}\sqrt{\mathbf{Var}(\psi'(X))}}$$

Now we present the main theorem of training with a prediction head, and set $m = 2$.

**Theorem 4.1** (learning with prediction head and BN, see Theorem F.2). *For every $d > 2$, let $N \geq \mathsf{poly}(d)$, $\eta \in (0, \frac{1}{\mathsf{poly}(d)}]$ be sufficiently small, and $\eta_E \in [\frac{\eta}{\alpha_1^{O(1)}}, \frac{\eta}{\mathsf{polylog}(d)}]$. Then with probability $1 - o(1)$, after training for $T = \mathsf{poly}(d)/\eta$ many iterations, we shall have for some $\ell \in [2]$:*

$$w_1^{(T)} = \beta_1 v_\ell + \varepsilon_1, \quad w_2^{(T)} = \beta_2 v_{3-\ell} + \varepsilon_2 \qquad with \quad |\beta_1|, |\beta_2| = \Theta(1), \|\varepsilon_1\|_2, \|\varepsilon_2\|_2 \leq \widetilde{O}(\frac{1}{\sqrt{d}})$$

*Furthermore, the objective converges:* $\mathbb{E}_{\mathcal{S} \sim \mathcal{D}^N}[L_{\mathcal{S}}(W^{(T)}, E^{(T)})] \leq \mathsf{OPT} + \frac{1}{\mathsf{poly}(d)} \leq O(\frac{1}{\log d})$. *Where $\mathsf{OPT}$ stands for the global optimum[3].*

Theorem 4.1 clearly shows the network learn all the desired features, even under huge imbalance between $v_1$ and $v_2$. This leads to the following corollary.

**Corollary 4.2.** *Under the same hyper-parameter in Theorem 4.1, with probability $1 - o(1)$, after training for $T = \mathsf{poly}(d)/\eta$ many iterations, then the encoder $f$ **avoids dimensional collapse:***

$$|\mathbf{Corr}(f_1(X), f_2(X))| \leq O(\frac{1}{\sqrt{d}}).$$

In contrast, learning without the prediction head will create strong correlations between any two neurons. To emphasize that this problem cannot be alleviated by having more neurons, we let the number of neurons $m$ be any positive integer in the following theorem.

**Theorem 4.3** (learning without prediction head but with BN, see Theorem G.1). *Let $N \geq \mathsf{poly}(d)$, $\eta = o(1)$ and the number of neurons $m \leq o(\alpha_1/\alpha_2)$. Suppose we freeze $E^{(t)} \equiv I_m$, then with probability $1 - o(1)$, after training for $T = \mathsf{poly}(d)/\eta$ many iterations, we shall have:*

$$w_j^{(T)} = \beta_j v_1 + \varepsilon_j \qquad with \quad |\beta_j| = \Theta(1), \|\varepsilon_j\|_2 \leq \widetilde{O}(\frac{1}{\sqrt{d}}) \qquad \qquad \text{for all } j \in [m]$$

*Furthermore, the objective converges:* $\mathbb{E}_{\mathcal{S} \sim \mathcal{D}^N}[L_{\mathcal{S}}(W^{(T)}, E^{(T)})] \leq \mathsf{OPT} + \frac{1}{\mathsf{poly}(d)} \leq O(\frac{1}{\log d})$. *This means the collapsed solution also reaches the global minimum of the objective. Again $\mathsf{OPT}$ stands for the global optimum.*

Note that since we have used BN as our output normalization, the learner is immune to complete collapse and must have a certain variance in the outputs. Immediately, we have a corollary.

**Corollary 4.4.** *Under the same hyper-parameter in Theorem 4.3, with probability $1 - o(1)$, after training with $E^{(t)} \equiv I_m$ for $T = \mathsf{poly}(d)/\eta$ many iterations, we shall have **dimensional collapse:***

$$|\mathbf{Corr}(f_i(X), f_j(X))| \geq 1 - O(\frac{1}{\sqrt{d}}), \qquad \text{for all } i, j \in [m].$$

In the following section, we shall give some intuitions by digging through the training process and separately discuss the four phases of the training process.

---

[3]Under our setting described in Section 2, the *global minimum* of our objective (3.3) in population is

$$\mathsf{OPT} = 2 - 2\frac{\mathbb{E}[|\mathcal{S}(X) \cap \mathcal{P}| \cdot |\mathcal{S}(X) \setminus \mathcal{P}|]}{\mathbb{E}[|\mathcal{S}(X) \cap \mathcal{P}|^2]} = \Theta(\frac{1}{\log d})$$

# 5 The Four Phases of the Learning Process

We divide the complete training process into four phases: phase I for learning the stronger feature, phase II for the substitution effect, phase III for the acceleration effect, and the end phase for convergence. The first three phases explain how the prediction head can help learn the base encoder, and the last phase explains why the off-diagonal entries often shrink later in training.

## 5.1 Phase I: Learning the Stronger Feature

At the beginning of training, the stronger feature $v_1$ enjoys a much larger gradient as opposed to the weaker feature $v_2$, so naturally, $v_1$ will be learned first. Without loss of generality, let us assume at initialization, the neuron $f_1$ has larger $v_1$ between $f_j, j \in [2]$, then we can show:

**Lemma 5.1** (learning the stronger feature, see Lemma C.13). *After some $t \geq T_1 = d^{2+o(1)}/\eta$, the feature $v_1$ in neuron $f_1$ will be learn to $\langle w_1^{(t)}, v_1 \rangle = \Omega(1)$, while all other features $\langle w_j^{(t)}, v_\ell \rangle = o(1)$ for $(j, \ell) \neq (1, 1)$. And the prediction head $\|E^{(t)} - I_2\|_2 \leq d^{-\Omega(1)}$ is still close to the initialization.*

In this phase, the prediction head has not come into play. The substitution effect can only happen after the feature $v_1$ in neuron $f_1$ is learned to a certain degree, and neuron $f_2$ remains largely unlearned.

## 5.2 Phase II: The Substitution Effect

To illustrate the substitution effect, let us keep assuming that neuron $f_1$ has already learned some significant amount of the strong feature $v_1$, say $w_1 = \beta_1 v_1 + residual$ with $|\beta_1| = \Omega(\|residual\|)$, then we have: (recall $f_j(\cdot), j \in [2]$ are the neurons of the encoder network)

**Lemma 5.2** (substitution effect, formal statement see Lemma D.8). *After $|\langle w_1^{(t)}, v_1 \rangle| = \Omega(1)$, we shall have $|E_{2,1}^{(t)}|$ increasing until $|E_{2,1}^{(t)} f_1(X^{(1)})| \gg |f_2(X^{(1)})|$ when $X$ is equipped with the strong feature $v_1$, for $T_2 - T_1 = o(T_1)$ iterations.*

**Intuition of the substitution effect.** After the stronger feature is learned in neuron $f_1$, the optimal way to align two positive representations $F_2(X^{(1)})$ and $G_2(X^{(2)})$ is not learning features in weight $w_2$, but use the prediction head to "substitute" the features in $f_1$ into $F_2$. This is how the substitution effect happens when trained with a prediction head.

## 5.3 Phase III: The Acceleration Effect

After the substitution of $v_1$ in $F_2$, our concern is, $w_2^{(t)}$ will learn $v_2$ and only $v_2$ eventually, according to the acceleration effect in the following lemma.

**Lemma 5.3** (acceleration effect, formal statement see Lemma E.8). *After $E_{2,1}^{(t)}$ is learned in Lemma 5.2, learning $v_2$ in $w_2^{(t)}$ will be much faster than $v_1$, until $\|w_2^{(t)} - \beta_2 v_2\| = o(1)$ for some $\beta_2 = \Theta(1)$.*

The acceleration effect is caused by the interactions between the prediction head, the stop gradient operation, and the normalization method (which in this case is the batch normalization).

**What is the role of the stop-gradient?** Thanks to the stop-gradient operation, when we compute the gradient $-\nabla_{w_2} F_2(X^{(1)}) \cdot \mathsf{StopGrad}[G_2(X^{(2)})]$ to learn $f_2$, this negative gradient will only try to maximize $f_2(X^{(1)}) \cdot f_2(X^{(2)})$, rather than to maximize $F_2(X^{(1)}) \cdot f_2(X^{(2)})$. This is because the stop-gradient is on $G$ not on $F$: while $F_2$ has a large component of $v_1$ borrowed from $f_1$ using $E$, $G_2$ does not have this component. So the gradient of $F_2$ is to align with the features in $G_2$ that does not contain many $v_1$, while the gradient of $G_2$ is to aligned with $F_2$ that contains a lot of $v_1$.

**What is the role of the output normalization?** Again due to the $\mathsf{StopGrad}$ operation, the gradient of $\widetilde{F_2}$ is taken with respect to the ratio $f_2(X^{(1)})/\sqrt{\mathbf{Var}[F_2(X^{(1)})]}$. As gradient descent tries to maximize this ratio, a direct computation gives

$$\nabla_{w_2} \frac{f_2(X^{(1)})}{\sqrt{\mathbf{Var}(F_2(X^{(1)}))}} \propto \sum_{\ell \in [2]} \left( [E_{2,1}^{(t)} \langle w_1^{(t)}, v_{3-\ell} \rangle^3]^2 + \mathbf{Var}[f_2(X^{(1)})] \right) \langle \nabla_{w_2} f_2(X^{(1)}), v_\ell \rangle v_\ell$$

which borrow the *substituted feature* $v_1$ from $f_1(\cdot)$ to adjust the gradient of $v_2$ in $f_2(\cdot)$, via the prediction head $E_{2,1}^{(t)}$. Without the output normalization, the learning of $v_1$ will dominate that of $v_2$ even when we train the prediction head.

### 5.4 The End Phase: Convergence

As the weak features are learned, we have already obtained a good encoder network $f(\cdot)$ as shown in Theorem 4.1. The rest of Theorem F.2 also contains the following result:

**Proposition 5.4** (convergence of the prediction head, see Theorem F.2c). *After some* $t \geq T = \mathrm{poly}(d)/\eta$ *iterations, we shall have* $\|E^{(t)} - I_2\|_F \leq \frac{1}{\mathrm{poly}(d)}$.

While we admit that only some of our real-world experiments in Figure 3 show the convergence to zero for the off-diagonal entries of the prediction head, most of the experiments do display a rise and fall trajectory pattern of off-diagonal entries consistently, which supports our theory to some degree.

## 6 Additional Related Work

**Self-supervised learning**  Self-supervised learning has created huge success in natural language processing [30, 86, 18] and established the pretrain-finetune paradigm for deep learning. In vision, contrastive learning [75, 41, 24, 20, 27, 28, 34, 64, 33] became dominant in many downstream tasks recently. Another approach is the generative learning [65, 16, 42], which also gives promising results. Applications such as [64, 66] also illustrate the power of contrastive learning in multiple domains.

**Theory of self-supervised learning**  The theoretical side of self-supervised learning developed quickly due to the success of contrastive learning. Since [12], plenty of papers have studied the contrastive learning. [25, 69] discussed many interesting phenomena associated with the negative term. Saunshi et al. [71] provided evidence that function class agnostic analyses is vacuous. [85] took a feature learning view, and inspired our analysis in the non-contrastive setting. For generative learning, [51, 74] provides downstream performance guarantees. [70, 84] studied the natural language tasks. [58] gave a recovery guarantee for tensors under hidden Markov models. [4] provided an optimization guarantee for GANs trained by stochastic gradient descent ascent.

**Feature learning theory of deep learning**  Our theoretical results are also inspired by the recent progress of the feature learning theory of neural networks [55, 56, 5, 3, 49, 90, 46]. [55] initiate the study of the speed difference in learning different types of features. [1] developed theory for learning two-layer neural networks beyond the *neural tangent kernel* (NTK) [7, 8, 6, 32, 11]. [5, 3, 2] further studied how features are learned in different deep learning tasks. Before this recent progress, [77, 89, 19, 72, 31, 53, 54] also studied how shallow neural networks can learn on certain simple data distributions, but all of them focus on the supervised learning. There are also plenty of studies [73, 38, 10, 60, 48, 67, 29] on the implicit bias of optimization in deep learning, but none of their techniques are designed for analyzing self-supervised learning.

## 7 Conclusion

In this paper, we showed how the prediction head can ensure the neural network learns all the features in non-contrastive learning through theoretical investigation. Our key contribution is that we proved the prediction head can leverage two effects called substitution effect and acceleration effect during the training process. We also gave an explanation for the dimensional collapse phenomenon. We believe our theory, although based on a very simple setup, can provide some insights into the inner workings of non-contrastive self-supervised learning.

## Acknowledgments and Disclosure of Funding

Funding in direct support of this work includes NSF Award 2145703.

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
