Figure 6: The feature learning process over synthetic data. When trained with the prediction head, after the strong feature is learned in the faster learning neuron, the weak feature can be learned in the slower learning neuron. When trained without the prediction head, both neurons will learn the strong feature and ignore the weak feature.

## A  Experiment Details

The framework we use in our experiments is shown in Figure 5. We use a modified version of the codebase shared by the authors of [33], and we use the same data augmentation in their implementation. All our experiments (except for Figure 7) use the following architecture and hyper-parameters: we choose standard ResNet-18 as base encoder architecture, $0.003$ as the learning rate for Adam optimizer, a two-layer MLP with ReLU activation and $512$ hidden neurons as the projection head, an identity-initialized but diagonally froze linear matrix (with shape (64x64)) as the prediction head and a non-tracking-stats, non-affine, non-momentum BN layer as the output normalization. Our experiments in Figure 3 use the same architecture and hyper-parameters, but some runs are trained with EMA with momentum $0.99$, with output BN replaced by $\ell_2$-norm or using different prediction heads (such as a two-layer MLP or a linear head, with Pytorch default initialization). Evaluation in Figure 1 is by training a linear classifier on top of frozen encoder with no data augmentation.

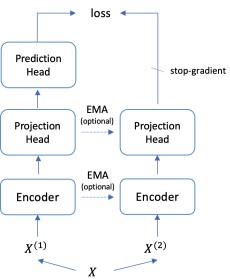

Figure 5: Framework.

## B  Notations and Gradients

In this section, we will give some useful notations and warm-up computations for the technical proofs in subsequent sections. We summarize here the notations that will also be defined in later sections:

**Notations.** We denote $\mathcal{E}_j = \mathbb{E}[\langle w_j, \xi_p \rangle^6]$, $\mathcal{E}_{j,3-j} = \mathbb{E}\left[ (\langle w_j, \xi_p \rangle^3 + E_{j,3-j} \langle w_{3-j}, \xi_p \rangle^3)^2 \right]$, and

$$C_0 = \frac{\mathbb{E}[|\mathcal{S}(X) \cap \mathcal{P}| \cdot |\mathcal{S}(X) \setminus \mathcal{P}|]}{2}, \quad C_1 = \frac{\mathbb{E}\left[|\mathcal{S}(X) \cap \mathcal{P}|^2\right]}{2}, \quad C_2 = P - |\mathcal{S}(X)|,$$

$$\bar{B}_{j,\ell}^3 = \mathsf{StopGrad}[\langle w_j, v_\ell \rangle^3], \qquad B_{j,\ell} = \langle w_j, v_\ell \rangle, \qquad Q_j = (\mathbb{E}[\mathsf{StopGrad}[G_j^2(X^{(2)})]])^{-1/2}.$$

and

$$U_j := \mathbb{E}[F_j^2(X^{(1)})] = \sum_{\ell \in [2]} C_1 \alpha_\ell^6 (B_{j,\ell}^3 + E_{j,3-j} B_{3-j,\ell}^3)^2 + C_2 \mathcal{E}_{j,3-j}$$

$$H_{j,\ell} := C_1 \alpha_\ell^6 (B_{j,\ell}^3 + E_{j,3-j} B_{3-j,\ell}^3)^2 + C_2 \mathcal{E}_{j,3-j},$$

$$K_{j,\ell} := C_1 \alpha_\ell^6 (B_{j,\ell}^3 + E_{j,3-j} B_{3-j,\ell}^3)(B_{j,3-\ell}^3 + E_{j,3-j} B_{3-j,3-\ell}^3)$$

Moreover, we denote $\Phi_j := Q_j / U_j^{3/2}$, and (recall $V := \mathrm{span}(v_1, v_2)$)

$$R_j := \langle \Pi_{V^\perp} w_j, w_j \rangle \qquad R_{1,2} := \langle \Pi_{V^\perp} w_1, w_2 \rangle \qquad \overline{R}_{1,2} := \frac{\langle \Pi_{V^\perp} w_1, w_2 \rangle}{\|\Pi_{V^\perp} w_1\|_2 \|\Pi_{V^\perp} w_2\|_2}$$

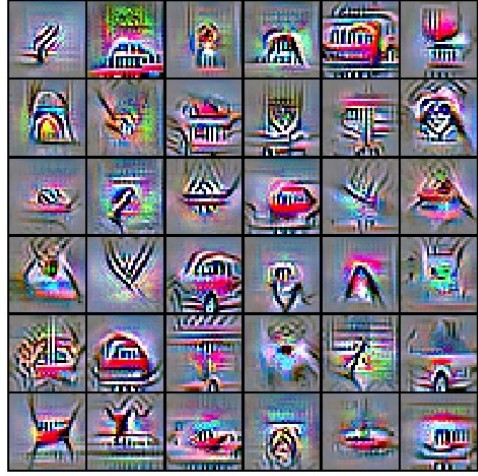
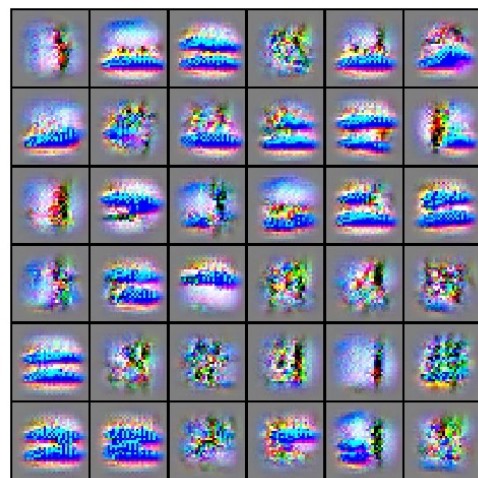

(a) Features learned with prediction head           (b) Features learned without prediction head

Figure 7: Feature visualization of deep neural network. We visualized the features of an Wide-ResNet-16x5 following the BYORL method by Gowal et al. [36], a adversarial robust version of BYOL. Features learned with prediction head obviously have more variety than features learned without the prediction head. Our feature visualization technique follows from [5].

For any $j \in [2]$, the gradient $-\nabla_{w_j} L(W, E)$ can be decomposed as

$$-\nabla_{w_j} L(W, E) = \sum_{\ell \in [2]} (\Lambda_{j,\ell} + \Gamma_{j,\ell} - \Upsilon_{j,\ell}) v_\ell - \sum_{(j',\ell) \in [2] \times [2]} \Sigma_{j',\ell} \nabla_{w_j} \mathcal{E}_{j',3-j'}$$

$$\Lambda_{j,\ell} := C_0 \Phi_j \alpha_\ell^6 B_{j,\ell}^5 H_{j,3-\ell}$$

$$\Gamma_{j,\ell} := C_0 \Phi_{3-j} E_{3-j,j} \alpha_\ell^6 B_{3-j,\ell}^3 B_{j,\ell}^2 H_{3-j,3-\ell}$$

$$\Upsilon_{j,\ell} := C_0 \alpha_{3-\ell}^6 \left( \Phi_j B_{j,3-\ell}^3 B_{j,\ell}^2 K_{j,\ell} + \Phi_{3-j} E_{3-j,j} B_{3-j,3-\ell}^3 B_{j,\ell}^2 K_{3-j,\ell} \right)$$

$$\Sigma_{j,\ell} := C_0 C_2 \Phi_j \alpha_\ell^6 B_{j,\ell}^3 (B_{j,\ell}^3 + E_{j,3-j} B_{3-j,\ell}^3)$$

Sometimes we need to decompose $\Upsilon_{j,\ell} = \Upsilon_{j,\ell,1} + \Upsilon_{j,\ell,2}$ which is straightforward from its expression. In Section E, we further define

$$\Xi_j^{(t)} = C_0 C_1 \alpha_1^6 \alpha_2^6 \Phi_j^{(t)} \left( (B_{1,1}^{(t)})^6 (B_{2,2}^{(t)})^6 + (B_{2,1}^{(t)})^6 (B_{1,2}^{(t)})^6 \right)$$

$$\Delta_{j,\ell}^{(t)} = C_0 \Phi_j^{(t)} \alpha_\ell^6 (B_{j,\ell}^{(t)})^3 (B_{3-j,\ell}^{(t)})^3 C_2 \mathcal{E}_{j,3-j}^{(t)}$$

for the gradients of the prediction head.

## B.1 Gradient Computation

Let us $L(W, E)$ to be the population version of the objective. Because $\mathbb{E}[F_j(X^{(1)})]$ and $\mathbb{E}[G_j(X^{(2)})]$ are both zero (which can be verified easily from the zero-mean assumptions of $z_p(X)$ and $\xi_p$), a direct computation gives:

$$L(W, E) = 2 - \sum_{j \in [2]} \frac{\mathbb{E}[F_j(X^{(1)}) \cdot \mathsf{StopGrad}[G_j(X^{(2)})]]}{\sqrt{\mathbb{E}[F_j^2(X^{(1)})]} \sqrt{\mathbb{E}[\mathsf{StopGrad}[G_j^2(X^{(2)})]]}}$$

We first calculate the normalizing quantity $\mathbb{E}[F_j^2(X^{(1)})]$:

$$\mathbb{E}[F_j^2(X^{(1)})] = \mathbb{E}\left[\left(\sum_{p\in[P]}\sigma(\langle w_j, X_p^{(1)}\rangle) + E_{j,3-j}\sigma(\langle w_{3-j}, X_p^{(1)}\rangle)\right)^2\right]$$

$$= \frac{1}{2}\sum_{\ell\in[2]}\mathbb{E}\left[|\mathcal{S}(X)\cap\mathcal{P}|^2\alpha_\ell^6(\langle w_j, v_\ell\rangle^3 + E_{j,3-j}\langle w_{3-j}, v_\ell\rangle^3)^2\right]$$

(Because all signal patches has the same sign within the same data)

$$+ \mathbb{E}\left[|\mathcal{P}\setminus\mathcal{S}(X)|(\langle w_j, \xi_p\rangle^3 + E_{j,3-j}\langle w_{3-j}, \xi_p\rangle^3)^2\right]$$

(Because noise patches are independent and have mean zero)

$$= \sum_{\ell\in[2]}\alpha_\ell^6(\langle w_j, v_\ell\rangle^3 + E_{j,3-j}\langle w_{3-j}, v_\ell\rangle^3)^2\frac{\mathbb{E}\left[|\mathcal{S}(X)\cap\mathcal{P}|^2\right]}{2} + (P - |\mathcal{S}(X)|)\mathcal{E}_{j,3-j}$$

where we let

$$\mathcal{E}_{j,3-j} \stackrel{\text{def}}{=} \mathbb{E}\left[(\langle w_j, \xi_p\rangle^3 + E_{j,3-j}\langle w_{3-j}, \xi_p\rangle^3)^2\right]$$
$$= \mathbb{E}\left[\langle w_j, \xi_p\rangle^6 + 2E_{j,3-j}\langle w_j, \xi_p\rangle^3\langle w_{3-j}, \xi_p\rangle^3 + E_{j,3-j}^2\langle w_{3-j}, \xi_p\rangle^6\right]$$

On the other hand, we have

$$\mathbb{E}[F_j(X^{(1)}) \cdot \mathsf{StopGrad}[G_j(X^{(2)})]]$$

$$= \mathbb{E}\left[\left(\sum_{p\in[P]}\sigma(\langle w_j, X_p^{(1)}\rangle) + E_{j,3-j}\sigma(\langle w_{3-j}, X_p^{(1)}\rangle)\right)\times\left(\sum_{p\in[P]}\sigma(\langle w_j, X_p^{(2)}\rangle)\right)\right]$$

$$= \frac{1}{2}\sum_{\ell\in[2]}\mathbb{E}\left[\sum_{p\in|\mathcal{S}(X)\cap\mathcal{P}}\alpha_\ell^3(\langle w_j, v_\ell\rangle^3 + E_{j,3-j}\langle w_{3-j}, v_\ell\rangle^3)\times\sum_{p\in|\mathcal{S}(X)\setminus\mathcal{P}}\alpha_\ell^3\mathsf{StopGrad}[\langle w_j, v_\ell\rangle^3]\right]$$

$$= \sum_{\ell\in[2]}\alpha_\ell^6(\langle w_j, v_\ell\rangle^3 + E_{j,3-j}\langle w_{3-j}, v_\ell\rangle^3)\cdot\mathsf{StopGrad}[\langle w_j, v_\ell\rangle^3]\cdot\frac{\mathbb{E}[|\mathcal{S}(X)\cap\mathcal{P}|\cdot|\mathcal{S}(X)\setminus\mathcal{P}|]}{2}$$

Now, by denoting

$$C_0 = \frac{\mathbb{E}[|\mathcal{S}(X)\cap\mathcal{P}|\cdot|\mathcal{S}(X)\setminus\mathcal{P}|]}{2}, \quad C_1 = \frac{\mathbb{E}\left[|\mathcal{S}(X)\cap\mathcal{P}|^2\right]}{2}, \quad C_2 = P - |\mathcal{S}(X)|,$$
$$\bar{B}_{j,\ell}^3 = \mathsf{StopGrad}[\langle w_j, v_\ell\rangle^3], \qquad B_{j,\ell} = \langle w_j, v_\ell\rangle, \qquad Q_j = (\mathbb{E}[\mathsf{StopGrad}[G_j^2(X^{(2)})]])^{-1/2}.$$

we denote $U_j := \mathbb{E}[F_j^2(X^{(1)})]$, where the expanded expression is

$$U_j = \mathbb{E}[F_j^2(X^{(1)})] = \sum_{\ell\in[2]}C_1\alpha_\ell^6(B_{j,\ell}^3 + E_{j,3-j}B_{3-j,\ell}^3)^2 + C_2\mathcal{E}_{j,3-j}$$

and we can rewrite the objective as follows

$$L(W, E) = 2 - \sum_{j\in[2]}\sum_{\ell\in[2]}\frac{Q_jC_0\alpha_\ell^6\bar{B}_{j,\ell}^3(B_{j,\ell}^3 + E_{j,3-j}B_{3-j,\ell}^3)}{U_j^{1/2}} \tag{B.1}$$

Now denote

$$H_{j,\ell} = C_1\alpha_\ell^6(B_{j,\ell}^3 + E_{j,3-j}B_{3-j,\ell}^3)^2 + C_2\mathcal{E}_{j,3-j},$$
$$K_{j,\ell} = C_1\alpha_\ell^6(B_{j,\ell}^3 + E_{j,3-j}B_{3-j,\ell}^3)(B_{j,3-\ell}^3 + E_{j,3-j}B_{3-j,3-\ell}^3)$$

It is easy to calculate

$$Q_j^{-2} = \mathbb{E}[\mathsf{StopGrad}[G_j^2(X^{(2)})]]$$

$$= \mathbb{E}\left[\left(\sum_{p\in[P]} \sigma(\langle w_j, X_p^{(2)}\rangle)\right)^2\right]$$

$$= \frac{1}{2}\sum_{\ell\in[2]} \alpha_\ell^6\langle w_j, v_\ell\rangle^6\mathbb{E}\left[|\mathcal{S}(X)\cap\mathcal{P}|^2\right] + \mathbb{E}\left[|\mathcal{P}\setminus|\mathcal{S}(X)|\langle w_j, \xi_p\rangle^6\right]$$

$$= \sum_{\ell\in[2]} C_1\alpha_\ell^6 B_{j,\ell}^6 + C_2\mathcal{E}_j$$

where $\mathcal{E}_j = \mathbb{E}[\langle w_j, \xi_p\rangle^6]$. And thus the gradient can be computed as (notice $\bar{B}_{j,\ell}^3 = B_{j,\ell}^3$)

$$-\nabla_{w_j}L(W,E) = \sum_{\ell\in[2]}\left(\frac{C_0 Q_j\alpha_\ell^6 H_{j,3-\ell}B_{j,\ell}^5}{U_j^{3/2}}\right)v_\ell + \sum_{\ell\in[2]}\left(\frac{C_0 Q_{3-j}E_{3-j,j}\alpha_\ell^6 B_{3-j,\ell}^3 B_{j,\ell}^2 H_{3-j,3-\ell}}{U_{3-j}^{3/2}}\right)v_\ell$$

$$- \sum_{\ell\in[2]}\left(\frac{C_0 Q_j\alpha_{3-\ell}^6 B_{j,3-\ell}^3 B_{j,\ell}^2 K_{j,\ell}}{U_j^{3/2}} + \frac{C_0 Q_{3-j}E_{3-j,j}\alpha_{3-\ell}^6 B_{3-j,3-\ell}^3 B_{j,\ell}^2 K_{3-j,\ell}}{U_{3-j}^{3/2}}\right)v_\ell$$

$$- \sum_{j'\in[2]}\sum_{\ell\in[2]} \frac{C_0 C_2 Q_{j'}\alpha_\ell^6 B_{j',\ell}^3(B_{j',\ell}^3 + E_{j',3-j'}B_{3-j',\ell}^3)}{U_{j'}^{3/2}}\nabla_{w_j}\mathcal{E}_{j',3-j'}$$

$$= \sum_{\ell\in[2]}(\Lambda_{j,\ell} + \Gamma_{j,\ell} - \Upsilon_{j,\ell})v_\ell - \sum_{(j',\ell)\in[2]\times[2]}\Sigma_{j',\ell}\nabla_{w_j}\mathcal{E}_{j',3-j'} \tag{B.2}$$

where

$$\nabla_{w_j}\mathcal{E}_{j,3-j} = 6\mathbb{E}[\langle w_j, \xi_p\rangle^5\xi_p + E_{j,3-j}\langle w_j, \xi_p\rangle^2\langle w_{3-j}, \xi_p\rangle^3\xi_p]$$

$$\nabla_{w_j}\mathcal{E}_{3-j,j} = 6\mathbb{E}[E_{3-j,j}^2\langle w_j, \xi_p\rangle^5\xi_p + E_{3-j,j}\langle w_{3-j}, \xi_p\rangle^3\langle w_j, \xi_p\rangle^2\xi_p]$$

As for the gradient of the prediction head, we can calculate

$$-\nabla_{E_{j,3-j}}L(W,E) = \sum_{\ell\in[2]}\frac{C_0 Q_j\alpha_\ell^6 B_{j,\ell}^3 B_{3-j,\ell}^3 U_j}{U_j^{3/2}}$$

$$- \sum_{\ell\in[2]}\frac{C_0 Q_j\alpha_\ell^6 B_{j,\ell}^3(B_{j,\ell}^3 + E_{j,3-j}B_{3-j,\ell}^3)\sum_{\ell'\in[2]}C_1\alpha_{\ell'}^6(B_{j,\ell'}^3 + E_{j,3-j}B_{3-j,\ell'}^3)B_{3-j,\ell'}^3}{U_j^{3/2}}$$

$$- \sum_{\ell\in[2]}\frac{C_0 C_2 Q_j\alpha_\ell^6 B_{j,\ell}^3(B_{j,\ell}^3 + E_{j,3-j}B_{3-j,\ell}^3)}{U_j^{3/2}}\nabla_{E_{j,3-j}}\mathcal{E}_{j,3-j}$$

$$= \sum_{\ell\in[2]}\frac{C_0 Q_j\alpha_\ell^6 B_{j,\ell}^3(B_{3-j,\ell}^3 H_{j,3-\ell} - B_{3-j,3-\ell}^3 K_{j,3-\ell})}{U_j^{3/2}}$$

$$- \sum_{\ell\in[2]}\Sigma_{j,\ell}\mathbb{E}\left[2\langle w_j, \xi_p\rangle^3\langle w_{3-j}, \xi_p\rangle^3 + 2E_{j,3-j}\langle w_{3-j}, \xi_p\rangle^6\right]$$

where $\Sigma_{j,\ell}$ is defined in (B.2). In fact, all the above gradient expressions can be simplified by letting $\Phi_j := Q_j/U_j^{3/2}$ for $j \in [2]$, which is what we shall do in later sections.

**Summarizing the notations.** We shall define some useful notations to simplify the proof. We define $V = \mathrm{span}(v_1, v_2)$. Let $\Pi_A$ be the projection operator to subspace $A \subset \mathbb{R}^d$, then

$$R_j := \langle\Pi_{V^\perp}w_j, w_j\rangle \qquad R_{1,2} := \langle\Pi_{V^\perp}w_1, w_2\rangle \qquad \overline{R}_{1,2} := \frac{\langle\Pi_{V^\perp}w_1, w_2\rangle}{\|\Pi_{V^\perp}w_1\|_2\|\Pi_{V^\perp}w_2\|_2}$$

## B.2 Some Useful Bounds for Gradients

In this section we use the superscript $^{(t)}$ to denote the iteration $t$ during training. Below we present a claim which comes from direct calculations of $\Sigma_{j,\ell}^{(t)}$ and $\nabla_{w_j}\mathcal{E}_{j',3-j'}^{(t)}$, which is very useful in the following sections.

**Claim B.1** (on $\Sigma_{j,\ell}^{(t)}$ and $\nabla_{w_j}\mathcal{E}_{j',3-j'}^{(t)}$). *Let $R_j$, $R_{1,2}^{(t)}$ be defined as above, then we have*

(a) $\Sigma_{j,\ell}^{(t)} = O(\Sigma_{1,1}^{(t)})\dfrac{(B_{j,\ell}^{(t)})^6 + E_{j,3-j}^{(t)}(B_{3-j,\ell}^{(t)})^3(B_{j,\ell}^{(t)})^3}{(B_{1,1}^{(t)})^6}\dfrac{\Phi_j^{(t)}}{\Phi_1^{(t)}};$

(b) $\langle \nabla_{w_j}\mathcal{E}_{j,3-j}^{(t)}, \Pi_{V^\top}w_j^{(t)}\rangle = \Theta([R_j^{(t)}]^3) \pm \Theta(E_{j,3-j}^{(t)})(\overline{R}_{1,2}^{(t)} + \varrho)[R_1^{(t)}]^{3/2}[R_2^{(t)}]^{3/2};$

(c) $\langle \nabla_{w_j}\mathcal{E}_{3-j,j}^{(t)}, w_j^{(t)}\rangle = \Theta((E_{3-j,j}^{(t)})^2)[R_j^{(t)}]^3 \pm O(E_{3-j,j}^{(t)})(\overline{R}_{1,2}^{(t)} + \varrho)[R_1^{(t)}]^{3/2}[R_2^{(t)}]^{3/2}$

(d) $\langle \nabla_{w_j}\mathcal{E}_{j,3-j}^{(t)}, w_{3-j}^{(t)}\rangle = (\Theta(\overline{R}_{1,2}^{(t)}) \pm \varrho)[R_j^{(t)}]^{5/2}[R_{3-j}^{(t)}]^{1/2} + O(E_{j,3-j}^{(t)})R_j^{(t)}[R_{3-j}^{(t)}]^2;$

(e) $\langle \nabla_{w_j}\mathcal{E}_{3-j,j}^{(t)}, w_{3-j}^{(t)}\rangle = ((E_{3-j,j}^{(t)})^2(\Theta(\overline{R}_{1,2}^{(t)}) \pm \varrho)[R_j^{(t)}]^{5/2}[R_{3-j}^{(t)}]^{1/2} + O(E_{3-j,j}^{(t)})R_j^{(t)}[R_{3-j}^{(t)}]^2)$

*Proof.* The part on $\Sigma_{j,\ell}^{(t)}$ is trivial from its expression, we shall focus on proving (b) – (d).
**On $\langle \nabla_{w_j}\mathcal{E}_{j',3-j'}^{(t)}, w_j^{(t)}\rangle$:** If $j = j'$, then

$$\langle \nabla_{w_j}\mathcal{E}_{j,3-j}^{(t)}, w_j^{(t)}\rangle = \Theta(1)\mathbb{E}[\langle w_j^{(t)}, \xi_p\rangle^6 + E_{j,3-j}^{(t)}\langle w_j^{(t)}, \xi_p\rangle^3\langle w_{3-j}^{(t)}, \xi_p\rangle^3]$$
$$= \Theta(1)\mathbb{E}[\langle w_j^{(t)}, \xi_p\rangle^6] + O(E_{j,3-j}^{(t)})\mathbb{E}[\langle w_j^{(t)}, \xi_p\rangle^3(\langle w_{3-j}^{(t)}, \xi_p\rangle^3$$
$$- \langle(I - \bar{w}_{j,t}\bar{w}_{j,t}^\top)w_{3-j}^{(t)}, \xi_p\rangle^3)]$$
$$+ O(E_{j,3-j}^{(t)})\mathbb{E}[\langle w_j^{(t)}, \xi_p\rangle^3\langle(I - \bar{w}_{j,t}\bar{w}_{j,t}^\top)w_{3-j}^{(t)}, \xi_p\rangle^3]$$

Write $\bar{w}_{j,t} = \dfrac{\Pi_{V^\perp}w_j^{(t)}}{\|\Pi_{V^\perp}w_j^{(t)}\|_2}$, we can derive

$$\mathbb{E}[\langle w_j^{(t)}, \xi_p\rangle^3(\langle w_{3-j}^{(t)}, \xi_p\rangle^3 - \langle(I - \bar{w}_{j,t}\bar{w}_{j,t}^\top)w_{3-j}^{(t)}, \xi_p\rangle^3)]$$
$$= \mathbb{E}[\langle w_j^{(t)}, \xi_p\rangle^3\langle\bar{w}_{j,t}\bar{w}_{j,t}^\top w_{3-j}^{(t)}, \xi_p\rangle O(\langle w_{3-j}^{(t)}, \xi_p\rangle^2)]$$
$$= O(\frac{R_{1,2}^{(t)}}{\|\Pi_{V^\perp}w_j^{(t)}\|_2^2})\mathbb{E}[\langle w_j^{(t)}, \xi_p\rangle^4\langle w_{3-j}^{(t)}, \xi_p\rangle^2]$$
$$\leq O(\frac{R_{1,2}^{(t)}}{\|\Pi_{V^\perp}w_j^{(t)}\|_2^2})\mathbb{E}[\langle w_j^{(t)}, \xi_p\rangle^6]^{\frac{2}{3}}\mathbb{E}[\langle w_{3-j}^{(t)}, \xi_p\rangle^6]^{\frac{1}{3}} \qquad \text{(by Hölder's inequality)}$$
$$\leq O(\overline{R}_{1,2}^{(t)})\|\Pi_{V^\perp}w_j^{(t)}\|_2^3\|\Pi_{V^\perp}w_{3-j}^{(t)}\|_2^3$$

and by our assumption on noise $\xi_p$, we also have

$$\mathbb{E}[\langle w_j^{(t)}, \xi_p\rangle^3\langle(I - \bar{w}_{j,t}\bar{w}_{j,t}^\top)w_{3-j}^{(t)}, \xi_p\rangle^3] \leq O(\varrho)\|\Pi_{V^\perp}w_j^{(t)}\|_2^3\|\Pi_{V^\perp}w_{3-j}^{(t)}\|_2^3$$

Combined with the fact that $\mathbb{E}[\langle w_j^{(t)}, \xi_p\rangle^6] = O(\|\Pi_{V^\perp}w_j^{(t)}\|_2^3)$, we can get

$$\langle \nabla_{w_j}\mathcal{E}_{j,3-j}^{(t)}, w_j^{(t)}\rangle = O(\|\Pi_{V^\perp}w_j^{(t)}\|_2^6) \pm O(E_{j,3-j}^{(t)})(R_{1,2}^{(t)} + \varrho)\|\Pi_{V^\perp}w_j^{(t)}\|_2^3\|\Pi_{V^\perp}w_{3-j}^{(t)}\|_2^3$$

when $j' = 3 - j$, we also have

$$\langle \nabla_{w_j}\mathcal{E}_{3-j,j}^{(t)}, w_j^{(t)}\rangle = \Theta(1)\mathbb{E}[(E_{3-j,j}^{(t)})^2\langle w_j^{(t)}, \xi_p\rangle^6 + E_{3-j,j}^{(t)}\langle w_j^{(t)}, \xi_p\rangle^3\langle w_{3-j}^{(t)}, \xi_p\rangle^3]$$
$$= O((E_{3-j,j}^{(t)})^2)\|\Pi_{V^\perp}w_j^{(t)}\|_2^6 \pm O(E_{3-j,j}^{(t)})(R_{1,2}^{(t)} + \varrho)\|\Pi_{V^\perp}w_j^{(t)}\|_2^3\|\Pi_{V^\perp}w_{3-j}^{(t)}\|_2^3$$

**On** $\langle \nabla_{w_j} \mathcal{E}^{(t)}_{j',3-j'}, w^{(t)}_{3-j} \rangle$**:** when $j' = j$, we have

$$
\begin{aligned}
\langle \nabla_{w_j} \mathcal{E}^{(t)}_{j,3-j}, w^{(t)}_{3-j} \rangle &= O(1)\mathbb{E}[\langle w^{(t)}_j, \xi_p \rangle^5 \langle w^{(t)}_{3-j}, \xi_p \rangle + E^{(t)}_{j,3-j} \langle w^{(t)}_j, \xi_p \rangle^2 \langle w^{(t)}_{3-j}, \xi_p \rangle^4] \\
&= O(1)\mathbb{E}[\langle w^{(t)}_j, \xi_p \rangle^5 \langle (I - \bar{w}_{j,t}\bar{w}^\top_{j,t} + \bar{w}_{j,t}\bar{w}^\top_{j,t})w^{(t)}_{3-j}, \xi_p \rangle] \quad\quad (B.3) \\
&\quad + O(1)\mathbb{E}[E^{(t)}_{j,3-j} \langle w^{(t)}_j, \xi_p \rangle^2 \langle w^{(t)}_{3-j}, \xi_p \rangle^4]
\end{aligned}
$$

Using Hölder's inequality and our assumpsion on $\xi_p$, we have

$$
\mathbb{E}[\langle w^{(t)}_j, \xi_p \rangle^5 \langle (I - \bar{w}_{j,t}\bar{w}^\top_{j,t})w^{(t)}_{3-j}, \xi_p \rangle] \lesssim \varrho \|\Pi_{V^\perp} w^{(t)}_j\|^5_2 \|\Pi_{V^\perp} w^{(t)}_{3-j}\|_2
$$

In the meantime, we also have

$$
\mathbb{E}[\langle w^{(t)}_j, \xi_p \rangle^5 \langle \bar{w}_{j,t}\bar{w}^\top_{j,t} w^{(t)}_{3-j}, \xi_p \rangle] = \Theta(\overline{R}^{(t)}_{1,2})\mathbb{E}[\langle w^{(t)}_j, \xi_p \rangle^6][R^{(t)}_j]^{-1/2}[R^{(t)}_{3-j}]^{1/2} = \Theta(\overline{R}^{(t)}_{1,2})[R^{(t)}_j]^{5/2}[R^{(t)}_{3-j}]^{1/2}
$$

for the last term in (B.3), we can also use Hölder's inequality to get

$$
E^{(t)}_{j,3-j}\mathbb{E}[\langle w^{(t)}_j, \xi_p \rangle^2 \langle w^{(t)}_{3-j}, \xi_p \rangle^4] \lesssim E^{(t)}_{j,3-j}\mathbb{E}[\langle w^{(t)}_j, \xi_p \rangle^6]^{1/3}\mathbb{E}[\langle w^{(t)}_{3-j}, \xi_p \rangle^6]^{2/3} \lesssim E^{(t)}_{j,3-j}R^{(t)}_j[R^{(t)}_{3-j}]^2
$$

Therefore, we can combine above analysis to get

$$
\langle \nabla_{w_j} \mathcal{E}^{(t)}_{j,3-j}, w^{(t)}_{3-j} \rangle = (\Theta(\overline{R}^{(t)}_{1,2}) \pm \varrho)[R^{(t)}_j]^{5/2}[R^{(t)}_{3-j}]^{1/2} + O(E^{(t)}_{j,3-j})R^{(t)}_j[R^{(t)}_{3-j}]^2
$$

When $j' = 3 - j$, we also have

$$
\begin{aligned}
\langle \nabla_{w_j} \mathcal{E}^{(t)}_{3-j,j}, w^{(t)}_{3-j} \rangle &= 6\mathbb{E}[(E^{(t)}_{3-j,j})^2 \langle w^{(t)}_j, \xi_p \rangle^5 \langle w^{(t)}_{3-j}, \xi_p \rangle + E^{(t)}_{3-j,j} \langle w^{(t)}_j, \xi_p \rangle^2 \langle w^{(t)}_{3-j}, \xi_p \rangle^4] \\
&= 6(E^{(t)}_{3-j,j})^2(\Theta(\overline{R}^{(t)}_{1,2}) \pm \varrho)[R^{(t)}_j]^{5/2}[R^{(t)}_{3-j}]^{1/2} + E^{(t)}_{3-j,j}R^{(t)}_j[R^{(t)}_{3-j}]^2
\end{aligned}
$$

which proves the claim. $\qquad\square$

## C   Phase I: Learning the Stronger Feature

In this section, we shall discuss the initial phase of learning the stronger feature. Firstly, we establish some properties at the initialization for our induction afterwards.

**Initialization properties.**   We prove the following properties for our network at initialization. Recall our initialization is $w^{(0)}_j \sim \mathcal{N}(0, I_d/d), \forall j \in [2]$ and $E^{(0)} = I_2$.

**Lemma C.1** (properties at initialization). *Recall that without loss of generality we let* $|B^{(0)}_{1,1}| = \max_{j \in [2]} |B^{(0)}_{j,1}|$. *With probability* $1 - o(1)$, *the following holds:*

*(a)* $\|w^{(0)}_j\|^2_2 = 1 \pm \widetilde{O}(\frac{1}{\sqrt{d}})$ *for all* $j \in [2]$, *and* $|\langle w^{(0)}_1, w^{(0)}_2 \rangle| \leq \widetilde{O}(\frac{1}{\sqrt{d}})$;

*(b)* $\max_{j,\ell} |B^{(0)}_{j,\ell}| \leq O(\sqrt{\log d/d})$ *and* $\min_{j,\ell} |B^{(0)}_{j,\ell}| \geq \Omega(\frac{1}{\log d})\max_{j,\ell} |B^{(0)}_{j,\ell}|$;

*(c)* $|B^{(0)}_{1,1}| \geq |B^{(0)}_{2,1}|(1 + \frac{1}{\log d})$;

*(d)* $\mathcal{E}^{(0)}_j = (1 - O(\frac{1}{d^3}))\sigma^6 \|w^{(0)}_j\|^6_2 = \Theta(1)$ *for all* $j \in [2]$;

*(e)* $H^{(0)}_{j,\ell} = C_2\mathcal{E}^{(0)}_j(1 + \widetilde{O}(\frac{1}{\sqrt{d}}))$ *for all* $(j, \ell) \in [2] \times [2]$;

*(f)* $U^{(0)}_j = C_2\mathcal{E}^{(0)}_j(1 + \widetilde{O}(\frac{\alpha^6_1}{\sqrt{d}}))$ *for all* $j \in [2]$;

*(g)* $(Q^{(0)}_j)^{-2} = C_2\mathcal{E}^{(0)}_j(1 + \widetilde{O}(\frac{\alpha^6_1}{\sqrt{d}}))$ *for all* $j \in [2]$;

*(h)* $K^{(0)}_{j,\ell} \leq \widetilde{O}(\alpha^6_\ell/d^3)$ *for all* $(j, \ell) \in [2] \times [2]$.

Let us first introduce a fact about Gaussian ratio distribution without proof.

**Fact C.2** (Gaussian ratio distribution)**.** If $X$ and $Y$ are two independent standard Gaussian variables, then the probability density of $Z = X/Y$ is $p(z) = \frac{1}{\pi(1+z^2)}, z \in (-\infty, \infty)$.

*Proof of Lemma C.1.* a. Norm bound comes from simple $\chi^2$ concentration inequality and our initialization $w_j^{(0)} \sim \mathcal{N}(0, \frac{I_d}{d})$. The inner product bound comes from Gaussian concentration.

b. It is from a direct calculation under our initialization, and some application of Gaussian c.d.f. and a union bound.

c. It is from a probability distribution of Gaussian ratio distribution from Fact C.2 to bound the probability of $|B_{1,1}^{(0)}|/|B_{2,1}^{(0)}| \le (1 + \frac{1}{\log d})$ (WLOG we let $|B_{1,1}^{(0)}| = \max_{j \in [2]} |B_{j,1}^{(0)}|$).

d. It can be directly proven from our assumption on noise $\xi_p$ in the subspace $V^\perp$ and (a).

e. Since at the initialization we have $B_{j,\ell}^{(0)} = \widetilde{O}(\frac{1}{\sqrt{d}}), j, \ell \in [2]$ and $E_{j,3-j}^{(0)} = 0$, it is easy to directly upper bound the errors.

f. Again from $B_{j,\ell}^{(0)} = \widetilde{O}(\frac{1}{\sqrt{d}}), \forall j, \ell \in [2]$ at initialization and a direct upper bound.

g. Proof is similar to (e).

h. Directly from a naive upper bound using (b).

$\square$

## C.1 Induction in Phase I

We define phase I as all iterations $t \le T_1$, where $T_1 := \min\{t : B_{1,1}^{(t)} \ge 0.01\}$, we will prove the existence of $T_1$ at the end of this section. We state the following induction hypotheses, which will hold throughout the phase I:

**Inductions C.3.** *For each $t \le T_1$, all of the followings hold:*

*(a).* $\|w_j^{(t)}\|_2 = \|w_j^{(0)}\|_2 \pm \widetilde{O}(\varrho + \frac{1}{\sqrt{d}})$ *for each $j \in [2]$;*

*(b).* $|B_{1,2}^{(t)}|, |B_{2,1}^{(t)}|, |B_{2,2}^{(t)}| = \widetilde{\Theta}(\frac{1}{\sqrt{d}})$;

*(c).* $|B_{1,1}^{(t)}| \ge \Omega(\frac{1}{\log d}) \max(|B_{1,2}^{(t)}|, |B_{2,2}^{(t)}|, |B_{2,1}^{(t)}|)$;

*(d).* $|E_{1,2}^{(t)}| \le \widetilde{O}(\varrho + \frac{1}{\sqrt{d}}) \frac{\eta_E}{\eta} |B_{1,1}^{(t)}|$ *and* $|E_{2,1}^{(t)}| \le \widetilde{O}(\frac{1}{d})$;

*(e).* $R_1^{(t)}, R_2^{(t)} = \Theta(1), |R_{1,2}^{(t)}| \le \widetilde{O}(\varrho + \frac{1}{\sqrt{d}})$

*Remark* C.4. Since we have chosen $\eta_E \le \eta$ and $\varrho \le \frac{1}{d^{\Omega(1)}}$, Induction C.3d implies $|E_{j,3-j}^{(t)}| = o(1)$ throughout $t \le T_1$.

We shall prove the above induction holds in later sections, but first we need some useful claims assuming our induction holds in this phase.

## C.2 Computing Variables at Phase I

Firstly we establish a claim controlling the noise terms $\mathcal{E}_j, \mathcal{E}_{j,3-j}$ during this phase.

**Claim C.5.** *At each iteration $t \le T_1$, if Induction C.3 holds, then*

*(a)* $\mathcal{E}_1^{(t)} = \mathcal{E}_2^{(t)} \pm O(\sum_{\ell \in [2]} |B_{j,\ell}^{(t)}| + \widetilde{O}(\varrho + \frac{1}{\sqrt{d}}))$

*(b)* $\mathcal{E}_j^{(t)} = \mathcal{E}_j^{(0)} \pm O(\sum_{\ell \in [2]} |B_{j,\ell}^{(t)}| + \widetilde{O}(\varrho + \frac{1}{\sqrt{d}}))$

(c) $\mathcal{E}^{(t)}_{j,3-j} = \mathcal{E}^{(t)}_j \pm \widetilde{O}(E^{(t)}_{j,3-j}(\varrho + \frac{1}{\sqrt{d}}) + (E^{(t)}_{j,3-j})^2)$;

*Proof.* For (a), we can simply write down

$$\mathcal{E}^{(t)}_j = \mathbb{E}[\langle w_j, \xi_p \rangle^6] = \sigma^6 \|\Pi_{V^\perp} w^{(t)}_j\|_2^6$$

Note that by Induction C.3a we always have $\|w^{(t)}_j\|_2 = \|w^{(0)}_j\|_2 \pm \widetilde{O}(\varrho + \frac{1}{\sqrt{d}})$, and by Lemma C.1a we also have $\|w^{(0)}_j\|_2 = (1 \pm \widetilde{O}(\frac{1}{\sqrt{d}}))\|w^{(0)}_j\|_2$, which implies

$$\|\Pi_{V^\perp} w^{(t)}_j\|_2 - \|\Pi_{V^\perp} w^{(t)}_{3-j}\|_2 = \|w^{(t)}_j\|_2 - \|w^{(t)}_{3-j}\|_2 \pm O(\sum_{j,\ell \in [2]^2} B^{(t)}_{j,\ell})$$

$$= \|w^{(0)}_j\|_2 - \|w^{(0)}_{3-j}\|_2 \pm O(\sum_{j,\ell \in [2]^2} B^{(t)}_{j,\ell}) \pm \widetilde{O}(\varrho + \frac{1}{\sqrt{d}})$$

$$= \widetilde{O}(\frac{1}{\sqrt{d}}) \pm O(\sum_{j,\ell \in [2]^2} B^{(t)}_{j,\ell}) \pm \widetilde{O}(\varrho + \frac{1}{\sqrt{d}})$$

By the elementary equality $x^n - y^n = (x - y)\sum_{0 \le i \le n-1} x^i y^{n-1-i}$, we can obtain (a). The proof of (b) is almost the same as (a), and the proof of (c) is just direct calculation. □

Equipped with Claim C.5, we can establish the following lemma, which will be frequently applied to bound the gradient in our induction argument.

**Lemma C.6** (variables control in phase I). *Suppose Induction C.3 holds at some iteration $t \le T_1$, then we have:*

(a) *if $\forall \ell \in [2], \alpha_\ell |B^{(t)}_{j,\ell}| \le O(1)$, then $\Phi^{(t)}_j = (C_2 \mathcal{E}^{(t)}_j)^{-2}(1 \pm \frac{1}{\text{polylog}(d)})$;*

(b) *if $\exists \ell \in [2], |B^{(t)}_{j,\ell}| \ge \Omega(\frac{1}{\alpha_\ell})$, then $\Phi^{(t)}_j = O((C_2 \mathcal{E}^{(t)}_j + \sum_{\ell \in [2]} C_1 \alpha^6_\ell (B^{(t)}_{j,\ell})^6)^{-2})$;*

(c) *if $\alpha_\ell |B^{(t)}_{j,\ell}| \le O(1)$, $H^{(t)}_{j,\ell} = C_2 \mathcal{E}^{(t)}_j(1 + \frac{1}{\text{polylog}(d)}) = \Theta(C_2)$, otherwise $H^{(t)}_{j,\ell} \in [\Omega(C_2), \widetilde{O}(\alpha^6_\ell)]$*

(d) *$|K^{(t)}_{j,\ell}| \le \widetilde{O}(\alpha^6_\ell / d^{3/2})$*

*Proof.* (a) From our assumptions that $|B^{(t)}_{1,2}|, |B^{(t)}_{2,1}|, |B^{(t)}_{2,2}| \le \widetilde{O}(\frac{1}{\sqrt{d}})$ and $\alpha_1 B^{(t)}_{1,1} \le O(1)$, and also the fact that $\mathcal{E}^{(t)}_j = \Omega(\sigma^6) = \Omega(1)$, $C_2 = \Theta(\text{polylog}(d)) \gg C_1$, we can calculate

$$U^{(t)}_j = \sum_{\ell \in [2]} C_1 \alpha^6_\ell ((B^{(t)}_{j,\ell})^3 + E^{(t)}_{j,3-j}(B^{(t)}_{3-j,\ell})^3)^2 + C_2 \mathcal{E}^{(t)}_{j,3-j}$$

$$= O(C_1) + C_2 \mathcal{E}^{(t)}_j + \widetilde{O}(\varrho + \frac{1}{\sqrt{d}})$$

$$= C_2 \mathcal{E}^{(t)}_j(1 \pm \frac{1}{\text{polylog}(d)})$$

Meanwhile, we can also compute similarly

$$Q^{(t)}_j = \sum_{\ell \in [2]} C_1 \alpha^6_\ell (B^{(t)}_{j,\ell})^6 + C_2 \mathcal{E}_j = C_2 \mathcal{E}^{(t)}_j(1 \pm \frac{1}{\text{polylog}(d)})$$

Therefore $\Phi^{(t)}_j = Q^{(t)}_j / (U^{(t)}_j)^{3/2} = (C_2 \mathcal{E}^{(t)}_j(1 \pm \frac{1}{\text{polylog}(d)}))^{-2}$ as desired.

(b) The proof is similar to that of (a).

(c) when $\alpha_1 B^{(t)}_{1,1} \le O(1)$, the proof is similar to (a). When $\alpha_1 B^{(t)}_{1,1} \ge O(1)$, we have from Induction C.3a and $H^{(t)}_{j,\ell}$'s expression that

$$H^{(t)}_{j,\ell} = C_1 \alpha^6_\ell ((B^{(t)}_{j,\ell})^3 + E^{(t)}_{j,3-j}(B^{(t)}_{3-j,\ell})^3)^2 + C_2 \mathcal{E}^{(t)}_{j,3-j} \le \widetilde{O}(\alpha^6_\ell)$$

And since $T_1 := \min\{t : B_{1,1}^{(t)} \geq 0.01\}$, so for $t \leq T_1$, we have

$$H_{j,\ell}^{(t)} \geq C_2 \mathcal{E}_{j,3-j}^{(t)} \overset{①}{\geq} C_2 \mathcal{E}_j^{(t)} - |E_{j,3-j}^{(t)}| \overset{②}{\geq} \Omega(C_2)$$

where ① is from Claim C.5b and ② is from Induction C.3d.

(d) Since we have assumed $|B_{1,2}^{(t)}|, |B_{2,1}^{(t)}|, |B_{2,2}^{(t)}| \leq \widetilde{O}(\frac{1}{\sqrt{d}})$, it is direct to bound $|K_{j,\ell}^{(t)}| \leq \widetilde{O}(\alpha_\ell^6/d^{1.5})$.

$\square$

**Claim C.7** (about $\Sigma_{j,\ell}^{(t)}$ and $\nabla_{w_j} \mathcal{E}_{j',3-j'}^{(t)}$). *If Induction C.3 holds at iteration $t \leq T_1$, then*

(a) $\Sigma_{j,\ell}^{(t)} = O(\Lambda_{1,1}^{(t)} B_{1,1}^{(t)}) \frac{(B_{j,\ell}^{(t)})^6 + E_{j,3-j}^{(t)}(B_{3-j,\ell}^{(t)})^3 (B_{j,\ell}^{(t)})^3}{(B_{1,1}^{(t)})^6} \frac{\Phi_j^{(t)}}{\Phi_1^{(t)}}$;

(b) $\langle \nabla_{w_j} \mathcal{E}_{j,3-j}^{(t)}, w_j^{(t)} \rangle = O(1) \pm O(E_{j,3-j}^{(t)})(R_{1,2}^{(t)} + \varrho)$;

(c) $\langle \nabla_{w_j} \mathcal{E}_{3-j,j}^{(t)}, w_j^{(t)} \rangle = O((E_{3-j,j}^{(t)})^2) \pm O(E_{3-j,j}^{(t)})(R_{1,2}^{(t)} + \varrho)$

(d) $|\langle \nabla_{w_j} \mathcal{E}_{j,3-j}^{(t)}, w_{3-j}^{(t)} \rangle| = O(R_{1,2}^{(t)} + \varrho) + O(E_{j,3-j}^{(t)})$;

(e) $|\langle \nabla_{w_j} \mathcal{E}_{3-j,j}^{(t)}, w_{3-j}^{(t)} \rangle| = O(R_{1,2}^{(t)} + \varrho)(E_{3-j,j}^{(t)})^2 + O(E_{3-j,j}^{(t)})$

*Proof.* Notice that $\|\Pi_{V^\perp} w_j^{(t)}\|_2 = \Theta(1), \forall j \in [2]$ for $t \leq T_1$, which is because of $\|w_j^{(t)}\|_2 = \sqrt{2} \pm o(1)$ from Induction C.3a and $\max_{j,\ell} |B_{j,\ell}^{(t)}| < 0.02^4$. Now we can apply Claim B.1 to obtain the bounds. $\square$

## C.3 Gradient Lemmas for Phase I

We first present an interesting lemma regarding the effects of Batch-Normalization on the gradients of weights. The following lemma allow us maintain the norm of weights to above a constant throughout phase I.

**Lemma C.8** (effects of BN on gradients). *For any $W = (w_1, w_2)$ and $E$, it holds*

(a) $\sum_{j \in [2]} \langle \nabla_{w_j} L(W, E), w_j \rangle = 0$;

*Further, if Induction C.3 holds for each $t \leq T_1$, we have*

(b) $|\langle \nabla_{w_j} L(W^{(t)}, E^{(t)}), w_j^{(t)} \rangle| \leq \widetilde{O}(\varrho + \frac{1}{\sqrt{d}})|\Lambda_{1,1}| \sum_{j \in [2]} |E_{j,3-j}^{(t)}|$ *for each $j \in [2]$.*

*Proof.* **Proof of (a):** We first calculate the gradient term as follows:

$$\nabla_W L(W, E) = \nabla_W \sum_{j \in [2]} \frac{\mathbb{E}[F_j(X^{(1)}) \cdot \mathsf{StopGrad}[G_j(X^{(2)})]]}{\sqrt{\mathbb{E}[F_j^2(X^{(1)})]}\sqrt{\mathbb{E}[\mathsf{StopGrad}[G_j^2(X^{(2)})]]}}$$

$$= \sum_{j \in [2]} \frac{\mathbb{E}[(\nabla_W F_j(X^{(1)})) \cdot [G(X^{(2)})]_j] \cdot \mathbb{E}[F_j^2(X^{(1)})]}{(\mathbb{E}[F_j^2(X^{(1)})])^{3/2}\sqrt{\mathbb{E}[G_j^2(X^{(2)})]}}$$

$$- \sum_{j \in [2]} \frac{\mathbb{E}[(\nabla_W F_j(X^{(1)})) \cdot F_j(X^{(1)})] \cdot \mathbb{E}[[F_j(X^{(1)}) \cdot [G(X^{(2)})]_j]}{(\mathbb{E}[F_j^2(X^{(1)})])^{3/2}\sqrt{\mathbb{E}[G_j^2(X^{(2)})]}}$$

Since by our definition $\langle \nabla_W F_j(X^{(1)}), W \rangle = \sum_{i \in [2]} \langle \nabla_{w_i}[F_j(X^{(1)}), w_i \rangle = 3[F_j(X^{(1)})$, we immediately have $\sum_{j \in [2]} \langle \nabla_{w_j} L(W, E), w_j \rangle = 0$.

---

[4]due to our choice of $\eta = \frac{1}{\mathrm{poly}(d)}$ is small, we can make sure when $T_1 = \min\{t : B_{1,1}^{(t)} \geq 0.01\}$, $B_{1,1}^{(T_1)} < 0.02$.

**Proof of (b):** Firstly we define a new notion

$$\nabla_{i,j} = \nabla_{w_i} \frac{\mathbb{E}[F_j(X^{(1)}) \cdot \mathsf{StopGrad}[G_j(X^{(2)})]]}{\sqrt{\mathbb{E}[F_j^2(X^{(1)})]}\sqrt{\mathbb{E}[\mathsf{StopGrad}[G_j^2(X^{(2)})]]}}$$

Then it is straghtforward to verify that $\sum_{i\in[2]}\langle\nabla_{i,j}, w_i\rangle = 0$ for any $j \in [2]$, which implies that $|\langle\nabla_{j',j}, w_{j'}\rangle| = |\langle\nabla_{3-j',j}, w_{3-j'}\rangle|$. So in order to obtain an upper bound for $|\langle\nabla_{w_j}L(W, E), w_j\rangle| = |\sum_{j'\in[2]}\langle\nabla_{j,j'}, w_j\rangle|$, we only need to upper bound $|\langle\nabla_{j,j'}, w_{3-j'}\rangle|$, each of which can be calculated as (ignoring all time superscript $^{(t)}$)

$$|\langle\nabla_{3-j,j}, w_{3-j}\rangle| = \frac{\mathbb{E}\left[\sum_{p\in[P]\cap\mathcal{P}} E_{j,3-j}\sigma(\langle w_{3-j}, X_p\rangle) \cdot [G(X^{(2)})]_j\right] \cdot \mathbb{E}[F_j^2(X^{(1)})]}{(\mathbb{E}[F_j^2(X^{(1)})])^{3/2}\sqrt{\mathbb{E}[G_j^2(X^{(2)})]}}$$

$$- \frac{\mathbb{E}\left[\sum_{p\in[P]\cap\mathcal{P}} E_{j,3-j}\sigma(\langle w_{3-j}, X_p\rangle) \cdot F_j(X^{(1)})\right] \cdot \mathbb{E}[[F_j(X^{(1)}) \cdot [G(X^{(2)})]_j]}{(\mathbb{E}[F_j^2(X^{(1)})])^{3/2}\sqrt{\mathbb{E}[G_j^2(X^{(2)})]}}$$

Now we compute

$$\mathbb{E}\left[\sum_{p\in[P]\cap\mathcal{P}} E_{j,3-j}\sigma(\langle w_{3-j}, X_p\rangle)[G(X^{(2)})]_j\right] = \mathbb{E}\left[\sum_{p\in[P]\cap\mathcal{P}} E_{j,3-j}\sigma(\langle w_{3-j}, X_p\rangle) \sum_{p\in[P]\backslash\mathcal{P}} \sigma(\langle w_j, X_p\rangle)\right]$$

$$= \sum_{\ell\in[2]} E_{j,3-j}C_0\alpha_\ell^6 B_{3-j,\ell}^3 B_{j,\ell}^3$$

and

$$\mathbb{E}\left[\sum_{p\in[P]\cap\mathcal{P}} E_{j,3-j}\sigma(\langle w_{3-j}, X_p\rangle) \cdot F_j(X^{(1)})\right]$$

$$= \mathbb{E}\left[\sum_{p\in[P]\cap\mathcal{P}} E_{j,3-j}\sigma(\langle w_{3-j}, X_p\rangle) \cdot \sum_{p\in[P]\cap\mathcal{P}} (\sigma(\langle w_j, X_p\rangle) + E_{j,3-j}\sigma(\langle w_{3-j}, X_p\rangle))\right]$$

$$= \sum_{\ell\in[2]} E_{j,3-j}C_1\alpha_\ell^6 B_{3-j,\ell}^3(B_{j,\ell}^3 + E_{j,3-j}B_{3-j,\ell}^3) + C_2 E_{j,3-j}\mathbb{E}[\langle w_j, \xi_p\rangle^3\langle w_{3-j}, \xi_p\rangle^3 + E_{j,3-j}\langle w_{3-j}, \xi_p\rangle^6]$$

So we can further obtain the nominator in the expression of $|\langle \nabla_{3-j,j}, w_{3-j} \rangle|$ as

$$
\mathbb{E}\left[ \sum_{p \in [P] \cap \mathcal{P}} E_{j,3-j} \sigma(\langle w_{3-j}, X_p \rangle) \cdot [G(X^{(2)})]_j \right] \cdot \mathbb{E}[F_j^2(X^{(1)})]
$$

$$
- \mathbb{E}\left[ \sum_{p \in [P] \cap \mathcal{P}} E_{j,3-j} \sigma(\langle w_{3-j}, X_p \rangle) \cdot F_j(X^{(1)}) \right] \cdot \mathbb{E}[[F_j(X^{(1)})] \cdot [G(X^{(2)})]_j]
$$

$$
= \left( \sum_{\ell \in [2]} E_{j,3-j} C_0 \alpha_\ell^6 B_{3-j,\ell}^3 B_{j,\ell}^3 \right) \cdot \left( \sum_{\ell \in [2]} C_1 \alpha_\ell^6 (B_{j,\ell}^3 + E_{j,3-j} B_{3-j,\ell}^3)^2 + C_2 \mathcal{E}_{j,3-j} \right)
$$

$$
- \left( \sum_{\ell \in [2]} E_{j,3-j} C_1 \alpha_\ell^6 B_{3-j,\ell}^3 (B_{j,\ell}^3 + E_{j,3-j} B_{3-j,\ell}^3) \right) \cdot \left( \sum_{\ell \in [2]} C_0 \alpha_\ell^6 B_{j,\ell}^3 (B_{j,\ell}^3 + E_{j,3-j} B_{3-j,\ell}^3) \right)
$$

$$
- C_2 E_{j,3-j} \mathbb{E}[\langle w_j, \xi_p \rangle^3 \langle w_{3-j}, \xi_p \rangle^3 + E_{j,3-j} \langle w_{3-j}, \xi_p \rangle^6] \cdot \left( \sum_{\ell \in [2]} C_0 \alpha_\ell^6 B_{j,\ell}^3 (B_{j,\ell}^3 + E_{j,3-j} B_{3-j,\ell}^3) \right)
$$

$$
= E_{j,3-j} \sum_{\ell \in [2]} C_0 \alpha_\ell^6 B_{3-j,\ell}^3 (B_{j,\ell}^3 H_{j,3-\ell} - B_{j,3-\ell}^3 K_{j,3-\ell})
$$

$$
- C_2 E_{j,3-j} \mathbb{E}[\langle w_j, \xi_p \rangle^3 \langle w_{3-j}, \xi_p \rangle^3 + E_{j,3-j} \langle w_{3-j}, \xi_p \rangle^6] \cdot \left( \sum_{\ell \in [2]} C_0 \alpha_\ell^6 B_{j,\ell}^3 (B_{j,\ell}^3 + E_{j,3-j} B_{3-j,\ell}^3) \right)
$$

Now can sum over $j' \in [2]$ to get

$$
|\langle \nabla_{w_j} L(W, E), w_j \rangle|
$$

$$
\leq \sum_{j \in [2]} \sum_{\ell \in [2]} C_0 E_{j,3-j} \left| \Phi_j \alpha_\ell^6 B_{3-j,\ell}^3 B_{j,\ell}^3 H_{j,3-\ell} \right| + \sum_{j \in [2]} \sum_{\ell \in [2]} \left| C_0 E_{j,3-j} \Phi_j \alpha_\ell^3 B_{3-j,\ell}^3 B_{j,3-\ell}^3 K_{j,3-\ell} \right|
$$

$$
+ \sum_{j \in [2]} \sum_{\ell \in [2]} \left| C_2 E_{j,3-j} \Phi_j \mathbb{E}[\langle w_j, \xi_p \rangle^3 \langle w_{3-j}, \xi_p \rangle^3 + E_{j,3-j} \langle w_{3-j}, \xi_p \rangle^6] C_0 \alpha_\ell^6 B_{j,\ell}^3 (B_{j,\ell}^3 + E_{j,3-j} B_{3-j,\ell}^3) \right|
$$

Next we are going to bound each term, for the first term of LHS we have

$$
\sum_{j \in [2]} \sum_{\ell \in [2]} \left| C_0 E_{j,3-j} \Phi_j \alpha_\ell^6 B_{3-j,\ell}^3 B_{j,\ell}^3 H_{j,3-\ell} \right| \leq \sum_{j \in [2]} \sum_{\ell \in [2]} |E_{j,3-j}| |\Lambda_{j,\ell}| \left| \frac{B_{3-j,\ell}^3}{B_{j,\ell}^2} \right|
$$

$$
\leq |\Lambda_{1,1}| \sum_{j \in [2]} |E_{j,3-j}| \left| \frac{B_{3-j,\ell}^3 B_{j,\ell}^3 \Phi_j}{B_{1,1}^5 \Phi_1} \right|
$$

$$
\leq \widetilde{O}\left( \frac{d^{o(1)}}{\sqrt{d}} \right) |\Lambda_{1,1}| \sum_{j \in [2]} |E_{j,3-j}|
$$

where the last inequality is because

- By Lemma C.6a,b, we have $\Phi_j^{(t)} / \Phi_1^{(t)} \leq O(\alpha_1^{O(1)}) \leq d^{o(1)}$ during $t \leq T_1$.

- $(B_{3-j,\ell}^{(t)})^3 (B_{j,\ell}^{(t)})^3 \leq \widetilde{O}(\frac{1}{\sqrt{d}})(B_{1,1}^{(t)})^5$ from Induction C.3b,c.

Similarly, we can also compute

$$
\sum_{j \in [2]} \sum_{\ell \in [2]} \left| C_0 E_{j,3-j} \Phi_j \alpha_\ell^3 B_{3-j,\ell}^3 B_{j,3-\ell}^3 K_{j,3-\ell} \right| \leq \sum_{j \in [2]} \sum_{\ell \in [2]} E_{j,3-j} |\Lambda_{1,1}| \left| \frac{B_{3-j,\ell}^3 B_{j,3-\ell}^3 K_{j,3-\ell}}{B_{1,1}^5 H_{j,3-\ell}} \right|
$$

$$
\leq \widetilde{O}\left( \frac{d^{o(1)}}{d^2} \right) |\Lambda_{1,1}| \sum_{j \in [2]} |E_{j,3-j}|
$$

and

$$\sum_{j\in[2]}\sum_{\ell\in[2]}\left|C_2 E_{j,3-j}\Phi_j\mathbb{E}[\langle w_j,\xi_p\rangle^3\langle w_{3-j},\xi_p\rangle^3 + E_{j,3-j}\langle w_{3-j},\xi_p\rangle^6]C_0\alpha_\ell^6 B_{j,\ell}^3(B_{j,\ell}^3 + E_{j,3-j}B_{3-j,\ell}^3)\right|$$

$$\overset{①}{\leq}\sum_{j\in[2]}\sum_{\ell\in[2]}|E_{j,3-j}\Lambda_{j,\ell}|\left|\frac{B_{j,\ell}^3 + E_{j,3-j}B_{3-j,\ell}^3}{B_{j,\ell}^2}\right|\left|\mathbb{E}[\langle w_j,\xi_p\rangle^3\langle w_{3-j},\xi_p\rangle^3 + E_{j,3-j}\langle w_{3-j},\xi_p\rangle^6]\right|$$

$$\overset{②}{\leq}\sum_{j\in[2]}\sum_{\ell\in[2]}|E_{j,3-j}\Lambda_{j,\ell}|\left|\frac{B_{j,\ell}^3 + E_{j,3-j}B_{3-j,\ell}^3}{B_{j,\ell}^2}\right|(O(R_{1,2}+\varrho)+O(E_{j,3-j}))$$

$$\leq\widetilde{O}(R_{1,2}+\varrho)|\Lambda_{1,1}|\sum_{j\in[2]}|E_{j,3-j}|$$

where ① is due to Lemma C.6c, ② is from the same calculation in Claim C.7 for $\mathbb{E}[\langle w_j,\xi_p\rangle^3\langle w_{3-j},\xi_p\rangle^3]$ and Induction C.3a. Now combining the above and Induction C.3e together we have

$$|\langle\nabla_{w_j}L(W,E),w_j\rangle|\leq\widetilde{O}(\varrho+\frac{1}{\sqrt{d}})|\Lambda_{1,1}|\sum_{j\in[2]}|E_{j,3-j}|$$

which gives the desired bound. $\qquad\square$

Next we give a lemma characterizing the gradient of feature $v_1$ in this phase.

**Lemma C.9** (learning feature $v_1$ in phase I). *For each $t\leq T_1$, if Induction C.3 holds at iteration $t$, then using notations of* (B.2), *we have:*

*(a)* $\langle-\nabla_{w_1}L(W^{(t)},E^{(t)}),v_1\rangle = (1\pm\widetilde{O}(\frac{1}{d}))\Lambda_{1,1}^{(t)}$

*(b)* $\langle-\nabla_{w_2}L(W^{(t)},E^{(t)}),v_1\rangle = (1\pm O(\frac{1}{\sqrt{d}}))\Lambda_{2,1}^{(t)}+\Gamma_{2,1}^{(t)}\leq(1\pm O(\frac{1}{\sqrt{d}}))\Lambda_{2,1}^{(t)}\pm\frac{(B_{2,1}^{(t)})^2}{(B_{1,1}^{(t)})^2}E_{1,2}^{(t)}\Lambda_{1,1}^{(t)}$

*Proof.* From (B.2), we write down the gradient formula for $B_{j,1}^{(t)}$ as follows:

$$\langle-\nabla_{w_j}L_{\mathcal{D}}(W^{(t)},E^{(t)}),v_1\rangle = \Lambda_{j,1}^{(t)}+\Gamma_{j,1}^{(t)}-\Upsilon_{j,1}^{(t)}$$

where (ignoring the superscript $^{(t)}$ for the RHS)

$$\Lambda_{j,1}^{(t)} = C_0\Phi_j\alpha_1^6 B_{j,1}^5 H_{j,2}$$
$$\Gamma_{j,1}^{(t)} = C_0\Phi_{3-j}E_{3-j,j}\alpha_1^6 B_{3-j,1}^3 B_{j,1}^2 H_{3-j,2}$$
$$\Upsilon_{j,1}^{(t)} = C_0\alpha_2^6\left(\Phi_j B_{j,2}^3 B_{j,1}^2 K_{j,1}+\Phi_{3-j}E_{3-j,j}B_{3-j,2}^3 B_{j,1}^2 K_{3-j,1}\right)$$

We first prove (a), and we deal with each term individually:
**Comparing $\Lambda_{1,1}^{(t)}$ and $\Gamma_{1,1}^{(t)}$:** When $t\leq T_{1,1}$, we have from Lemma C.6a that

$$\Phi_1^{(t)}H_{1,2}^{(t)} = \frac{1}{C_2\mathcal{E}_1^{(t)}}(1\pm\frac{1}{\mathsf{polylog}(d)}) = \frac{1}{C_2\mathcal{E}_2^{(t)}}(1\pm\frac{1}{\mathsf{polylog}(d)}) = \Phi_2^{(t)}H_{2,2}^{(t)}(1\pm\frac{1}{\mathsf{polylog}(d)})$$

Further, by Induction C.3b,c,d and our definition of stage 1, we know $E_{1,2}^{(t)}\leq\widetilde{O}(\frac{1}{d})$. Now from Induction C.3b that $B_{2,1}^{(t)}\leq\widetilde{O}(\frac{1}{\sqrt{d}})$, together we have

$$\Gamma_{1,1}^{(t)} = C_0\alpha_1^6 E_{2,1}^{(t)}\Phi_2^{(t)}H_{2,2}^{(t)}(B_{2,1}^{(t)})^3(B_{1,1}^{(t)})^2\leq\widetilde{O}(\frac{1}{d})C_0\alpha_1^6\Phi_1^{(t)}H_{1,2}^{(t)}(B_{1,1}^{(t)})^5 = \widetilde{O}(\frac{\Lambda_{1,1}^{(t)}}{d})$$

When $t\in[T_{1,1},T_1]$, by Lemma C.6b we have

$$\Phi_1^{(t)}H_{1,2}^{(t)}\geq\Omega(\frac{C_2}{(C_1\alpha_1^6(B_{1,1}^{(t)})^6+O(C_2))^2})\geq\omega(\frac{1}{d^{0.1}}),\quad\text{and}\quad E_{2,1}^{(t)}\Phi_2^{(t)}H_{2,2}^{(t)}\leq\widetilde{O}(\frac{1}{d})$$

Now from our definition of stage 2, it holds that $B_{1,1}^{(t)} \geq \Omega(\frac{1}{\alpha_1})$ while $B_{2,1}^{(t)} \leq \widetilde{O}(\frac{1}{\sqrt{d}})$ by Induction C.3b, which gives

$$\Gamma_{1,1}^{(t)} = C_0 \alpha_1^6 E_{2,1}^{(t)} \Phi_2^{(t)} H_{2,2}^{(t)} (B_{2,1}^{(t)})^3 (B_{1,1}^{(t)})^2 \leq \widetilde{O}(\frac{1}{d}) C_0 \alpha_1^6 \Phi_1^{(t)} H_{1,2}^{(t)} (B_{1,1}^{(t)})^5 = \widetilde{O}(\frac{\Lambda_{1,1}^{(t)}}{d})$$

**Comparing $\Lambda_{1,1}^{(t)}$ and $\Upsilon_{1,1}^{(t)}$:** Now consider $\Upsilon_{1,1}^{(t)}$, by Lemma C.6, we can follow the same analysis as above to get

$$\Phi_j^{(t)} K_{j,\ell}^{(t)} \leq \widetilde{O}(\frac{\alpha_1^{O(1)}}{d^{3/2}}) \Phi_1^{(t)} H_{1,2}^{(t)} \qquad\qquad \text{for any } (j,\ell) \in [2] \times [2]$$

Combined with $E_{2,1}^{(t)} \leq o(1)$, we can derive

$$\Upsilon_{1,1}^{(t)} = C_0 \alpha_2^6 \left( \Phi_1^{(t)} K_{1,1}^{(t)} (B_{1,2}^{(t)})^3 (B_{1,1}^{(t)})^2 + E_{1,2}^{(t)} \Phi_2^{(t)} K_{2,1}^{(t)} (B_{2,2}^{(t)})^3 (B_{1,1}^{(t)})^2 \right)$$

$$\leq \widetilde{O}(\frac{\alpha_1^{O(1)} \alpha_2^6}{d^{3/2}}) C_0 \alpha_1^6 \Phi_1^{(t)} H_{1,2}^{(t)} (B_{1,1}^{(t)})^5$$

$$= \widetilde{O}(\frac{\Lambda_{1,1}^{(t)}}{d^{3/2-o(1)}}) \qquad\qquad (\text{since } C_1 = \widetilde{O}(1) \text{ and } \alpha_1, \alpha_2 = d^{o(1)})$$

**Comparing $\Lambda_{2,1}^{(t)}$ and $\Upsilon_{2,1}^{(t)}$:** Till now (a) is proved, we can deal with (b) by only comparing $\Lambda_{2,1}^{(t)}$ with $\Upsilon_{2,1}^{(t)}$. Similar to the above arguments, we have by Induction C.3b we know $K_{j,1}^{(t)} = \widetilde{O}(\frac{C_1 \alpha_1^6}{d^{3/2}}), \forall j \in [2]$, and thus

$$\Phi_j^{(t)} K_{j,\ell}^{(t)} \leq \widetilde{O}(\frac{\alpha_1^6}{d^{3/2}}) \Phi_2^{(t)} H_{2,2}^{(t)} \qquad\qquad \text{for any } (j,\ell) \in [2] \times [2]$$

By Induction C.3e we know $E_{1,2}^{(t)} \leq \widetilde{O}(\varrho + \frac{1}{\sqrt{d}})$. Also, note that from Induction C.3b we have $\widetilde{O}((B_{1,2}^{(t)})^3/d) \leq \widetilde{O}((B_{2,1}^{(t)})^5)$, and thus

$$E_{1,2}^{(t)} \Phi_1^{(t)} K_{1,1}^{(t)} (B_{1,2}^{(t)})^3 (B_{2,1}^{(t)})^2 \leq \widetilde{O}(\varrho + \frac{1}{\sqrt{d}}) \widetilde{O}(\frac{\alpha_1^6}{d^{5/2}}) \Phi_2^{(t)} H_{2,2}^{(t)} \widetilde{O}(B_{1,2}^{(t)})^3 \leq O(\frac{1}{d^{3/2}}) \Phi_2^{(t)} H_{2,2}^{(t)} (B_{2,1}^{(t)})^5$$

So together we have

$$|\Upsilon_{2,1}^{(t)}| = |C_0 \alpha_2^6 \left( \Phi_2^{(t)} K_{2,1}^{(t)} (B_{2,2}^{(t)})^3 (B_{2,1}^{(t)})^2 + E_{2,1}^{(t)} \Phi_1^{(t)} K_{1,1}^{(t)} (B_{1,2}^{(t)})^3 (B_{2,1}^{(t)})^2 \right)|$$

$$\leq O(\frac{1}{d^{3/2}}) C_0 \alpha_1^6 \Phi_2^{(t)} H_{2,2}^{(t)} |(B_{2,1}^{(t)})^5|$$

$$= O(\frac{1}{d^{3/2}}) |\Lambda_{2,1}^{(t)}|$$

**Comparing $\Gamma_{2,1}^{(t)}$ with $\Lambda_{1,1}^{(t)}$:** It suffices to notice that

$$|\Gamma_{2,1}^{(t)}| \leq |E_{1,2}^{(t)}| C_0 \alpha_1^6 \Phi_1^{(t)} H_{1,2}^{(t)} |B_{1,1}^{(t)}|^3 (B_{2,1}^{(t)})^2 = \frac{(B_{2,1}^{(t)})^2}{(B_{1,1}^{(t)})^2} |E_{1,2}^{(t)}| |\Lambda_{1,1}^{(t)}|$$

Combining the bounds for $\Lambda_{2,1}^{(t)}$ and $\Gamma_{2,1}^{(t)}$, we obtain the proof of (b). $\qquad\square$

Then we can also calculate the gradients of feature $v_2$ in this phase.

**Lemma C.10** (learning feature $v_2$ in phase I). *For each $t \leq T_1$, if Induction C.3 holds at iteration $t$, then using notations of* (B.2)*, we have for each $j \in [2]$:*

$$\langle -\nabla_{w_j} L(W^{(t)}, E^{(t)}), v_2 \rangle = \left( 1 \pm \widetilde{O}(\alpha_1^6)(E_{3-j,j}^{(t)} + (B_{j,1}^{(t)})^3) \right) \Lambda_{j,2}^{(t)} \qquad\qquad \text{(C.1)}$$

*Proof.* Again as in the proof of Lemma C.9, we expand the notations: (ignoring the superscript $(t)$ for the RHS)

$$\Lambda_{j,2}^{(t)} = C_0 \alpha_2^6 \Phi_j H_{j,1} B_{j,2}^5$$

$$\Gamma_{j,2}^{(t)} = C_0 \alpha_2^6 \Phi_j E_{3-j,j} B_{3-j,2}^3 B_{j,2}^2 H_{3-j,1}$$

$$\Upsilon_{j,2}^{(t)} = C_0 \alpha_1^6 \left( \Phi_j B_{j,1}^3 B_{j,2}^2 K_{j,2} + \Phi_{3-j} E_{3-j,j} B_{3-j,1}^3 B_{j,2}^2 K_{3-j,2} \right)$$

We first compare $\Lambda_{j,2}^{(t)}$ and $\Gamma_{j,2}^{(t)}$ as follows: Lemma C.6 we have

- $B_{3-j,2}^{(t)} \leq \widetilde{O}(B_{j,2}^{(t)})$ by Induction C.3b;

- From Lemma C.6a,b we can have $\Phi_{3-j}^{(t)} \leq \widetilde{O}(\alpha_1^{O(1)}) \Phi_j^{(t)}, \forall j \in [2]$.

Together they imply:

$$C_0 \alpha_2^6 E_{3-j,j}^{(t)} (B_{3-j,2}^{(t)})^3 (B_{j,2}^{(t)})^2 \Phi_{3-j}^{(t)} H_{3-j,1}^{(t)} \leq \widetilde{O}(\alpha_1^{O(1)} E_{3-j,j}^{(t)}) C_0 \alpha_2^6 \Phi_j^{(t)} H_{j,2}^{(t)} (B_{j,2}^{(t)})^5$$
$$= \widetilde{O}(\alpha_1^{O(1)} E_{j,3-j}^{(t)}) \Lambda_{j,2}^{(t)} \quad\quad (C.2)$$

Now we turn to compare $\Lambda_{j,2}^{(t)}$ with $\Upsilon_{j,2}^{(t)}$. We split $\Upsilon_{j,2}^{(t)}$ into two terms $\Upsilon_{j,2,1}^{(t)}, \Upsilon_{j,2,2}^{(t)}$

$$\Upsilon_{j,2,1}^{(t)} = C_0 \alpha_1^6 \Phi_j^{(t)} (B_{j,1}^{(t)})^3 (B_{j,2}^{(t)})^2 K_{j,2}^{(t)}, \quad \Upsilon_{j,2,2}^{(t)} = C_0 \alpha_1^6 \Phi_{3-j}^{(t)} E_{3-j,j}^{(t)} (B_{3-j,1}^{(t)})^3 (B_{j,2}^{(t)})^2 K_{3-j,2}^{(t)}$$

For $\Upsilon_{j,2,1}^{(t)}$, we can calculate

$$\Upsilon_{j,2,1}^{(t)} = C_0 \alpha_1^6 \Phi_j^{(t)} (B_{j,1}^{(t)})^3 (B_{j,2}^{(t)})^2 K_{j,2}^{(t)}$$
$$\leq \widetilde{O}(\frac{C_1 \alpha_2^6}{d^{3/2}})(B_{j,1}^{(t)})^3 \cdot C_0 \alpha_1^6 \Phi_j^{(t)} H_{j,1}^{(t)} (B_{j,2}^{(t)})^2 \quad\quad (K_{j,\ell}^{(t)} \leq \widetilde{O}(\frac{C_1 \alpha_\ell^6}{d^{3/2}}) \text{ from Lemma C.6d})$$
$$\leq \widetilde{O}(\alpha_1^6 (B_{j,1}^{(t)})^3) C_0 \alpha_2^6 \Phi_j^{(t)} H_{j,1}^{(t)} (B_{j,2}^{(t)})^5 \quad\quad (\widetilde{O}(\frac{C_1}{d^{3/2}}) \leq \widetilde{O}((B_{j,2}^{(t)})^3) \text{ from Induction C.3b})$$
$$= \widetilde{O}(\alpha_1^6 (B_{j,1}^{(t)})^3) \Lambda_{j,2}^{(t)} \quad\quad (C.3)$$

And for $\Upsilon_{j,2,1}^{(t)}$, we use Induction C.3b and Lemma C.6d again to get

$$(B_{3-j,1}^{(t)})^3 (B_{3-j,2}^{(t)})^2 K_{3-j,2}^{(t)} \leq \widetilde{O}(C_1 \alpha_2^6 (B_{j,2}^{(t)})^5)$$

and thus combined with $\Phi_{3-j}^{(t)} \leq \widetilde{O}(\alpha_1^6) \Phi_j^{(t)}, \forall j \in [2]$ from Lemma C.6a,b, we can derive

$$\Upsilon_{j,2,2}^{(t)} = C_0 \alpha_1^6 \Phi_{3-j}^{(t)} E_{j,3-j}^{(t)} (B_{3-j,1}^{(t)})^3 (B_{j,2}^{(t)})^2 K_{3-j,2}^{(t)}$$
$$\leq \widetilde{O}(\alpha_1^6 E_{3-j,j}^{(t)}) C_0 \alpha_2^6 \Phi_j^{(t)} H_{j,1}^{(t)} (B_{j,2}^{(t)})^5$$
$$= \widetilde{O}(\alpha_1^6 E_{3-j,j}^{(t)}) \Lambda_{j,2}^{(t)} \quad\quad (C.4)$$

Now combine the results of (C.2), (C.3) and (C.4) finishes the proof of (C.1). □

**Lemma C.11** (learning prediction head $E_{1,2}, E_{2,1}$ in phase I)**.** *If Induction C.3 holds at iteration $t \leq T_1$, then we have*

(a) $-\nabla_{E_{1,2}} L(W^{(t)}, E^{(t)}) = O(\Lambda_{1,1}^{(t)} B_{1,1}^{(t)}) \left( -O(E_{1,2}^{(t)}) + \widetilde{O}(\frac{(B_{1,2}^{(t)})^3}{(B_{1,1}^{(t)})^3}) + O(R_{1,2}^{(t)}) \right);$

(b) $-\nabla_{E_{2,1}} L(W^{(t)}, E^{(t)}) = \widetilde{O}(\frac{(B_{1,2}^{(t)})^3}{(B_{1,1}^{(t)})^2}) \Lambda_{1,1}^{(t)} + \sum_{\ell \in [2]} C_2 \Lambda_{2,\ell}^{(t)} B_{2,\ell}^{(t)} \left( -O(E_{2,1}^{(t)}) + O(R_{1,2}^{(t)}) \right)$

*Proof.* We first write down the gradient for $E_{j,3-j}^{(t)}$: (ignoring the time superscript $(t)$)

$$-\nabla_{E_{j,3-j}} L(W, E) = \sum_{\ell \in [2]} C_0 \Phi_j \alpha_\ell^6 B_{j,\ell}^3 (B_{3-j,\ell}^3 H_{j,3-\ell} - B_{3-j,3-\ell}^3 K_{j,3-\ell}) - \sum_{\ell \in [2]} \Sigma_{j,\ell} \nabla_{E_{j,3-j}} \mathcal{E}_{j,3-j}$$

where $\nabla_{E_{j,3-j}} \mathcal{E}_{j,3-j} = \mathbb{E}\left[2\langle w_j, \xi_p\rangle^3 \langle w_{3-j}, \xi_p\rangle^3 + 2E_{j,3-j}\langle w_{3-j}, \xi_p\rangle^6\right]$. Thus we have

$$\nabla_{E_{j,3-j}} \mathcal{E}_{j,3-j}^{(t)} = O(1)E_{j,3-j}^{(t)} + O(R_{1,2}^{(t)})$$

and by Claim C.5 and Lemma C.6a,b

$$\Sigma_{j,\ell}^{(t)} = O(\Lambda_{1,1}^{(t)} B_{1,1}^{(t)})\frac{(B_{j,\ell}^{(t)})^6 + E_{j,3-j}^{(t)}(B_{3-j,\ell}^{(t)})^3(B_{j,\ell}^{(t)})^3}{(B_{1,1}^{(t)})^6}\frac{\Phi_j^{(t)}}{\Phi_1^{(t)}} \le O(\Lambda_{1,1}^{(t)} B_{1,1}^{(t)})$$

Now let us look at $\nabla_{E_{1,2}} L(W^{(t)}, E^{(t)})$, first we consider the term

$$\sum_{\ell \in [2]} C_0 \Phi_1^{(t)} \alpha_\ell^6 (B_{1,\ell}^{(t)})^3 ((B_{2,\ell}^{(t)})^3 H_{1,3-\ell}^{(t)} - (B_{2,3-\ell}^{(t)})^3 K_{1,3-\ell}^{(t)})$$

Using Lemma C.6 and Induction C.3b,c, we know

- $H_{1,1}^{(t)} \le \widetilde{O}(H_{1,2}^{(t)})$ at $t \le T_{1,1}$ and $H_{1,1}^{(t)} \le \widetilde{O}(\alpha_1^6 H_{1,2}^{(t)})$ for $t \in [T_{1,1}, T_1]$;

- $B_{2,1}^{(t)}, B_{1,2}^{(t)}, B_{2,2}^{(t)} \le \widetilde{O}(B_{2,1}^{(t)}) \le \widetilde{O}(B_{1,1}^{(t)})$;

- $K_{1,3-\ell}^{(t)} \le \widetilde{O}(\alpha_1^6/d^{3/2})$.

It can be computed that

$$C_0\Phi_1^{(t)} \alpha_2^6 (B_{1,2}^{(t)})^3 (B_{2,2}^{(t)})^3 H_{1,1}^{(t)} \le \widetilde{O}(1)\left(\frac{B_{2,1}^{(t)}}{B_{1,1}^{(t)}}\right)^3 C_0\Phi_1^{(t)} \alpha_1^3 (B_{1,1}^{(t)})^6 H_{1,2}^{(t)}$$

$$\sum_{\ell \in [2]}\left|C_0\Phi_1^{(t)} \alpha_\ell^6 (B_{1,\ell}^{(t)})^3 (B_{2,\ell}^{(t)})^3 K_{1,3-\ell}^{(t)}\right| \le \widetilde{O}(\frac{\alpha_1^6}{d^{3/2}})\frac{(B_{2,1}^{(t)})^3}{(B_{1,1}^{(t)})^3} C_0\Phi_1^{(t)} \alpha_1^6 (B_{1,1}^{(t)})^6 H_{1,2}^{(t)}$$

Now we turn to $\nabla_{E_{2,1}} L(W^{(t)}, E^{(t)})$, similarly we have

$$C_0\Phi_2^{(t)} \alpha_1^6 (B_{2,1}^{(t)})^3 (B_{1,1}^{(t)})^3 H_{2,2}^{(t)} \le \widetilde{O}(1)\left(\frac{B_{2,1}^{(t)}}{B_{1,1}^{(t)}}\right)^3 C_0\Phi_1^{(t)} \alpha_1^6 (B_{1,1}^{(t)})^6 H_{1,2}^{(t)}$$

and since $H_{2,1}^{(t)} \le O(C_2) = O(H_{1,2}^{(t)})$ by Lemma C.6c, we can go through the same arguments again to obtain

$$\left|C_0\Phi_2^{(t)} \alpha_2^6 (B_{1,2}^{(t)})^3 (B_{2,2}^{(t)})^3 H_{2,1}^{(t)}\right| \le \widetilde{O}(1)\left(\frac{B_{1,2}^{(t)}}{B_{1,1}^{(t)}}\right)^3 C_0\Phi_1^{(t)} \alpha_1^6 (B_{1,1}^{(t)})^6 H_{1,2}^{(t)}$$

$$\left|C_0\Phi_2^{(t)} \alpha_2^6 (B_{1,2}^{(t)})^3 (B_{2,1}^{(t)})^3 K_{2,1}^{(t)}\right| \le \widetilde{O}(\frac{\alpha_1^6}{d^{3/2}})\left(\frac{B_{1,2}^{(t)}}{B_{1,1}^{(t)}}\right)^3 C_0\Phi_1^{(t)} \alpha_1^6 (B_{1,1}^{(t)})^6 H_{1,2}^{(t)}$$

Now the proof is complete. $\qquad\square$

Also, we will need the following lemma controlling gradient bounds for the noise term.

**Lemma C.12** (update of $R_{1,2}^{(t)}$ in phase I). *Suppose Induction C.3 holds at iteration $t \le T_1$, then we have*

*(a)* $|\langle -\nabla_{w_1} L(W^{(t)}, E^{(t)}), \Pi_{V^\perp} w_2^{(t)}\rangle| \le \widetilde{O}(\frac{1}{\sqrt{d}} + \varrho)\Lambda_{1,1}^{(t)} B_{1,1}^{(t)}$

*(b)* $|\langle -\nabla_{w_2} L(W^{(t)}, E^{(t)}), \Pi_{V^\perp} w_1^{(t)}\rangle| \le \widetilde{O}(\frac{1}{\sqrt{d}} + \varrho)\Lambda_{1,1}^{(t)} B_{1,1}^{(t)}$

*Proof.* **Proof of (a):** Firstly, by Claim C.7a, we can directly write

$$\langle \nabla_{w_1} L(W^{(t)}, E^{(t)}), \Pi_{V^\perp} w_2^{(t)} \rangle = -\sum_{j,\ell} \Sigma_{j,\ell}^{(t)} \langle \nabla_{w_1} \mathcal{E}_{j,3-j}^{(t)}, w_2^{(t)} \rangle$$

$$= -\Lambda_{1,1}^{(t)} B_{1,1}^{(t)} \sum_{(j,\ell)\in[2]^2} \frac{(B_{j,\ell}^{(t)})^6 + E_{j,3-j}^{(t)}(B_{3-j,\ell}^{(t)})^3(B_{j,\ell}^{(t)})^3}{(B_{1,1}^{(t)})^6} \frac{\Phi_j^{(t)}}{\Phi_1^{(t)}} \langle \nabla_{w_1} \mathcal{E}_{j,3-j}^{(t)}, w_1^{(t)} \rangle \tag{C.5}$$

Now we discuss each summand respectively: for $(j,\ell) = (1,1)$, we have

$$\frac{(B_{j,\ell}^{(t)})^6 + E_{j,3-j}^{(t)}(B_{3-j,\ell}^{(t)})^3(B_{j,\ell}^{(t)})^3}{(B_{1,1}^{(t)})^6} = 1 + E_{1,2}^{(t)}\frac{(B_{2,1}^{(t)})^3}{(B_{1,1}^{(t)})^3} = 1 + o(\frac{1}{d^{3/2}(B_{1,1}^{(t)})^3}) \tag{C.6}$$

where the last one is due to Induction C.3d. And for $\ell = 2$, we can see from Induction C.3b and d, that $\max_{(j,\ell)\neq(1,1)} |B_{j,\ell}^{(t)}| = \widetilde{O}(\frac{1}{\sqrt{d}})$ and $E_{j,3-j}^{(t)} \leq o(1)$ to give

$$\frac{(B_{j,2}^{(t)})^6 + E_{j,3-j}^{(t)}(B_{3-j,2}^{(t)})^3(B_{j,2}^{(t)})^3}{(B_{1,1}^{(t)})^6} \frac{\Phi_j^{(t)}}{\Phi_1^{(t)}} \leq \widetilde{O}(\frac{1}{d^3})\frac{1}{(B_{1,1}^{(t)})^6}\frac{\Phi_j^{(t)}}{\Phi_1^{(t)}}$$

On one hand, when $t \leq T_{1,1}$, we have $\alpha_\ell B_{j,\ell}^{(t)} \leq O(1)$ for all $(j,\ell) \in [2]^2$, so Lemma C.6a applies for both $\Phi_j^{(t)}$ and results in $\Phi_2^{(t)}/\Phi_1^{(t)} \leq O(1)$. We can also apply Induction C.3c to have $B_{j,2}^{(t)}/B_{1,1}^{(t)} \leq \widetilde{O}(1)$. On the other hand, when $t \in [T_{1,1}, T_1]$, we have by Induction C.3b and Lemma C.6a,b that $\Phi_2^{(t)}/\Phi_1^{(t)} \leq \widetilde{O}(\alpha_1^{O(1)}) = d^{o(1)}$, but now $B_{1,1}^{(t)} = d^{-o(1)} \gg \widetilde{O}(d^{-1/2})$, therefore

$$\widetilde{O}(\frac{1}{d^3})\frac{1}{(B_{1,1}^{(t)})^6}\frac{\Phi_2^{(t)}}{\Phi_1^{(t)}} \leq \widetilde{O}(\frac{1}{d^{3/2}})\frac{1}{(B_{1,1}^{(t)})^3}$$

So together, they imply

$$\frac{(B_{j,2}^{(t)})^6 + E_{j,3-j}^{(t)}(B_{3-j,2}^{(t)})^3(B_{j,2}^{(t)})^3}{(B_{1,1}^{(t)})^6} \frac{\Phi_j^{(t)}}{\Phi_1^{(t)}} \leq \widetilde{O}(\frac{1}{d^{3/2}(B_{1,1}^{(t)})^3}) \tag{C.7}$$

and similarly, we have

$$\frac{(B_{2,1}^{(t)})^6 + E_{2,1}^{(t)}(B_{1,1}^{(t)})^3(B_{2,1}^{(t)})^3}{(B_{1,1}^{(t)})^6} \frac{\Phi_2^{(t)}}{\Phi_1^{(t)}} \leq \widetilde{O}(\frac{1}{d^{3/2}(B_{1,1}^{(t)})^3}) \tag{C.8}$$

Next we turn to $\langle \nabla_{w_1} \mathcal{E}_{j,3-j}^{(t)}, w_2^{(t)} \rangle$. When $j = 1$, we can apply Claim C.7d to get

$$\langle \nabla_{w_1} \mathcal{E}_{1,2}^{(t)}, w_2^{(t)} \rangle = O(R_{1,2}^{(t)} + \varrho) + O(E_{1,2}^{(t)}) = O(\varrho + \frac{1}{\sqrt{d}}) + O(E_{1,2}^{(t)}) \leq O(\varrho + \frac{1}{\sqrt{d}}) \tag{C.9}$$

and when $j = 2$, we can apply Claim C.7e to get

$$\langle \nabla_{w_1} \mathcal{E}_{2,1}^{(t)}, w_2^{(t)} \rangle = -(E_{2,1}^{(t)})^2 O(R_{1,2}^{(t)} + \varrho) + O(E_{2,1}) = \widetilde{O}(\frac{1}{d^2})(\varrho + \frac{1}{\sqrt{d}}) + O(\frac{1}{d}) \tag{C.10}$$

Combining (C.5), (C.6), (C.7), (C.8), (C.9), and (C.10) completes the proof of (a).

**Proof of (b):** The $\Sigma_{j,\ell}^{(t)}$ part is the same as in the proof of (a), so we only deal with $\langle \nabla_{w_2} \mathcal{E}_{1,2}^{(t)}, w_1^{(t)} \rangle$ and $\langle \nabla_{w_2} \mathcal{E}_{2,1}^{(t)}, w_1^{(t)} \rangle$ here. For $\langle \nabla_{w_2} \mathcal{E}_{2,1}^{(t)}, w_1^{(t)} \rangle$, we apply Claim C.7d to get

$$\langle \nabla_{w_2} \mathcal{E}_{2,1}^{(t)}, w_1^{(t)} \rangle = O(R_{1,2}^{(t)} + \varrho) + O(1)E_{1,2}^{(t)} \tag{C.11}$$

and for $\langle \nabla_{w_2} \mathcal{E}_{1,2}^{(t)}, w_1^{(t)} \rangle$, we have

$$\langle \nabla_{w_2} \mathcal{E}_{1,2}^{(t)}, w_1^{(t)} \rangle = O(R_{1,2}^{(t)} + \varrho)(E_{2,1}^{(t)})^2 + O(1)E_{2,1}^{(t)} \tag{C.12}$$

Inserting (C.6), (C.7), (C.8) and (C.11), (C.12) into the expression of $\langle -\nabla_{w_2} L(W^{(t)}, E^{(t)}), \Pi_{V^\perp} w_1^{(t)} \rangle$ finishes the proof of (b). $\qquad\square$

## C.4 At the End of Phase I

**Lemma C.13** (Phase I). *Suppose $\eta \leq \frac{1}{\text{poly}(d)}$ is sufficiently small, then Induction C.3 holds for at least all $t \leq T_1 = O(\frac{d^2}{\eta})$, and at iteration $t = T_1$, we have*

(a) $B_{1,1}^{(T_1)} = \Omega(1)$;

(b) $\|w_j^{(T_1)}\|_2 = 1 \pm \widetilde{O}(\varrho + \frac{1}{\sqrt{d}})$;

(c) $B_{2,1}^{(T_1)} = \widetilde{\Theta}(\frac{1}{\sqrt{d}})$ *and* $B_{j,2}^{(T_1)} = B_{j,2}^{(0)}(1 \pm o(1))$ *for $j \in [2]$;*

(d) $E_{2,1}^{(T_1)} = \widetilde{O}(\frac{\eta_E/\eta}{d})$ *and* $E_{1,2}^{(T_1)} \leq \widetilde{O}(\varrho + \frac{1}{\sqrt{d}})$;

(e) $R_{1,1}^{(T_1)}, R_2^{(T_1)} = \Theta(1)$ *and* $R_{1,2}^{(T_1)} = \widetilde{O}(\varrho + \frac{1}{\sqrt{d}})$.

*Proof.* We begin by first prove the existence of $T_1 := \min\{t : B_{1,1}^{(t)} \geq 0.01\} = O(\frac{d^2}{\eta})$ if Induction C.3 holds whenever $B_{1,1}^{(t)} \leq 0.01$, then we will turn back to prove Induction C.3 holds throughout $t \leq T_1$. We split the analysis into two stages:

**Proof of $T_1 \leq O(\frac{d^2}{\eta})$:** By Lemma C.9a we can write down the update of $B_{1,1}^{(t)}$ as

$$B_{1,1}^{(t+1)} = B_{1,1}^{(t)} + \eta(1 \pm \widetilde{O}(\frac{1}{d}))\Lambda_{1,1}^{(t)} = B_{1,1}^{(t)} + \eta(1 \pm \widetilde{O}(\frac{1}{d}))\Phi_1^{(t)} C_0 \alpha_1^6 H_{1,2}^{(t)}(B_{1,1}^{(t)})^5 \qquad \text{(C.13)}$$

When $\alpha_1 B_{1,1}^{(t)} \leq O(1)$, by Lemma C.6a,c we have $\Phi_1^{(t)} = \Theta(\frac{1}{C_2^2})$ and $H_{1,2}^{(t)} = \Omega(C_2)$, this means we can lower bound the update as

$$B_{1,1}^{(t+1)} \geq B_{1,1}^{(t)} + \Omega(\frac{\eta C_0 \alpha_1^6}{C_2})(B_{1,1}^{(t)})^5$$

since $\frac{C_0 \alpha_1^6}{C_2}$ is a constant, we know there exist some $t' \geq 0$ such that $B_{1,1}^{(t')} \geq \Omega(\frac{1}{\alpha_1})$. Also recall that $T_{1,1} := \min\{t : B_{1,1}^{(t)} \geq \Omega(\frac{1}{\alpha_1})\}$. So by Lemma H.1, where $\eta = \frac{1}{\text{poly}(d)}, C_t = \Omega(\frac{C_0 \alpha_1^6}{C_2})$ $\delta = \frac{1}{\text{polylog}(d)}$ and $A = \Omega(\frac{1}{\alpha_1}), \log(A/B_{1,1}^{(0)}) = \widetilde{O}(1)$, we have

$$T_{1,1} = O(\frac{C_2}{\eta C_0 \alpha_1^6}) \sum_{x_t \leq O(\frac{1}{\alpha_1})} \eta C_t \leq O(\frac{C_2}{\eta C_0 \alpha_1^6})\left(O(1) + \frac{\widetilde{O}(\eta)}{B_{1,1}^{(0)}}\right)\frac{1}{(B_{1,1}^{(0)})^4} \leq \widetilde{O}(\frac{1}{\eta \alpha_1^6 (B_{1,1}^{(0)})^4})$$

Since $(B_{1,1}^{(0)})^4 \geq \widetilde{\Omega}(\frac{1}{d^2})$ from our initialization, we have $T_{1,1} \leq O(\frac{d^2}{\eta})$ and thus $T_{1,1}$ exists. Now we consider when $B_{1,1}^{(t)} \geq \Omega(\frac{1}{\alpha_1})$. Now by Lemma C.6b,c, we have $\Phi_1^{(t)} \geq \Omega((C_2 + \alpha_1^6)^{-2})$, which gives an update:

$$B_{1,1}^{(t+1)} \geq B_{1,1}^{(t)} + \Omega(\frac{\eta C_0 \alpha_1^6}{(C_2 + \alpha_1^6)^2})(B_{1,1}^{(t)})^5$$

so again by Lemma H.1, choosing $C_t = \Omega(\frac{C_0 \alpha_1^6}{(C_2 + \alpha_1^6)^2})$,

$$T_1 = \frac{O((C_2 + \alpha_1^6)^2)}{\eta C_0 \alpha_1^6} \sum_{x_t \in [\Omega(\frac{1}{\alpha_1}), 0.01]} \eta C_t \leq \left(O(1) + \frac{\widetilde{O}(\eta)}{B_{1,1}^{(T_{1,1})}}\right)\frac{\widetilde{O}(\alpha_1^{12})}{(B_{1,1}^{(T_{1,1})})^4} \leq \widetilde{O}(\frac{\alpha_1^6}{\eta (B_{1,1}^{(T_{1,1})})^4}) \leq O(\frac{\alpha_1^6}{\eta})$$

where $O(\frac{\alpha_1^6}{\eta}) \ll O(\frac{d^2}{\eta})$, so we have proved that $T_1$ exist. Now we begin to prove that Induction C.3 holds for all $t \leq T_1$.

**Proof of Induction C.3:** We first prove (b)–(d), and then come back to prove (a) and (d). At $t = 0$, we know all induction holds from Properties C.1. Now we suppose Induction C.3 holds for

all iterations $\leq t - 1$ and prove it holds at $t$.

**The growth of $B_{2,1}^{(t)}$:** Applying Lemma C.9, we have for $t \leq T_{1,1}$

$$B_{1,1}^{(t+1)} \geq B_{1,1}^{(t)} + \eta(1 - \widetilde{O}(\frac{1}{d}))\Lambda_{1,1}^{(t)}$$

$$B_{2,1}^{(t+1)} \leq B_{2,1}^{(t)} + \eta(1 + O(\frac{1}{\sqrt{d}}))\Lambda_{2,1}^{(t)} + \eta\frac{(B_{2,1}^{(t)})^2}{(B_{1,1}^{(t)})^2}E_{1,2}^{(t)}\Lambda_{1,1}^{(t)}$$

For some $t_1' := \min\{t : B_{1,1}^{(t)} \geq \frac{\Omega(1)}{d^{0.49}}\}$, we have $E_{1,2}^{(t)} \leq \widetilde{O}(B_{1,1}^{(t)}\varrho) \lesssim \frac{1}{d^{0.49}}$ during $t \leq t_1'$, and

$$\frac{(B_{2,1}^{(t)})^2}{(B_{1,1}^{(t)})^2}E_{1,2}^{(t)}\Lambda_{1,1}^{(t)} \lesssim \frac{(B_{2,1}^{(t)})^2}{d^{0.49}(B_{1,1}^{(t)})^2}\Lambda_{1,1}^{(t)} \leq \widetilde{O}(\frac{1}{d^{0.49}})\Lambda_{2,1}^{(t)}$$

which allow us to give an upper bound to $B_{2,1}^{(t+1)}$ as

$$B_{2,1}^{(t+1)} \leq (1 + O(\frac{1}{\sqrt{d}}))\Lambda_{2,1}^{(t)} + \widetilde{O}(\frac{1}{d^{0.49}})\Lambda_{2,1}^{(t)}$$

$$\leq (1 + \widetilde{O}(\frac{1}{d^{0.49}}))\Phi_2^{(t)}C_0\alpha_1^6 C_2\mathcal{E}_2^{(t)}(1 + \frac{1}{\mathsf{polylog}(d)})(B_{2,1}^{(t)})^5 \qquad \text{(when } t \leq t_1')$$

Since we also have

$$B_{1,1}^{(t+1)} \geq (1 - \widetilde{O}(\frac{1}{d}))\Lambda_{1,1}^{(t)} \geq (1 - \widetilde{O}(\frac{1}{d}))\Phi_1^{(t)}C_0\alpha_1^6\mathcal{E}_1^{(t)}(1 - \frac{1}{\mathsf{polylog}(d)})(B_{1,1}^{(t)})^5$$

Since $B_{1,1}^{(0)} \geq B_{2,1}^{(0)}(1 + \Omega(\frac{1}{\log d}))$, we can now apply Corollary H.2 to the two sequence $B_{1,1}^{(t+1)}$ and $B_{2,1}^{(t+1)}$, where $S_t = \frac{\Phi_1^{(t)}\mathcal{E}_1^{(t)}}{\Phi_2^{(t)}\mathcal{E}_2^{(t)}}(1 + \frac{1}{\mathsf{polylog}(d)})$ to get

$$B_{1,1}^{(t_1')} \geq \frac{1}{d^{0.499}} \quad \text{while} \quad B_{2,1}^{(t_1')} \leq \widetilde{O}(\frac{1}{\sqrt{d}})$$

Note that here the update of $B_{2,1}^{(t)}$ at every step satisfies $\mathrm{sign}(B_{2,1}^{(t+1)} - B_{2,1}^{(t)}) = \mathrm{sign}(B_{2,1}^{(t)})$ which implies $B_{2,1}^{(t_1')} = \widetilde{\Theta}(\frac{1}{\sqrt{d}})$. Now for every $T \in [t_1', T_1]$, we can apply Lemma H.3 to get that

$$\sum_{t \in [t_1', T]} \eta\frac{(B_{2,1}^{(t)})^2}{(B_{1,1}^{(t)})^2}E_{1,2}^{(t)}\Lambda_{1,1}^{(t)} \leq \widetilde{O}(\varrho + \frac{1}{\sqrt{d}})O(\frac{1}{B_{1,1}^{(t_1')}})\max_{t \leq T}\{(B_{2,1}^{(t)})^2\} \leq O(\frac{1}{d^{0.5+\Omega(1)}})$$

Suppose we have proved that $B_{2,1}^{(t)} \leq \widetilde{O}(\frac{1}{\sqrt{d}})$ for each $t \leq T$, we define a new sequence

$$\widetilde{B}_{2,1}^{(t+1)} = \widetilde{B}_{2,1}^{(t)} + \eta(1 + \widetilde{O}(\frac{1}{d^{0.49}}))\Phi_2^{(t)}C_0\alpha_1^6 C_2\mathcal{E}_2^{(t)}(1 + \frac{1}{\mathsf{polylog}(d)})(\widetilde{B}_{2,1}^{(t)})^5,$$

$$\text{where } \widetilde{B}_{2,1}^{(t_1')} = B_{2,1}^{(t_1')} + \sum_{t \in [t_1', T]} \eta\frac{(B_{2,1}^{(t)})^2}{(B_{1,1}^{(t)})^2}E_{1,2}^{(t)}\Lambda_{1,1}^{(t)} = (1 \pm o(1))\widetilde{B}_{2,1}^{(t_1')}$$

It can be directly seen that $|\widetilde{B}_{2,1}^{(t)} - \widetilde{B}_{2,1}^{(0)}| \geq |B_{2,1}^{(t)} - B_{2,1}^{(0)}|$ for all $t \in [t_1', T]$. Notice that now $\widetilde{B}_{2,1}^{(t_1')} \leq d^{\Omega(1)}B_{1,1}^{(t_1')}$, we can now apply Corollary H.2 again to get

$$|B_{2,1}^{(T)} - B_{2,1}^{(0)}| \leq |\widetilde{B}_{2,1}^{(T)} - \widetilde{B}_{2,1}^{(0)}| \leq \frac{1}{\sqrt{d}\mathsf{polylog}(d)} \qquad \text{(for every } T \leq T_{1,1})$$

Now we deal with $t \in [T_{1,1}, T_1]$. During this stage, we can directly apply Corollary H.2 to $\widetilde{B}_{2,1}^{(t)}$ and $B_{1,1}^{(t)}$, where $S_t = \frac{\Phi_1^{(t)}H_{1,2}^{(t)}}{\Phi_2^{(t)}H_{2,2}^{(t)}} \leq O(\alpha_1^{O(1)})$, to get that

$$|B_{2,1}^{(T)} - B_{2,1}^{(0)}| \leq |\widetilde{B}_{2,1}^{(T)} - \widetilde{B}_{2,1}^{(0)}| \leq \frac{1}{\sqrt{d}\mathsf{polylog}(d)} \qquad \text{(for every } T \leq T_1)$$

And thus by Lemma C.1, we have $B_{2,1}^{(T)} = B_{2,1}^{(0)}(1 \pm o(1))$.

**The growth of $B_{1,2}^{(t)}$ and $B_{2,2}^{(t)}$:** By Lemma C.10, we can write down the update as

$$B_{j,2}^{(t+1)} = B_{j,2}^{(t)} + \eta \left( 1 \pm \widetilde{O}(\alpha_1^6)(E_{3-j,j}^{(t)} + (B_{j,1}^{(t)})^3) \right) \Lambda_{j,2}^{(t)}$$

Since $B_{2,1}^{(t)} \le \widetilde{O}(\frac{1}{\sqrt{d}})$ and $E_{1,2}^{(t)} \le \widetilde{O}(\varrho + \frac{1}{\sqrt{d}})B_{1,1}^{(t)}$, $E_{2,1}^{(t)} \le \widetilde{O}(\frac{1}{d})$ because we chose $\eta_E \le \eta$, we only need to care about $(B_{1,1}^{(t)})^3$ in the update expression. Now define $t_2' := \min\{t : B_{1,1}^{(t)} \ge \Omega(\frac{1}{\alpha_1^2})\}$, we have

- For $t \le t_2'$, by Corollary H.2 and setting $x_t = B_{1,1}^{(t)}$, $C_t = (1 - \widetilde{O}(\frac{1}{d}))\Phi_1^{(t)}C_0\alpha_1^6 H_{1,2}^{(t)}$, $S_t = O(\frac{\alpha_2^6\Phi_j^{(t)}H_{j,1}^{(t)}}{\alpha_1^6\Phi_1^{(t)}H_{1,2}^{(t)}}) \le \widetilde{O}(\frac{\alpha_2^6}{\alpha_1^6}) \ll \frac{1}{\text{polylog}(d)}$ (by Lemma C.6a,c), we have $|B_{j,2}^{(t)} - B_{j,2}^{(0)}| \le O(\frac{\alpha_2^6}{\alpha_1^6}\frac{1}{\sqrt{d}}) \lesssim \frac{1}{\sqrt{d}\text{polylog}(d)}$ for all $t \le t_2'$, which implies $B_{j,2}^{(t_2')} = B_{j,2}^{(0)} \pm \frac{1}{\sqrt{d}\text{polylog}(d)} \in [\Omega(\frac{1}{\sqrt{d}\log d}), O(\frac{\sqrt{\log d}}{\sqrt{d}})]$ by Lemma C.1.

- For $t \in [t_2', T_1]$, we can use Corollary H.2 again and let $x_t = B_{1,1}^{(t)}$, we know $B_{1,1}^{(t_2')} \ge d^{\Omega(1)}B_{2,1}^{(t_2')}$. Setting $C_t = (1 - \widetilde{O}(\frac{1}{d}))\Phi_1^{(t)}C_0\alpha_1^6 H_{1,2}^{(t)}$, $S_t = O((1 + \alpha_1^6)\frac{\alpha_2^6\Phi_j^{(t)}H_{j,1}^{(t)}}{\alpha_1^6\Phi_1^{(t)}H_{1,2}^{(t)}}) \le O(\alpha^{O(1)})$, we can have $|B_{j,2}^{(t)} - B_{j,2}^{(t_2')}| \lesssim \frac{1}{\sqrt{d}\text{polylog}(d)}$, which implies $B_{j,2}^{(t)} \in [\Omega(\frac{1}{\sqrt{d}\log d}), O(\frac{\sqrt{\log d}}{\sqrt{d}})]$ for all $t \in [t_2', T_1]$.

This proves Induction C.3b. Indeed, simple calculations also proves Induction C.3c, since the update of $B_{1,1}^{(t)}$ is always larger than others' during $t \le T_1$.

**For Induction C.3d:** From Lemma C.11, we can write down the update

$$-\nabla_{E_{1,2}}L(W^{(t)}, E^{(t)}) = O(\Lambda_{1,1}^{(t)}B_{1,1}^{(t)}) \left( -C_1 E_{1,2}^{(t)} + \widetilde{O}(\frac{(B_{1,2}^{(t)})^3}{(B_{1,1}^{(t)})^3}) + C_2(R_{1,2}^{(t)} + \varrho) \right)$$

for some constants $C_1, C_2 = \Theta(1)$. Applying Lemma H.3 to $O(\Lambda_{1,1}^{(t)}B_{1,1}^{(t)})\frac{(B_{1,2}^{(t)})^3}{(B_{1,1}^{(t)})^3}$, we can obtain

$$\sum_{t \le T} O(\eta_E \Lambda_{1,1}^{(t)}B_{1,1}^{(t)})\frac{(B_{1,2}^{(t)})^3}{(B_{1,1}^{(t)})^3} = \frac{\eta_E}{\eta} \sum_{t \le T} O(\eta \Lambda_{1,1}^{(t)})\frac{(B_{1,2}^{(t)})^3}{(B_{1,1}^{(t)})^2} \le \widetilde{O}(\frac{\eta_E/\eta}{d^{3/2}})\frac{1}{B_{1,1}^{(0)}} \le \widetilde{O}(\frac{\eta_E/\eta}{d})$$

So here it suffices to notice that whenever $|E_{1,2}^{(t)}| < 2\frac{C_2}{C_1}(R_{1,2}^{(t)} + \varrho)$ (which is obviously satisified at $t = 0$), we would have

$$O(\Lambda_{1,1}^{(t)}B_{1,1}^{(t)}) \left( -O(E_{1,2}^{(t)}) + C_2(R_{1,2}^{(t)} + \varrho) \right) = -O(\Lambda_{1,1}^{(t)}B_{1,1}^{(t)})\widetilde{O}(R_{1,2}^{(t)} + \varrho) \le O(\Lambda_{1,1}^{(t)}B_{1,1}^{(t)})\widetilde{O}(\varrho + \frac{1}{\sqrt{d}})$$

In that case, we will always have (since $E_{1,2}^{(0)} = 0$)

$$E_{1,2}^{(t+1)} \le \left| \sum_{t \le T} \widetilde{O}(\eta_E \Lambda_{1,1}^{(t)}B_{1,1}^{(t)})\frac{(B_{1,2}^{(t)})^3}{(B_{1,1}^{(t)})^3} \right| + \sum_{s \le t} O(\eta_E \Lambda_{1,1}^{(s)}B_{1,1}^{(s)})(R_{1,2}^{(s)} + \varrho) \le \widetilde{O}(\varrho + \frac{1}{\sqrt{d}})\frac{\eta_E}{\eta}B_{1,1}^{(t+1)}$$

Similarly for $\nabla_{E_{2,1}}L(W^{(t)}, E^{(t)})$, we can write down

$$-\nabla_{E_{2,1}}L(W^{(t)}, E^{(t)}) = \widetilde{O}(\frac{(B_{1,2}^{(t)})^3}{(B_{1,1}^{(t)})^2})\Lambda_{1,1}^{(t)} + \sum_{\ell \in [2]} C_2 \Lambda_{2,\ell}^{(t)}B_{2,\ell}^{(t)} \left( -O(E_{2,1}^{(t)}R_2^{(t)}) + O(R_{1,2}^{(t)}) \right)$$

by Lemma H.3, we have

$$\sum_{t \leq T_1} \eta_E \widetilde{O}\left(\frac{(B_{1,2}^{(t)})^3}{(B_{1,1}^{(t)})^2}\right)\Lambda_{1,1}^{(t)} \leq \widetilde{O}\left(\frac{\eta_E/\eta}{d}\right)$$

and since from previous comparison results we know that

$$\sum_{t \leq T_1} \sum_{\ell \in [2]} \eta_E C_2 \Lambda_{2,\ell}^{(t)} B_{2,\ell}^{(t)} = \frac{\eta_E}{\eta} \sum_{t \leq T_1} \sum_{\ell \in [2]} \eta C_2 \Lambda_{2,\ell}^{(t)} B_{2,\ell}^{(t)} \leq \widetilde{O}\left(\frac{\eta_E/\eta}{d}\right)$$

we can then prove the claim.

**For Induction C.3a:** We can write down the update of $\|w_j^{(t)}\|_2^2$ as follows:

$$\|w_j^{(t+1)}\|_2^2 = \|w_j^{(t)} - \eta \nabla_{w_j} L(W^{(t)}, E^{(t)})\|_2^2$$
$$= \|w_j^{(t)}\|_2^2 - \eta \langle \nabla_{w_j} L(W^{(t)}, E^{(t)}), w_j^{(t)} \rangle + \eta^2 \|\nabla_{w_j} L(W^{(t)}, E^{(t)})\|_2^2$$

from (B.2) and Induction C.3a,b,c at iteration $t$ and our assumption on $\xi_p$, we know

$$\|\nabla_{w_j} L(W^{(t)}, E^{(t)})\|_2^2 \leq \widetilde{O}(d)$$

which allow us to choose $\eta \leq \frac{1}{\mathsf{poly}(d)}$ to be small enough so that $\eta d T_1 \leq \frac{1}{\eta \mathsf{poly}(d)}$. Then by Lemma C.8b, we have

$$\|w_j^{(t+1)}\|_2^2 = \|w_j^{(0)}\|_2^2 \pm \eta \sum_{s \leq t} |\langle \nabla_{w_j} L(W^{(s)}, E^{(s)}), w_j^{(s)} \rangle| \pm \frac{1}{\mathsf{poly}(d)}$$
$$\leq \|w_j^{(0)}\|_2^2 \pm \eta \sum_{s \leq t} \widetilde{O}\left(\varrho + \frac{1}{\sqrt{d}}\right)|\Lambda_{1,1}^{(s)}| \sum_{j \in [2]} |E_{j,3-j}^{(s)}| \pm \frac{1}{\mathsf{poly}(d)}$$

Since from the above analysis of the update of $B_{1,1}^{(t)}$, we know $\sum_{t \leq T_1} \Lambda_{1,1}^{(t)} \leq O(1)$. Moreover, we also know that $|B_{1,1}^{(t)}|$ is increasing and $\mathrm{sign}(\Lambda_{1,1}^{(t)}) = \mathrm{sign}(\Lambda_{1,1}^{(s)})$ for any $s, t \leq T_1$. Thus they imply $\sum_{s \leq t} |\Lambda_{1,1}^{(s)}| = |\sum_{s \leq t} \Lambda_{1,1}^{(s)}| = O(1)$, which can be combine with Induction C.3d to prove the claim.

**Proof of Induction C.3e:** We can write down the update of $R_{1,2}^{(t)} = \langle \Pi_{V^\perp} w_1^{(t)}, w_2^{(t)} \rangle$ as follows

$$\langle \Pi_{V^\perp} w_1^{(t+1)}, w_2^{(t+1)} \rangle = \langle \Pi_{V^\perp} w_1^{(t)} - \Pi_{V^\perp} \eta \nabla_{w_1} L(W^{(t)}, E^{(t)}), \Pi_{V^\perp} w_2^{(t)} - \Pi_{V^\perp} \eta \nabla_{w_2} L(W^{(t)}, E^{(t)}) \rangle$$
$$= R_{1,2}^{(t)} - \eta \langle \nabla_{w_1} L(W^{(t)}, E^{(t)}), \Pi_{V^\perp} w_2^{(t)} \rangle - \eta \langle \nabla_{w_2} L(W^{(t)}, E^{(t)}), \Pi_{V^\perp} w_1^{(t)} \rangle$$
$$+ \eta^2 \langle \Pi_{V^\perp} \nabla_{w_1} L(W^{(t)}, E^{(t)}), \Pi_{V^\perp} \nabla_{w_2} L(W^{(t)}, E^{(t)}) \rangle$$

By Cauchy-Schwarz inequality and the same analysis above we have

$$|\langle \Pi_{V^\perp} \nabla_{w_1} L(W^{(t)}, E^{(t)}), \Pi_{V^\perp} \nabla_{w_2} L(W^{(t)}, E^{(t)}) \rangle| \leq \|\nabla_{w_1} L(W^{(t)}, E^{(t)})\|_2 \|\nabla_{w_2} L(W^{(t)}, E^{(t)})\|_2$$
$$\leq \widetilde{O}(d)$$

so by our choice of $\eta$

$$\sum_{t \leq T_1} \eta^2 |\langle \Pi_{V^\perp} \nabla_{w_1} L(W^{(t)}, E^{(t)}), \Pi_{V^\perp} \nabla_{w_2} L(W^{(t)}, E^{(t)}) \rangle| \leq \frac{1}{\mathsf{poly}(d)}$$

and by Lemma C.12 we have

$$\left| -\eta \langle \nabla_{w_1} L(W^{(t)}, E^{(t)}), \Pi_{V^\perp} w_2^{(t)} \rangle - \eta \langle \nabla_{w_2} L(W^{(t)}, E^{(t)}), \Pi_{V^\perp} w_1^{(t)} \rangle \right| \leq \eta \widetilde{O}(\Lambda_{1,1}^{(t)} B_{1,1}^{(t)})\left(\varrho + \frac{1}{\sqrt{d}}\right)$$

which implies

$$|\langle \Pi_{V^\perp} w_1^{(t+1)}, w_2^{(t+1)}\rangle| \le |\langle \Pi_{V^\perp} w_1^{(0)}, w_2^{(0)}\rangle| + \sum_{s\le t}\sum_{j\in[2]} \eta|\langle \nabla_{w_j} L(W^{(s)}, E^{(s)}), \Pi_{V^\perp} w_{3-j}^{(s)}\rangle| + \frac{1}{\mathsf{poly}(d)}$$

$$\le \widetilde{O}(\frac{1}{\sqrt{d}}) + \sum_{s\le t}\eta\widetilde{O}(\Lambda_{1,1}^{(s)}B_{1,1}^{(s)}) + \frac{1}{\mathsf{poly}(d)}$$

$$\le \widetilde{O}(\frac{1}{\sqrt{d}}) + \widetilde{O}(\varrho + \frac{1}{\sqrt{d}})B_{1,1}^{(t+1)}$$

$$\le \widetilde{O}(\varrho + \frac{1}{\sqrt{d}})$$

which completes the proof of Induction C.3. As for (a) – (e) of Lemma C.13, they are just direct corrolary of our induction at $t = T_1$. $\qquad\square$

## D  Phase II: The Substitution Effect of Prediction Head

In this phase, As $B_{1,1}^{(t)}$ is learned to become very large ($B_{1,1}^{(t)} \gtrsim \|w_1^{(t)}\|_2$). The focus now shift to grow $E_{2,1}^{(t)}$, because we want $C_1\alpha_1^6((B_{2,1}^{(t)})^3 + E_{2,1}^{(t)}(B_{1,1}^{(t)})^3)^2$ in $H_{2,1}^{(t)}$ to dominate $\mathcal{E}_{2,1}^{(t)}$. We can write down the gradient of $E_{2,1}^{(t)}$ as

$$-\nabla_{E_{2,1}} L(W^{(t)}, E^{(t)}) = \sum_{\ell\in[2]} C_0\Phi_2^{(t)}\alpha_\ell^6(B_{2,\ell}^{(t)})^3((B_{1,\ell}^{(t)})^3 H_{2,3-\ell}^{(t)} - (B_{2,3-\ell}^{(t)})^3 K_{2,3-\ell}^{(t)}) - \sum_{\ell\in[2]} \Sigma_{2,\ell}^{(t)}\nabla_{E_{2,1}}\mathcal{E}_{2,1}^{(t)}$$

Now let us define

$$T_2 := \min\{t : R_2^{(t)} < \frac{1}{\log d}|E_{1,2}^{(t)}|\} \tag{D.1}$$

We will prove that $E_{2,1}^{(T_2)}$ reaches at most $O(\sqrt{\eta_E/\eta})$ and the following induction hypothesis holds throughout $t \in [T_1, T_2]$. In this phase, the learning of $E_{2,1}^{(t)}$ is much faster than the growth of the first feature $v_1$ such that $T_2 - T_1 = o(T_1/\sqrt{d})$, which is due to the acceleration effects brought by $B_{1,1}^{(t)} = \Omega(1)$ during this phase.

### D.1  Induction in Phase II

We will be based on the following induction hypothesis during phase II.

**Inductions D.1** (Phase II). *When $t \in [T_1, T_2]$, we hypothesize the followings would hold*

(a) $B_{1,1}^{(t)} = \Theta(1)$, $B_{j,\ell}^{(t)} = B_{j,\ell}^{(T_1)}(1 \pm o(1)) = \widetilde{\Theta}(\frac{1}{\sqrt{d}})$ *for* $(j,\ell) \ne (1,1)$ *and* $\mathrm{sign}(B_{j,\ell}^{(t)}) = \mathrm{sign}(B_{j,\ell}^{(T_1)})$;

(b) $|R_{1,2}^{(t)}| = \widetilde{O}(\varrho + \frac{1}{\sqrt{d}})\alpha_1^{O(1)}[R_1^{(t)}]^{1/2}[R_2^{(t)}]^{1/2}$;

(c) $R_1^{(t)} \in [\Omega(\frac{1}{d^{3/4}\alpha_1^2}), O(1)]$, $R_2^{(t)} \in [\Omega(\frac{1}{\log d}\sqrt{\eta_E/\eta}), O(1)]$;

(d) $E_{1,2}^{(t)} \le \widetilde{O}(\varrho + \frac{1}{\sqrt{d}})[R_1^{(t)}]^{3/2}$ *and* $E_{2,1}^{(t)} \le O(\sqrt{\eta_E/\eta})$.

Under Induction D.1, we have some results as direct corollary.

**Claim D.2.** *At each iteration $t \in [T_1, T_2]$, if Induction C.3 holds, then*

(a) $\mathcal{E}_j^{(t)} = \Theta(C_2[R_j^{(t)}]^3)$;

(b) $\mathcal{E}_{j,3-j}^{(t)} = \mathcal{E}_j^{(t)} \pm \widetilde{O}(E_{j,3-j}^{(t)}(\varrho + \frac{1}{\sqrt{d}})[R_1^{(t)}]^{3/2}[R_2^{(t)}]^{3/2}) + O((E_{j,3-j}^{(t)})^2[R_{3-j}^{(t)}]^3)$ *for each $j \in [2]$;*

*Proof.* It is trivial to derive (a) from the expression of $\mathcal{E}_j^{(t)}$ and our assumption of $\xi_p$. For (b) it suffices to directly calculate the expression of $\mathcal{E}_{j,3-j}^{(t)}$ along with Induction D.1b. $\qquad\square$

**Lemma D.3** (variables control in phase II). *In Phase II ($t \in [T_1, T_2]$), if Induction D.1 holds, then*

*(a)* $\Phi_1^{(t)} = \widetilde{\Theta}(\frac{1}{\alpha_1^{12}})$, $\Phi_2^{(t)} = \Theta((C_2[R_2^{(t)}]^3 + C_1\alpha_1^6(E_{2,1}^{(t)})^2)^{-2})$;

*(b)* $K_{1,\ell}^{(t)} = \widetilde{O}(\alpha_\ell^6/d^{3/2})$, $K_{2,\ell}^{(t)} = \widetilde{O}(E_{2,1}^{(t)}\alpha_\ell^6/d^{3/2} + \alpha_\ell^6/d^3)$

*(c)* $H_{1,1}^{(t)} = \Theta(C_1\alpha_1^6)$, $H_{1,2}^{(t)} = \widetilde{O}([R_1^{(t)}]^3)$, $H_{2,2}^{(t)} = \Theta(C_2[R_2^{(t)}]^3)$, $H_{2,1}^{(t)} = \Theta(C_2[R_2^{(t)}]^3 + C_1\alpha_1^6(E_{2,1}^{(t)})^2)$.

*Proof.* The proof of (a) directly follows from Induction D.1a,c and Claim D.2. The proof of (b) follows directly from the expression of $K_{j,\ell}$ and Induction D.1a,d. The proof of (c) is also similar. $\square$

## D.2 Gradient Lemmas for Phase II

**Lemma D.4** (learning prediction head $E_{1,2}, E_{2,1}$ in phase II). *If Induction D.1 holds at iteration $t \in [T_1, T_2]$, then we have*

*(a)* $-\nabla_{E_{1,2}}L(W^{(t)}, E^{(t)}) = (1 \pm \widetilde{O}(\frac{\alpha_1^{O(1)}}{d^{3/2}}))\Sigma_{1,1}^{(t)}(-2E_{1,2}^{(t)}[R_2^{(t)}]^3 \pm O(\overline{R}_{1,2}^{(t)} + \varrho)[R_1^{(t)}]^{3/2}[R_2^{(t)}]^{3/2})$

$$\pm \Sigma_{1,1}^{(t)}\widetilde{O}(\frac{\eta_E/\eta}{\sqrt{d}})\max\{[R_1^{(t)}]^3, \frac{\alpha_1^{O(1)}}{d^{5/2}}\},$$

*(b)* $-\nabla_{E_{2,1}}L(W^{(t)}, E^{(t)}) = (1 \pm \widetilde{O}(\frac{\alpha_1^{O(1)}}{d^{3/2}}))C_0\Phi_2^{(t)}\alpha_1^6(B_{2,1}^{(t)})^3(B_{1,1}^{(t)})^3H_{2,2}^{(t)}$

$$\pm O(\Sigma_{2,1}^{(t)})(|E_{2,1}^{(t)}|[R_1^{(t)}]^3 \pm O(\overline{R}_{1,2}^{(t)} + \varrho)[R_1^{(t)}]^{3/2}[R_2^{(t)}]^{3/2})$$

*Proof.* We first write down the gradient for $E_{j,3-j}^{(t)}$: (ignoring the time superscript $^{(t)}$)

$$-\nabla_{E_{j,3-j}}L(W, E) = \sum_{\ell \in [2]} C_0\Phi_j\alpha_\ell^6B_{j,\ell}^3(B_{3-j,\ell}^3H_{j,3-\ell} - B_{3-j,3-\ell}^3K_{j,3-\ell}) - \sum_{\ell \in [2]}\Sigma_{j,\ell}\nabla_{E_{j,3-j}}\mathcal{E}_{j,3-j}$$

where $\nabla_{E_{j,3-j}}\mathcal{E}_{j,3-j} = \mathbb{E}\left[2\langle w_j, \xi_p\rangle^3\langle w_{3-j}, \xi_p\rangle^3 + 2E_{j,3-j}\langle w_{3-j}, \xi_p\rangle^6\right]$. Thus we have

$$\nabla_{E_{j,3-j}}\mathcal{E}_{j,3-j}^{(t)} = 2E_{j,3-j}^{(t)}[R_{3-j}^{(t)}]^3 \pm O(\overline{R}_{1,2}^{(t)} + \varrho)[R_1^{(t)}]^{3/2}[R_2^{(t)}]^{3/2}$$

and by Claim B.1 and Induction D.1a, if $(j, \ell) \neq (1, 1)$

$$\Sigma_{j,\ell}^{(t)} = O(\Sigma_{1,1}^{(t)})\frac{(B_{j,\ell}^{(t)})^6 + E_{j,3-j}^{(t)}(B_{3-j,\ell}^{(t)})^3(B_{j,\ell}^{(t)})^3}{(B_{1,1}^{(t)})^6}\frac{\Phi_j^{(t)}}{\Phi_1^{(t)}} \leq o(\frac{1}{d^{3/2}})\Sigma_{1,1}^{(t)}\frac{\Phi_j^{(t)}}{\Phi_1^{(t)}}$$

Therefore for $j = 1$:

$$\sum_{\ell \in [2]}\Sigma_{1,\ell}^{(t)}\nabla_{E_{1,2}}\mathcal{E}_{1,2}^{(t)} = (1 \pm \widetilde{O}(\frac{\alpha_1^{O(1)}}{d^{3/2}}))\Sigma_{1,1}^{(t)}\nabla_{E_{1,2}}\mathcal{E}_{1,2}^{(t)}$$

Now by Induction D.1a,c and Lemma D.3b,c we have $(B_{1,\ell}^{(t)})^3H_{1,3-\ell}^{(t)} \leq \max\{\Theta(C_2[R_1^{(t)}]^3), \widetilde{O}(\frac{\alpha_1^6}{d^{3/2}})\}$, which leads to the bounds

$$|(B_{1,\ell}^{(t)})^3(B_{2,\ell}^{(t)})^3H_{1,3-\ell}^{(t)}| \leq \widetilde{O}(\frac{1}{d^{3/2}})\max\{[R_1^{(t)}]^3, \frac{\alpha_1^6}{d^3}\}, \qquad |(B_{1,\ell}^{(t)})^3(B_{2,3-\ell}^{(t)})^3K_{1,3-\ell}^{(t)}| \leq \widetilde{O}(\frac{1}{d^3})$$

which implies

$$\left|\sum_{\ell \in [2]}C_0\Phi_1^{(t)}\alpha_\ell^6(B_{1,\ell}^{(t)})^3((B_{2,\ell}^{(t)})^3H_{1,3-\ell}^{(t)} - (B_{2,3-\ell}^{(t)})^3K_{1,3-\ell}^{(t)})\right| \lesssim \widetilde{O}(\frac{\eta_E/\eta}{\sqrt{d}})\Sigma_{1,1}^{(t)}\max\{[R_1^{(t)}]^3, \frac{\alpha_1^{O(1)}}{d^{5/2}}\}$$

Combining above together, we have

$$-\nabla_{E_{1,2}}L(W^{(t)}, E^{(t)})$$

$$= (1 + o(\frac{1}{d^{3/2}}))\Sigma_{1,1}^{(t)}(-2E_{1,2}^{(t)}[R_2^{(t)}]^3 \pm O(\overline{R}_{1,2}^{(t)} + \varrho)[R_1^{(t)}]^{3/2}[R_2^{(t)}]^{3/2} \pm \widetilde{O}(\frac{\eta_E/\eta}{\sqrt{d}})\max\{[R_1^{(t)}]^3, \frac{\alpha_1^{O(1)}}{d^{5/2}}\})$$

For $-\nabla_{E_{2,1}}L(W^{(t)}, E^{(t)})$, the expression is slightly different, we first observe that by Induction D.1a

$$\Delta_{2,2}^{(t)} \leq \widetilde{O}(\frac{1}{d^{3/2}})\Delta_{2,1}^{(t)}$$

Meanwhile, by Induction D.1a and Lemma D.3b,c , we have

$$\Xi_2^{(t)} \leq \widetilde{O}(\frac{\alpha_1^{O(1)}}{d^3})C_0C_2\Phi_2^{(t)}[R_2^{(t)}]^3,$$

Moreover, we can also calculate $\Sigma_{2,1}^{(t)} = C_0C_2\alpha_1^6 E_{2,1}^{(t)}\Phi_2^{(t)}(B_{2,1}^{(t)})^3) = \widetilde{O}(\frac{\alpha_1^6}{d^{3/2}})\Phi_2^{(t)}$, $\Sigma_{2,2}^{(t)} = \widetilde{O}(\frac{\alpha_2^6}{d^3})\Phi_2^{(t)}$, which gives

$$\sum_{\ell \in [2]} \Sigma_{2,\ell}^{(t)}\nabla_{E_{2,1}}\mathcal{E}_{2,1}^{(t)} = \Sigma_{2,1}^{(t)}(-\Theta(E_{2,1}^{(t)})[R_1^{(t)}]^3 \pm O(\overline{R}_{1,2}^{(t)} + \varrho)[R_1^{(t)}]^{3/2}[R_2^{(t)}]^{3/2})$$

Now we combine the above results and get

$$-\nabla_{E_{2,1}}L(W^{(t)}, E^{(t)}) = (1 \pm \widetilde{O}(\frac{\alpha_1^{O(1)}}{d^{3/2}}))C_0\Phi_2^{(t)}\alpha_1^6(B_{2,1}^{(t)})^3(B_{1,1}^{(t)})^3\mathcal{E}_{2,1}^{(t)}$$

$$\pm O(\Sigma_{2,1}^{(t)})(|E_{2,1}^{(t)}|[R_1^{(t)}]^3 \pm O(\overline{R}_{1,2}^{(t)} + \varrho)[R_1^{(t)}]^{3/2}[R_2^{(t)}]^{3/2})$$

$$\square$$

**Lemma D.5** (reducing noise in phase II). *Suppose Induction D.1 holds at $t \in [T_1, T_2]$, then*

*(a)* $\langle-\nabla_{w_1}L(W^{(t)}, E^{(t)}), \Pi_{V^\perp}w_1^{(t)}\rangle = \Sigma_{1,1}^{(t)}\Theta(-[R_1^{(t)}]^3 \pm \widetilde{O}(|E_{1,2}^{(t)}| + \frac{|E_{2,1}^{(t)}|^2}{d^{3/2}})(\overline{R}_{1,2}^{(t)} + \varrho)[R_1^{(t)}]^{3/2}[R_2^{(t)}]^{3/2})$;

*(b)* $\langle-\nabla_{w_1}L(W^{(t)}, E^{(t)}), \Pi_{V^\perp}w_2^{(t)}\rangle = \Sigma_{1,1}^{(t)}((-\Theta(\overline{R}_{1,2}^{(t)}) + O(\varrho))[R_1^{(t)}]^{5/2}[R_2^{(t)}]^{1/2} + \widetilde{O}(|E_{1,2}^{(t)}| + \frac{|E_{2,1}^{(t)}|^2}{d^{3/2}})R_1^{(t)}[R_2^{(t)}]^2)$

*And furthermore*

*(c)* $\langle-\nabla_{w_2}L(W^{(t)}, E^{(t)}), \Pi_{V^\perp}w_2^{(t)}\rangle = -\Theta([R_2^{(t)}]^3)\left(\Sigma_{1,1}^{(t)}\Theta((E_{1,2}^{(t)})^2) + \sum_{\ell \in [2]}\Sigma_{2,\ell}^{(t)}\right)$

$$\pm O\left(\sum_{j,\ell}\Sigma_{j,\ell}^{(t)}E_{j,3-j}^{(t)}(\overline{R}_{1,2}^{(t)} + \varrho)[R_1^{(t)}]^{3/2}[R_2^{(t)}]^{3/2}\right);$$

*(d)* $\langle-\nabla_{w_2}L(W^{(t)}, E^{(t)}), \Pi_{V^\perp}w_1^{(t)}\rangle = \left(\Sigma_{1,1}^{(t)}\Theta((E_{1,2}^{(t)})^2) + \sum_{\ell \in [2]}\Sigma_{2,\ell}^{(t)}\right)(-\Theta(\overline{R}_{1,2}^{(t)}) \pm O(\varrho))[R_2^{(t)}]^{5/2}[R_1^{(t)}]^{1/2}$

$$+ O\left(\sum_{j,\ell}\Sigma_{j,\ell}^{(t)}E_{j,3-j}^{(t)}R_2^{(t)}[R_1^{(t)}]^2\right)$$

*Proof.* The proof can be obtained directly from some calculation using Claim B.1 as follows:
**Proof of (a):** From (B.2), we can obtain that

$$\langle-\nabla_{w_1}L(W^{(t)}, E^{(t)}), \Pi_{V^\perp}w_1^{(t)}\rangle = -\sum_{j,\ell}\Sigma_{j,\ell}^{(t)}\langle\nabla_{w_1}\mathcal{E}_{j,3-j}^{(t)}, w_1^{(t)}\rangle$$

Now from Claim B.1a and Induction D.1a, we know $(B_{j,\ell}^{(t)})^3 \leq \widetilde{O}(\frac{1}{d^{3/2}})$ and the following

$$\Sigma_{j,\ell}^{(t)} = O(\Sigma_{1,1}^{(t)})\frac{(B_{j,\ell}^{(t)})^6 + E_{j,3-j}^{(t)}(B_{3-j,\ell}^{(t)})^3(B_{j,\ell}^{(t)})^3}{(B_{1,1}^{(t)})^6}\frac{\Phi_j^{(t)}}{\Phi_1^{(t)}} \leq \widetilde{O}(\frac{E_{j,3-j}^{(t)}}{d^{3/2}})\Sigma_{1,1}^{(t)}\frac{\Phi_j^{(t)}}{\Phi_1^{(t)}}$$
$$\text{for any } (j,\ell) \neq (1,1)$$

From Induction D.1a,c, we know $((B_{2,\ell}^{(t)})^3 + E_{2,1}^{(t)}(B_{1,\ell}^{(t)})^3)^2 \leq \widetilde{O}(\frac{1}{d^{3/2}})E_{2,1}^{(t)}$ and $R_2^{(t)} = \Theta(1)$, which by Claim D.2a,b and Lemma D.3a gives $\Phi_2^{(t)}/\Phi_1^{(t)} \leq \widetilde{O}(\alpha_1^{O(1)})$. Combine the bounds above, we can obtain $\Sigma_{j,\ell}^{(t)} = \widetilde{O}(E_{j,3-j}^{(t)}/d^{3/2})\Sigma_{1,1}^{(t)}$. We can then directly apply Claim B.1 to prove Lemma D.5a as follows

$$\langle -\nabla_{w_1}L(W^{(t)}, E^{(t)}), \Pi_{V^\perp}w_1^{(t)}\rangle$$
$$= (1 \pm \widetilde{O}(E_{1,2}^{(t)}))\Sigma_{1,1}^{(t)}\Big(-\Theta([R_1^{(t)}]^3) \pm O(E_{1,2}^{(t)})(\overline{R}_{1,2}^{(t)} + \varrho)[R_1^{(t)}]^{3/2}[R_2^{(t)}]^{3/2}\Big)$$
$$+ \widetilde{O}(E_{2,1}^{(t)}/d^{3/2})\Sigma_{1,1}^{(t)}\Big(-\Theta((E_{2,1}^{(t)})^2)[R_1^{(t)}]^3 \pm O(E_{2,1}^{(t)})(\overline{R}_{1,2}^{(t)} + \varrho)[R_1^{(t)}]^{3/2}[R_2^{(t)}]^{3/2}\Big)$$
$$= \Theta(\Sigma_{1,1}^{(t)})\Big(-[R_1^{(t)}]^3 \pm \widetilde{O}(|E_{1,2}^{(t)}| + \frac{|E_{2,1}^{(t)}|^2}{d^{3/2}})(\overline{R}_{1,2}^{(t)} + \varrho)[R_1^{(t)}]^{3/2}[R_2^{(t)}]^{3/2}\Big)$$
$$\text{(Since } |E_{1,2}^{(t)}| \leq d^{-\Omega(1)} \text{ by Induction D.1c,d)}$$

**Proof of (b):** For Lemma D.5b, we can use the same analysis for $\Sigma_{1,1}^{(t)}$ above and Claim B.1(d,e) to get (again we have used $\Sigma_{j,\ell}^{(t)} = \widetilde{O}(E_{j,3-j}^{(t)})\Sigma_{1,1}^{(t)} = o(\Sigma_{1,1}^{(t)})$)

$$\langle -\nabla_{w_1}L(W^{(t)}, E^{(t)}), \Pi_{V^\perp}w_2^{(t)}\rangle$$
$$= (1 \pm \widetilde{O}(E_{1,2}^{(t)}))\Sigma_{1,1}^{(t)}\Big((-\Theta(\overline{R}_{1,2}^{(t)}) \pm O(\varrho))[R_1^{(t)}]^{5/2}[R_2^{(t)}]^{1/2} + E_{1,2}^{(t)}R_1^{(t)}[R_2^{(t)}]^2\Big)$$
$$+ \widetilde{O}(E_{2,1}^{(t)}/d^{3/2})\Sigma_{1,1}^{(t)}\Big((-\Theta(\overline{R}_{1,2}^{(t)}) + O(\varrho))(E_{2,1}^{(t)})^2[R_1^{(t)}]^{5/2}[R_2^{(t)}]^{1/2} + E_{2,1}^{(t)}R_1^{(t)}[R_2^{(t)}]^2\Big)$$
$$= \Sigma_{1,1}^{(t)}((-\Theta(\overline{R}_{1,2}^{(t)}) + O(\varrho))[R_1^{(t)}]^{5/2}[R_2^{(t)}]^{1/2} + \widetilde{O}(|E_{1,2}^{(t)}| + \frac{|E_{2,1}^{(t)}|^2}{d^{3/2}})R_1^{(t)}[R_2^{(t)}]^2)$$

**Proof of (c):** Similarly to the proof of (a), we can also expand as follows

$$\langle -\nabla_{w_2}L(W^{(t)}, E^{(t)}), \Pi_{V^\perp}w_2^{(t)}\rangle$$
$$= (1 \pm O(E_{1,2}^{(t)}))\Sigma_{1,1}^{(t)}\Big(-[R_2^{(t)}]^3\Theta((E_{1,2}^{(t)})^2) \pm O(E_{1,2}^{(t)})(\overline{R}_{1,2}^{(t)} + \varrho)[R_1^{(t)}]^{3/2}[R_2^{(t)}]^{3/2}\Big)$$
$$- \sum_{\ell\in[2]}\Sigma_{2,\ell}^{(t)}\Big([R_2^{(t)}]^3 \pm O(E_{2,1}^{(t)})(\overline{R}_{1,2}^{(t)} + \varrho)[R_1^{(t)}]^{3/2}[R_2^{(t)}]^{3/2}\Big)$$
$$= -[R_2^{(t)}]^3\Big(\Sigma_{1,1}^{(t)}\Theta((E_{1,2}^{(t)})^2) + \sum_{\ell\in[2]}\Sigma_{2,\ell}^{(t)}\Big) \pm O\Big(\sum_{j,\ell}\Sigma_{j,\ell}^{(t)}E_{j,3-j}^{(t)}(\overline{R}_{1,2}^{(t)} + \varrho)[R_1^{(t)}]^{3/2}[R_2^{(t)}]^{3/2}\Big)$$

**Proof of (d):** Similarly, we can calculate (again by $\Sigma_{j,\ell}^{(t)} = \widetilde{O}(E_{j,3-j}^{(t)})\Sigma_{1,1}^{(t)} = o(\Sigma_{1,1}^{(t)})$)

$$\langle -\nabla_{w_2}L(W^{(t)}, E^{(t)}), \Pi_{V^\perp}w_1^{(t)}\rangle$$
$$= \sum_{\ell\in[2]}\Sigma_{1,\ell}^{(t)}\Big((-\Theta(\overline{R}_{1,2}^{(t)}) \pm O(\varrho))(E_{1,2}^{(t)})^2[R_2^{(t)}]^{5/2}[R_1^{(t)}]^{1/2} + E_{1,2}^{(t)}R_2^{(t)}[R_1^{(t)}]^2\Big)$$
$$+ \sum_{\ell\in[2]}\Sigma_{2,\ell}^{(t)}\Big((-\Theta(\overline{R}_{1,2}^{(t)}) \pm O(\varrho))[R_2^{(t)}]^{5/2}[R_1^{(t)}]^{1/2} + E_{2,1}^{(t)}R_2^{(t)}[R_1^{(t)}]^2\Big)$$
$$= (1 \pm \widetilde{O}(E_{1,2}^{(t)}))\Sigma_{1,1}^{(t)}\Big((-\Theta(\overline{R}_{1,2}^{(t)}) \pm O(\varrho))(E_{1,2}^{(t)})^2[R_2^{(t)}]^{5/2}[R_1^{(t)}]^{1/2} + E_{1,2}^{(t)}R_2^{(t)}[R_1^{(t)}]^2\Big)$$
$$+ \sum_{\ell\in[2]}\Sigma_{2,\ell}^{(t)}\Big((-\Theta(\overline{R}_{1,2}^{(t)}) \pm O(\varrho))[R_2^{(t)}]^{5/2}[R_1^{(t)}]^{1/2} + E_{1,2}^{(t)}R_2^{(t)}[R_1^{(t)}]^2\Big)$$
$$= \Big(\Sigma_{1,1}^{(t)}\Theta((E_{1,2}^{(t)})^2) + \sum_{\ell\in[2]}\Sigma_{2,\ell}^{(t)}\Big)(-\Theta(\overline{R}_{1,2}^{(t)}) \pm O(\varrho))[R_2^{(t)}]^{5/2}[R_1^{(t)}]^{1/2} + O\Big(\sum_{j,\ell}\Sigma_{j,\ell}^{(t)}E_{j,3-j}^{(t)}R_2^{(t)}[R_1^{(t)}]^2\Big)$$

which completes the proof. $\qquad\square$

**Lemma D.6** (learning feature $v_2$ in phase II). *For each $t \in [T_1, T_2]$, if Induction D.1 holds at iteration $t$, then we have for each $j \in [2]$:*

$$|\langle -\nabla_{w_j} L(W^{(t)}, E^{(t)}), v_2 \rangle| \leq \widetilde{O}\left(\frac{\alpha_2^6 \alpha_1^6}{d^{5/2}}\right)\left(\Phi_j^{(t)}(|E_{j,3-j}^{(t)}| + [R_j^{(t)}]^3) + \Phi_{3-j}^{(t)}(|E_{3-j,j}^{(t)}|[R_{3-j}^{(t)}]^3 + \frac{|E_{3-j,j}^{(t)}|^2}{d^{3/2}})\right)$$

*Proof.* Again as in the proof of Lemma C.9, we expand the notations: (ignoring the superscript $^{(t)}$ for the RHS)

$$\langle -\nabla_{w_j} L(W^{(t)}, E^{(t)}), v_2 \rangle = \Lambda_{j,2}^{(t)} + \Gamma_{j,2}^{(t)} - \Upsilon_{j,2}^{(t)} \tag{D.2}$$

where

$$\Lambda_{j,2}^{(t)} = C_0 \alpha_2^6 \Phi_j^{(t)} H_{j,1}^{(t)} (B_{j,2}^{(t)})^5$$
$$\Gamma_{j,2}^{(t)} = C_0 \alpha_2^6 \Phi_{3-j}^{(t)} E_{3-j,j}^{(t)} (B_{3-j,2}^{(t)})^3 (B_{j,2}^{(t)})^2 H_{3-j,1}^{(t)}$$
$$\Upsilon_{j,2}^{(t)} = C_0 \alpha_1^6 \left( \Phi_j^{(t)} (B_{j,1}^{(t)})^3 (B_{j,2}^{(t)})^2 K_{j,2}^{(t)} + \Phi_{3-j}^{(t)} E_{3-j,j}^{(t)} (B_{3-j,1}^{(t)})^3 (B_{j,2}^{(t)})^2 K_{3-j,2}^{(t)} \right)$$

Now we further write $\Upsilon_{j,2}^{(t)} = \Upsilon_{j,2,1}^{(t)} + \Upsilon_{j,2,2}^{(t)}$, where

$$\Upsilon_{j,2,1}^{(t)} = C_0 \alpha_1^6 \Phi_j^{(t)} (B_{j,1}^{(t)})^3 (B_{j,2}^{(t)})^2 K_{j,2}^{(t)}, \qquad \Upsilon_{j,2,2}^{(t)} = \Phi_{3-j}^{(t)} E_{3-j,j}^{(t)} (B_{3-j,1}^{(t)})^3 (B_{3-j,2}^{(t)})^2 K_{3-j,2}^{(t)}$$

According to (D.2), we can first compute

$$\Lambda_{j,2}^{(t)} - \Upsilon_{j,2,1}^{(t)} = C_0 \alpha_2^6 \Phi_j^{(t)} (B_{j,2}^{(t)})^5 H_{j,1}^{(t)} - C_0 \alpha_1^6 \Phi_j^{(t)} (B_{j,1}^{(t)})^3 (B_{j,2}^{(t)})^2 K_{j,2}^{(t)}$$

$$= C_0 \alpha_2^6 \Phi_j^{(t)} (B_{j,2}^{(t)})^5 \left( C_1 \alpha_1^6 ((B_{j,1}^{(t)})^3 + E_{j,3-j}^{(t)} (B_{3-j,1}^{(t)})^3)^2 + C_2 \mathcal{E}_{j,3-j}^{(t)} \right)$$
$$\quad - C_0 \alpha_1^6 \Phi_j^{(t)} (B_{j,1}^{(t)})^3 (B_{j,2}^{(t)})^2 C_1 \alpha_2^6 ((B_{j,2}^{(t)})^3 + E_{j,3-j}^{(t)} (B_{3-j,2}^{(t)})^3)((B_{j,1}^{(t)})^3 + E_{j,3-j}^{(t)} (B_{3-j,1}^{(t)})^3)$$

$$= C_0 \alpha_2^6 C_1 \alpha_1^6 \Phi_j^{(t)} (B_{j,2}^{(t)})^5 \left( E_{j,3-j}^{(t)} (B_{3-j,1}^{(t)})^3 (B_{j,1}^{(t)})^3 + (E_{j,3-j}^{(t)})^2 (B_{3-j,1}^{(t)})^6 \right)$$
$$\quad - C_0 \alpha_2^6 C_1 \alpha_1^6 \Phi_j^{(t)} (B_{j,2}^{(t)})^2 (B_{3-j,2}^{(t)})^3 E_{j,3-j}^{(t)} \left( (B_{j,1}^{(t)})^6 + E_{j,3-j}^{(t)} (B_{3-j,1}^{(t)})^3 (B_{j,1}^{(t)})^3 \right)$$
$$\quad + C_0 \alpha_2^6 \Phi_j^{(t)} (B_{j,2}^{(t)})^5 C_2 \mathcal{E}_{j,3-j}^{(t)}$$

Then we can apply Induction D.1a,c,d, Claim D.2a,b and Lemma D.3a,c to get

$$|\Lambda_{j,2}^{(t)} - \Gamma_{j,2,1}^{(t)}| \leq \widetilde{O}\left(\frac{\alpha_2^6}{\alpha_1^6 d^{5/2}}\right) \Phi_j^{(t)} (|E_{j,3-j}^{(t)}| + [R_j^{(t)}]^3)$$

where the last inequality is due to Lemma D.3a,c. Similarly, we can also compute for $\Gamma_{j,2}^{(t)} - \Upsilon_{j,2,2}^{(t)}$:

$$|\Gamma_{j,2}^{(t)} - \Upsilon_{j,2,2}^{(t)}| \leq \left| C_0 \alpha_2^6 \Phi_{3-j}^{(t)} E_{3-j,j}^{(t)} (B_{3-j,2}^{(t)})^3 (B_{j,2}^{(t)})^2 H_{3-j,1}^{(t)} \right|$$
$$\quad + \left| C_0 \alpha_1^6 \Phi_{3-j}^{(t)} E_{3-j,j}^{(t)} (B_{3-j,1}^{(t)})^3 (B_{j,2}^{(t)})^2 K_{3-j,2}^{(t)} \right|$$
$$\leq \widetilde{O}\left(\frac{\alpha_1^6 \alpha_2^6}{d^{5/2}}\right) \Phi_{3-j}^{(t)} |E_{3-j,j}^{(t)}|([R_{3-j}^{(t)}]^3 + \frac{|E_{3-j,j}^{(t)}|}{d^{3/2}})$$

This completes the proof $\qquad\square$

**Lemma D.7** (learning feature $v_1$ in Phase II). *For each $t \in [T_1, T_2]$, if Induction D.1 holds at iteration $t$, then we have:*

(a) $\langle -\nabla_{w_1} L(W^{(t)}, E^{(t)}), v_1 \rangle = \Theta(\Sigma_{1,1}^{(t)})[R_1^{(t)}]^3 + \Gamma_{1,1}^{(t)} \pm \widetilde{O}(\alpha_1^{O(1)}/d^{5/2})$;

(b) $\langle -\nabla_{w_2} L(W^{(t)}, E^{(t)}), v_1 \rangle = \widetilde{O}(\alpha_1^{O(1)}/d^{5/2}) + \widetilde{O}(\frac{\alpha_1^6}{d}) E_{1,2}^{(t)} \Phi_1^{(t)} [R_1^{(t)}]^3$

*Proof.* As in the proof of Lemma D.6, we expand the gradient terms:

$$\langle -\nabla_{w_j} L(W^{(t)}, E^{(t)}), v_1 \rangle = \Lambda_{j,2}^{(t)} + \Gamma_{j,2}^{(t)} - \Upsilon_{j,2}^{(t)} \tag{D.3}$$

where

$$\Lambda_{j,1}^{(t)} = C_0 \alpha_1^6 \Phi_j^{(t)} H_{j,2}^{(t)} (B_{j,1}^{(t)})^5$$

$$\Gamma_{j,1}^{(t)} = C_0 \alpha_1^6 \Phi_{3-j}^{(t)} E_{3-j,j}^{(t)} (B_{3-j,1}^{(t)})^3 (B_{j,1}^{(t)})^2 H_{3-j,2}^{(t)}$$

$$\Upsilon_{j,1}^{(t)} = C_0 \alpha_1^6 \left( \Phi_j^{(t)} (B_{j,2}^{(t)})^3 (B_{j,1}^{(t)})^2 K_{j,1}^{(t)} + \Phi_{3-j}^{(t)} E_{3-j,j}^{(t)} (B_{3-j,2}^{(t)})^3 (B_{j,1}^{(t)})^2 K_{3-j,1}^{(t)} \right)$$

Indeed, when $j = 1$, by Induction D.1a and Lemma D.3a,c, we can compute

$$\Lambda_{1,1}^{(t)} = C_0 \alpha_1^6 \Phi_1^{(t)} (B_{1,1}^{(t)})^5 H_{1,2}^{(t)} = \Theta(\Sigma_{1,1}^{(t)})[R_1^{(t)}]^3$$

and with additionally Lemma D.3b, we also have

$$|\Upsilon_{1,1}^{(t)}| = \left| C_0 \alpha_1^6 \left( \Phi_1^{(t)} (B_{1,2}^{(t)})^3 (B_{1,1}^{(t)})^2 K_{1,1}^{(t)} + \Phi_2^{(t)} E_{2,j}^{(t)} (B_{2,2}^{(t)})^3 (B_{1,1}^{(t)})^2 K_{2,1}^{(t)} \right) \right| \leq \widetilde{O}(\frac{\alpha_1^{O(1)}}{d^{5/2}})$$

which gives the proof of (a). For (b), we can also apply Induction D.1a and Lemma D.3a,c to get

$$\Lambda_{2,1}^{(t)} = C_0 \alpha_1^6 \Phi_2^{(t)} H_{2,2}^{(t)} (B_{2,1}^{(t)})^5 \leq \widetilde{O}(\alpha_1^{O(1)}/d^{5/2})$$

$$\Gamma_{2,1}^{(t)} = C_0 \alpha_1^6 \Phi_1^{(t)} E_{1,2}^{(t)} (B_{1,1}^{(t)})^3 (B_{2,1}^{(t)})^2 H_{1,2}^{(t)} \leq \widetilde{O}(\frac{1}{d}) E_{1,2}^{(t)} \Phi_1^{(t)} \frac{[R_1^{(t)}]^3}{\alpha_1^6}$$

$$\Upsilon_{2,1}^{(t)} = C_0 \alpha_1^6 \left( \Phi_2^{(t)} (B_{2,2}^{(t)})^3 (B_{2,1}^{(t)})^2 K_{2,1}^{(t)} + \Phi_1^{(t)} E_{1,2}^{(t)} (B_{1,2}^{(t)})^3 (B_{2,1}^{(t)})^2 K_{1,1}^{(t)} \right) \leq \widetilde{O}(\frac{\alpha_1^6}{d^4})$$

this finishes the proof. $\qquad\square$

## D.3 At the End of Phase II

Now we shall present the main theorem of this section, which gives the result of prediction head $E_{2,1}^{(t)}$ growth after the feature $v_1$ is learned in the first stage.

**Lemma D.8** (Phase II). *Suppose $\eta = \frac{1}{\mathrm{poly}(d)}$ is sufficiently small, then Induction D.1 holds for all iteration $t \in [T_1, T_2]$, and at iteration $t = T_2$, the followings holds:*

*(a)* $B_{1,1}^{(T_2)} = \Theta(1)$, $B_{j,\ell}^{(T_2)} = B_{j,\ell}^{(T_1)}(1 \pm o(1)) = \widetilde{\Theta}(\frac{1}{\sqrt{d}})$ for $(j, \ell) \neq (1, 1)$

*(b)* $R_1^{(T_2)} \leq \widetilde{O}(\frac{1}{d^{3/4}})$, $R_2^{(T_2)} = \Theta(\sqrt{\eta_E/\eta})$, and $\overline{R}_{1,2}^{(T_2)} \leq \widetilde{O}(\varrho + \frac{1}{\sqrt{d}})$;

*(c)* $|E_{1,2}^{(T_2)}| = \widetilde{O}(\varrho + \frac{1}{\sqrt{d}})[R_1^{(t)}]^{3/2}[R_2^{(t)}]^{3/2}$ and $|E_{2,1}^{(T_2)}| = \Theta(\sqrt{\eta_E/\eta})$

*Where the part of learning $E_{2,1}^{(t)}$ is what we called substitution effect. One can easily verify that $|E_{2,1}^{(t)} f_1(X^{(1)})| \gg |f_2(X^{(1)})|$ when $X$ is equipped with feature $v_1$, as stated in Lemma 5.2.*

*Proof.* We first will prove Induction D.1 holds for all iteration $t \in [T_1, T_2]$. We shall first prove that if Induction D.1 continues to hold when $R_2^{(t)} \geq |E_{2,1}^{(t)}|$, we shall have $[R_1^{(t)}]$ decreasing at an exponential rate.

**Proof of the decrease of $R_1^{(t)}$:** Firstly, we write down the update of $R_1^{(t)}$ using Lemma D.5a:

$$R_1^{(t+1)} = R_1^{(t)} + \eta \Sigma_{1,1}^{(t)} \Theta(-[R_1^{(t)}]^3 \pm O(|E_{1,2}^{(t)}| + \frac{|E_{2,1}^{(t)}|^2}{d^{3/2}})(\overline{R}_{1,2}^{(t)} + \varrho)[R_1^{(t)}]^{3/2}[R_2^{(t)}]^{3/2})$$

from the expression of $\Sigma_{1,1}^{(t)}$ in (B.2), and by Induction D.1a and Lemma D.3a,c, we can compute

$$\Sigma_{1,1}^{(t)} = \Theta(C_0 C_2 \Phi_1^{(t)}) = \Theta(\frac{C_0 C_2}{\alpha_1^{12}})$$

Moreover, from Induction D.1c we know that

$$(|E_{1,2}^{(t)}| + \frac{|E_{2,1}^{(t)}|^2}{d^{3/2}})[R_1^{(t)}]^{3/2}[R_2^{(t)}]^{3/2} \le (\widetilde{\Theta}(\frac{1}{d^{3/2}}) + \widetilde{O}(\varrho + \frac{1}{\sqrt{d}})[R_1^{(t)}]^{3/2})[R_1^{(t)}]^{3/2}[R_2^{(t)}]^{3/2}$$

$$\le (\widetilde{\Theta}(\frac{1}{d^{3/2}}) + \widetilde{O}(\varrho + \frac{1}{\sqrt{d}})[R_1^{(t)}]^{3/2})[R_1^{(t)}]^{3/2}$$

Therefore whenever $R_1^{(t)} \ge \frac{\alpha_1^{18}}{d^{3/4}}$ (which $t \le T_2$ suffices), we shall have always have

$$(\overline{R}_{1,2}^{(t)} + \varrho)(\widetilde{\Theta}(\frac{1}{d^{3/2}}) + \widetilde{O}(\varrho + \frac{1}{\sqrt{d}})[R_1^{(t)}]^{3/2})[R_1^{(t)}]^{3/2} \le o([R_1^{(t)}]^3)$$

which implies, if we set $T_2' := \min\{t : R_1^{(t)} \ge \frac{1}{d^{3/4}\alpha_1^2}\}$, then for all $t \in [T_1, T_2']$, we will have

$$R_1^{(t+1)} = R_1^{(t)} + \eta\Sigma_{1,1}^{(t)}\Theta(-[R_1^{(t)}]^3 \pm O(|E_{1,2}^{(t)}| + \frac{|E_{2,1}^{(t)}|^2}{d^{3/2}})(\overline{R}_{1,2}^{(t)} + \varrho)[R_1^{(t)}]^{3/2}[R_2^{(t)}]^{3/2})$$

$$= R_1^{(t)} - \Theta(\eta\Sigma_{1,1}^{(t)})[R_1^{(t)}]^3 \qquad\qquad\qquad \text{(D.4)}$$

$$\le R_1^{(t)}(1 - \Theta(\frac{\eta C_0 C_2}{\alpha_1^{12}})\frac{1}{d^{3/2}\alpha_1^2}) \qquad\qquad\qquad (\text{since } R_1^{(t)} \ge \frac{1}{d^{3/4}})$$

From the last inequality we know that after $T_2 = T_1 + \widetilde{\Theta}(\frac{d^{1.5}}{\eta\alpha_1^{\Omega(1)}})$, we shall have $R_1^{(t)} \le O(\frac{\alpha_1^{O(1)}}{d^{3/4}})$. Moreover, suppose $T_2' < T_2$, (which just mean $R_1^{(s)} \le O(\frac{1}{d^{3/4}\alpha_1^2})$ for some iteration $s \in [T_1, T_2]$) we also have

$$R_1^{(t+1)} = R_1^{(t)} - \Theta(\eta\Sigma_{1,1}^{(t)})[R_1^{(t)}]^3$$

$$\ge R_1^{(t)}(1 - \Theta(\frac{\eta C_0 C_2}{\alpha_1^{14}})\frac{1}{d^{3/2}})$$

So when $T_2 \le T_1 + \widetilde{O}(\frac{d^{1.5}\alpha_1^{12}}{\eta})$ iterations, we will have $R_1^{(t)} \ge R_1^{(s)}(1 - \Theta(\frac{\eta C_0 C_2}{d^{3/2}\alpha_1^{14}}))^{T_2-T_1} \ge \Omega(R_1^{(t)})$ for all $t \in [s, T_2]$, which means we have a lower bound $R_1^{(t)} \ge \frac{1}{d^{3/4}\alpha_1^2}$ throughout $t \in [T_1, T_2]$. This proves Lemma D.8a and also our induction on $R_1^{(t)}$.

**Proof of induction for $E_{1,2}^{(t)}$:** By Lemma D.4a, we can write

$$-\nabla_{E_{1,2}}L(W^{(t)}, E^{(t)}) = (1 + \widetilde{O}(\frac{\alpha_1^{O(1)}}{d^{3/2}}))\Sigma_{1,1}^{(t)}(-2E_{1,2}^{(t)}[R_2^{(t)}]^3 \pm O(\overline{R}_{1,2}^{(t)} + \varrho)[R_1^{(t)}]^{3/2}[R_2^{(t)}]^{3/2})$$

$$\pm \Sigma_{1,1}^{(t)}\widetilde{O}(\frac{\eta_E/\eta}{\sqrt{d}})\max\{[R_1^{(t)}]^3, \frac{\alpha_1^{O(1)}}{d^{5/2}}\}$$

$$= -\Theta(\Sigma_{1,1}^{(t)}[R_2^{(t)}]^3)E_{1,2}^{(t)} \pm O(\Sigma_{1,1}^{(t)})\Big((\overline{R}_{1,2}^{(t)} + \varrho)[R_1^{(t)}]^{3/2}[R_2^{(t)}]^{3/2} + \widetilde{O}(\frac{\eta_E/\eta}{\sqrt{d}})[R_1^{(t)}]^3\Big)$$

Since again from Induction D.1b,c that $\overline{R}_{1,2}^{(t)} \le \widetilde{O}(\varrho + \frac{1}{\sqrt{d}})$, $R_1^{(t)} = O(1)$, $R_2^{(t)} \in [\sqrt{\eta_E/\eta}, O(1)]$, we can obtain the update of $E_{1,2}^{(t)}$ as

$$E_{1,2}^{(t+1)} = E_{1,2}^{(t)}(1 - \Theta(\eta_E\Sigma_{1,1}^{(t)}[R_2^{(t)}]^3)) \pm \widetilde{O}(\eta_E\Sigma_{1,1}^{(t)})\Big((\varrho + \frac{1}{\sqrt{d}})[R_1^{(t)}]^{3/2}[R_2^{(t)}]^{3/2} + \widetilde{O}(\frac{\eta_E/\eta}{\sqrt{d}})[R_1^{(t)}]^3\Big)$$

$$= E_{1,2}^{(t)}(1 - \Theta(\eta_E\Sigma_{1,1}^{(t)}[R_2^{(t)}]^3)) \pm \widetilde{O}(\varrho + \frac{1}{\sqrt{d}})\eta_E\Sigma_{1,1}^{(t)}[R_1^{(t)}]^{3/2}$$

$$= E_{1,2}^{(t)}(1 - \Theta(\eta_E\Sigma_{1,1}^{(t)}[R_2^{(t)}]^3)) \pm \eta_E\Sigma_{1,1}^{(t)}J_{1,2}^{(t)}$$

where $J_{1,2}^{(t)} = \widetilde{C}(\varrho + \frac{1}{\sqrt{d}})[R_1^{(t)}]^{3/2} > 0$ and $\widetilde{C} = \widetilde{\Theta}(1)$ is larger than the hidden constant (including the $\mathsf{polylog}(d)$ factors) of $E_{2,1}^{(T_1)} \leq \widetilde{O}(\varrho + \frac{1}{\sqrt{d}})$ in Lemma C.13d. And then we can compute

$$
\begin{aligned}
J_{1,2}^{(t+1)} &= \widetilde{C}(\varrho + \frac{1}{\sqrt{d}})[R_1^{(t+1)}]^{3/2} \\
&= \widetilde{C}(\varrho + \frac{1}{\sqrt{d}})[R_1^{(t)}]^{3/2}(1 - \Theta(\eta\Sigma_{1,1}^{(t)})[R_1^{(t)}]^2)^{3/2} \qquad \text{(due to calculations in (D.4))} \\
&= J_{1,2}^{(t)}(1 - \Theta(\eta^{3/2}(\Sigma_{1,1}^{(t)})^{3/2})[R_1^{(t)}]^3) \qquad \text{(because } \eta\Sigma_{1,1}^{(t)} = \frac{\alpha_1^{O(1)}}{\mathsf{poly}(d)} \text{ is very small)}
\end{aligned}
$$

Now by Lemma C.13d, we know $|E_{1,2}^{(T_1)}| \leq J_{1,2}^{(T_1)}$; then we begin our induction that $|E_{1,2}^{(t)}| < (\log\log d)J_{1,2}^{(t)}$ at for all iterations $t \in [T_1, T_2]$. Now assume we have $|E_{1,2}^{(t)}| = \frac{1}{2}(\log\log d)J_{1,2}^{(t)}$[5], from above calculations it holds that $|E_{1,2}^{(t+1)}| = |E_{1,2}^{(t)}|(1 - \Theta(\eta\Sigma_{1,1}^{(t)}[R_1^{(t)}]^3))$. Then we would have

$$
\frac{J_{1,2}^{(t+1)}}{J_{1,2}^{(t)}} \geq (1 - \Theta(\eta^{3/2}(\Sigma_{1,1}^{(t)})^{3/2})[R_1^{(t)}]^3) \geq (1 - \Theta(\eta_E\Sigma_{1,1}^{(t)}[R_2^{(t)}]^3)) \geq \frac{|E_{1,2}^{(t+1)}|}{|E_{1,2}^{(t)}|}
$$
$$
\text{(because of the range of } R_1^{(t)} \text{ and } R_2^{(t)})
$$

This proved that $|E_{1,2}^{(t+1)}| \lesssim \log\log d \cdot J_{1,2}^{(t+1)} \leq \widetilde{O}(\varrho + \frac{1}{\sqrt{d}})[R_1^{(t+1)}]^{3/2}$ and also the induction can go on until $t = T_2$.

**Proof of the growth of $E_{2,1}^{(t)}$ and $T_2 \leq T_1 + O(\frac{d^{1.5}}{\eta\alpha_1^4})$:** According to Lemma D.4b, we can write down the update of $E_{2,1}^{(t)}$ as

$$
\begin{aligned}
-\nabla_{E_{2,1}}L(W^{(t)}, E^{(t)}) = (1 &\pm O(\frac{\alpha_1^{O(1)}}{d^{3/2}}))\Delta_{2,1}^{(t)} \\
&\pm O(\Sigma_{2,1}^{(t)})(|E_{2,1}^{(t)}|[R_1^{(t)}]^3 \pm O(\overline{R}_{1,2}^{(t)} + \varrho)[R_1^{(t)}]^{3/2}[R_2^{(t)}]^{3/2})
\end{aligned}
$$

Then, from Lemma D.3a,c and Induction D.1, we have

$$
O(\Sigma_{2,1}^{(t)})(|E_{2,1}^{(t)}|[R_1^{(t)}]^3 \pm O(\overline{R}_{1,2}^{(t)} + \varrho)[R_1^{(t)}]^{3/2}[R_2^{(t)}]^{3/2}) \leq O(\frac{\mathsf{polylog}(d)}{d^{3/2}\alpha_1^2})\Phi_2^{(t)} \leq O(\frac{1}{d^{3/2}\alpha_1})\Phi_2^{(t)}
$$

and also

$$
\left|(1 \pm \widetilde{O}(\frac{\alpha_1^6}{d^{0.3}}))C_0\Phi_2^{(t)}\alpha_1^6(B_{2,1}^{(t)})^3(B_{1,1}^{(t)})^3H_{2,2}^{(t)}\right| \geq \widetilde{\Theta}(\frac{\alpha_1^6}{d^{3/2}})\Phi_2^{(t)}
$$

Now by Lemma D.3a and Induction D.1a, it allow us to simplify the update to

$$
\begin{aligned}
E_{2,1}^{(t+1)} &= E_{2,1}^{(t)} - \eta_E\nabla_{E_{2,1}}L(W^{(t)}, E^{(t)}) \\
&= E_{2,1}^{(t)} + (1 \pm \frac{1}{\alpha_1^{\Omega(1)}})\eta_E C_0 C_2\alpha_1^6\Phi_2^{(t)}(B_{2,1}^{(t)})^3(B_{1,1}^{(t)})^3\mathcal{E}_{2,1}^{(t)} \\
&\geq E_{2,1}^{(t)} + \eta_E\widetilde{\Theta}(\frac{1}{d^{3/2}\alpha_1^6})\mathsf{sign}(B_{1,1}^{(t)})\mathsf{sign}(B_{2,1}^{(t)}) \qquad \text{(by Induction D.1 and Claim D.2)}
\end{aligned}
$$

---

[5]If we want $|E_{1,2}^{(t)}| > (\log\log d)J_{1,2}^{(t)}$, then as long as $\eta = \frac{1}{\mathsf{poly}(d)}$ is small enough, we can always assume to have found some iteration $t' \in (T_1, t]$ such that $|E_{1,2}^{(t')}| = \frac{1}{2}(\log\log d)J_{1,2}^{(t)}$, and we set $t = t'$ and start our argument from that iteration.

Now since $\text{sign}(B_{j,1}^{(t)}) = \text{sign}(B_{j,1}^{(T_1)})$, we know there is an iteration $T_{2,1}' \leq T_1 + O(\frac{d^{1/2}\alpha_1^{O(1)}}{\eta})$ such that for all $t \in [T_{2,1}', T_2]$, it holds

$$
\begin{aligned}
|E_{2,1}^{(t)}| &= \left| E_{2,1}^{(T_1)} + \sum_{t \in [T_1, T_{2,1}']} \Theta(\eta_E C_0 C_2 \alpha_1^6) \Phi_2^{(t)} (B_{2,1}^{(t)})^3 (B_{1,1}^{(t)})^3 [R_2^{(t)}]^3 \right| \\
&= \left| |E_{2,1}^{(T_1)}| \pm \sum_{s \in [T_1, T_{2,1}']} \eta_E \widetilde{\Theta}(\frac{1}{d^{3/2}\alpha_1^{O(1)}}) \right| \\
&\in \left[ 2|E_{2,1}^{(T_1)}|, \widetilde{O}(\frac{\alpha_1^{O(1)}}{d}) \right]
\end{aligned}
$$

and thus $\text{sign}(E_{2,1}^{(t)}) = \prod_{j \in [2]} \text{sign}(B_{j,1}^{(t)})$ and $|E_{2,1}^{(t)}|$ will be increasing during $t \in [T_{2,1}', T_2]$. Thus as long as $R_2^{(t)} \geq |E_{2,1}^{(t)}|$ continues to hold, after at most $\widetilde{\Theta}(\frac{d^{1.5}}{\eta\alpha_1^6})$ iterations starting from $T_1$, we shall have $|E_{2,1}^{(t)}| \geq \Omega(\sqrt{\eta_E/\eta})$.

However, in order to actually prove $|E_{2,1}^{(T_2)}| = \Theta(\sqrt{\eta_E/\eta})$, we will need to ensure that (1) there exist some constant $C = \Omega(\sqrt{\eta_E/\eta})$ such that $|E_{2,1}^{(t)}| > C$ while $R_2^{(s)} \geq \frac{1}{\log d}|E_{2,1}^{(t)}|$ for all $s \in [T_1, t]$; (2) we shall have a upper bound $|E_{2,1}^{(t)}| < O(\sqrt{\eta_E/\eta})$. They will be done below.

**Proof of** $E_{2,1}^{(T_2)} = \Theta(\sqrt{\eta_E/\eta})$ **and** $T_2 = T_1 + \widetilde{O}(\frac{d^{3/2}\alpha_1^{O(1)}}{\eta})$**:** In fact, Induction D.1c are already proved since we have already calculated the dynamics of $R_1^{(t)}$ and its upper bound and lower bound. In this part we are going to prove $T_2 = T_1 + \widetilde{\Theta}(\frac{d^{1.5}\alpha_1^{12}}{\eta})$ (which means that $R_2^{(t)} \leq |E_{2,1}|$ can be achieved in $\widetilde{O}(\frac{d^{3/2}\alpha_1^{12}}{\eta})$ many iterations). From Lemma D.5c, we can write down the update for $R_2^{(t)}$ as

$$
\begin{aligned}
R_2^{(t+1)} &= R_2^{(t)} - 2\eta \langle \nabla_{w_2} L(W^{(t)}, E^{(t)}), \Pi_{V^\perp} w_2^{(t)} \rangle + \eta^2 \|\Pi_{V^\perp} \nabla_{w_2} L(W^{(t)}, E^{(t)})\|_2^2 \\
&= R_2^{(t)} - \eta\Theta([R_2^{(t)}]^3) \left( \Sigma_{1,1}^{(t)} \Theta((E_{1,2}^{(t)})^2) + \sum_{\ell \in [2]} \Sigma_{2,\ell}^{(t)} \right) \\
&\quad \pm \eta O\left( \sum_{j,\ell} \Sigma_{j,\ell}^{(t)} E_{j,3-j}^{(t)} (\overline{R}_{1,2}^{(t)} + \varrho)[R_1^{(t)}]^{3/2}[R_2^{(t)}]^{3/2} \right) + \frac{\eta}{\text{poly}(d)}
\end{aligned}
$$

where we have used the fact that $\|\Pi_{V^\perp} \nabla_{w_2} L(W^{(t)}, E^{(t)})\|_2^2 \leq \widetilde{O}(d^2)$ from our assumption on the noise $\xi_p$ and a simple bound for $\Sigma_{j,\ell}^{(t)}$ as we have done before. Next we can resort to Induction D.1d that $|E_{1,2}^{(t)}| \leq \widetilde{O}(\varrho + \frac{1}{\sqrt{d}})[R_1^{(t)}]^{3/2}$ to derive

$$
\sum_{s \in [T_1, t]} \eta \Sigma_{1,1}^{(s)} \Theta((E_{1,2}^{(s)})^2) \leq \sum_{s \in [T_1, t]} \widetilde{O}(\varrho^2 + \frac{1}{d}) \eta \Sigma_{1,1}^{(s)} [R_1^{(s)}]^3
$$

$$
\leq \widetilde{O}(\varrho^2 + \frac{1}{d})
$$

which is because $\sum_{t \in [T_1, T_2]} \Theta(\eta \Sigma_{1,1}^{(t)})[R_1^{(t)}]^3 \leq O(1)$ and $\Sigma_{1,1}^{(t)} > 0$ as we have calculated in the proof of Induction D.1a above. Similarly, we can also bound

$$
\sum_{s \in [T_1, t]} \Sigma_{1,\ell}^{(s)} |E_{1,2}^{(s)}| (|\overline{R}_{1,2}^{(s)}| + \varrho)[R_1^{(s)}]^{3/2}[R_2^{(s)}]^{3/2} \leq \sum_{s \in [T_1, t]} \widetilde{O}(\varrho^2 + \frac{1}{d}) \eta \Sigma_{1,\ell}^{(s)} [R_1^{(s)}]^3 \leq \widetilde{O}(\varrho + \frac{1}{\sqrt{d}})
$$

Moreover, because $T_2 \leq T_1 + \widetilde{O}(\frac{d^{3/2}\alpha_1^{12}}{\eta})$ and $|E_{2,1}^{(t)}| \leq O(1)$, $\Phi_2^{(t)} \leq \alpha_1^{O(1)}$ from Induction D.1, we have for each $t \leq T_2$:

$$\sum_{s\in[T_1,t]} \eta\Sigma_{2,\ell}^{(s)}|E_{2,1}^{(s)}|(|\overline{R}_{1,2}^{(s)}| + \varrho)[R_1^{(s)}]^{3/2}[R_2^{(s)}]^{3/2} \leq \widetilde{O}(\frac{|E_{2,1}^{(s)}|^2}{d^{3/2}}) \sum_{s\in[T_1,t]} \eta\Phi_2^{(s)}\widetilde{O}(\varrho + \frac{1}{\sqrt{d}})$$

$$\leq \widetilde{O}(\frac{\eta}{d^{3/2}}) \cdot \widetilde{O}(\varrho + \frac{1}{\sqrt{d}}) \cdot \widetilde{O}(\frac{d^{3/2}\alpha_1^{12}}{\eta})$$

$$\leq \widetilde{O}(\varrho + \frac{1}{\sqrt{d}})\alpha_1^{O(1)} = o(\frac{1}{\log d})$$

Thus combining all the bounds above, we have proved that for each $t \in [T_1, T_2]$, it holds

$$R_2^{(t)} = R_2^{(T_1)} - \sum_{s\in[T_1,t]} \Theta(\eta\Sigma_{2,1}^{(t)})[R_2^{(t)}]^3 \pm o(1)$$

$$= R_2^{(T_1)} - \sum_{s\in[T_1,t]} \Theta(\eta C_0 C_2)E_{2,1}^{(t)}\alpha_1^6\Phi_2^{(t)}(B_{2,1}^{(t)})^3(B_{1,1}^{(t)})^3[R_2^{(t)}]^3 \pm o(\frac{1}{\log d}) \qquad\text{(D.5)}$$

$$= R_2^{(T_1)} - \sum_{s\in[T_1,t]} \eta E_{2,1}^{(t)}\widetilde{\Theta}(\frac{1}{d^{3/2}})\Phi_2^{(t)}[R_2^{(t)}]^3 \cdot \text{sign}(E_{2,1}^{(t)}) \cdot \text{sign}(B_{2,1}^{(T_1)}) \cdot \text{sign}(B_{1,1}^{(T_1)}) \pm o(\frac{1}{\log d})$$

$$\text{(D.6)}$$

where the last equality is because $\text{sign}(B_{j,\ell}^{(t)}) \equiv \text{sign}(B_{j,\ell}^{(T_1)})$ by Induction D.1a. Now from what we have proved above on the growth of $E_{2,1}^{(t)}$ that $\text{sign}(E_{2,1}^{(t)}) = \text{sign}(B_{1,1}^{(t)}B_{2,1}^{(t)}) \equiv \text{sign}(B_{1,1}^{(T_1)}B_{2,1}^{(T_1)})$ throughout the rest of phase II (which is just $t \in [T_{2,1}', T_2]$). Recall that

$$R_2^{(T_{2,1}')} = R_2^{(T_1)} \pm o(1), \quad \text{and} \quad E_{2,1}^{(t)} - E_{2,1}^{(T_{2,1}')} = \sum_{s\in[T_{2,1}',t]} \Theta(\eta_E C_0 C_2)\Phi_2^{(s)}(B_{2,1}^{(s)})^3(B_{1,1}^{(s)})^3$$

The above arguments imply for $t \in [T_{2,1}', T_2]$:

$$R_2^{(t+1)} = R_2^{(T_1)} - \sum_{s\in[T_{2,1}',t]} \Theta(\eta C_0 C_2)E_{2,1}^{(s)}\Phi_2^{(s)}(B_{2,1}^{(s)})^3(B_{1,1}^{(s)})^3[R_2^{(t)}]^3 \pm o(\frac{1}{\log d})$$

$$= R_2^{(T_1)} - \Theta(\frac{\eta}{\eta_E}|E_{2,1}^{(t)}|^2) - o(\frac{1}{\log d})$$

Now we can confirm

(1) there exist a constant $C = \Theta(\sqrt{\eta_E/\eta})$ such that $E_{2,1}^{(t)} = C$ if $R_2^{(t)}$ falls below $\frac{1}{\log d}|E_{2,1}^{(t)}|$;

(2) $T_2 = T_1 + \widetilde{\Theta}(\frac{d^{3/2}\alpha_1^{12}}{\eta})$ due to the growth $|E_{2,1}^{(t+1)}| = |E_{2,1}^{(t)}| + \eta_E\widetilde{\Theta}(\frac{1}{d^{3/2}\alpha_1^{12}\sqrt{\eta_E/\eta}})$ for $t \in [T_{2,1}', T_2]$.

which are the desired results.

**Proof of Induction D.1a:** We first obtain from Lemma D.7a that the update of $B_{1,1}^{(t)}$ can be written as

$$B_{1,1}^{(t+1)} = B_{1,1}^{(t)} + \eta\left(\Theta(\Sigma_{1,1}^{(t)})\text{sign}(B_{1,1}^{(t)})[R_1^{(t)}]^3 + \Gamma_{1,1}^{(t)} \pm \widetilde{O}(\alpha_1^{O(1)}/d^{5/2})\right)$$

Now by what we have calculated above in (D.4), the total decrease of $R_1^{(t)}$ is (since $R_1^{(t)}$ is monotone in this phase)

$$\sum_{t\in[T_1,T_2]} \Theta(\eta\Sigma_{1,1}^{(t)})[R_1^{(t)}]^3 \leq O(R_1^{(T_1)} - R_1^{(T_2)}) \leq O(1)$$

And also since $T_2 \leq T_1 + \widetilde{\Theta}(\frac{d^{3/2}\alpha_1^{12}}{\eta})$, we can bound

$$\sum_{t \in [T_1, T_2]} \widetilde{O}(\alpha_1^6/d^{5/2}) \leq \widetilde{O}(\alpha_1^{O(1)}/d^{5/2}) \cdot \widetilde{O}(\frac{d^{3/2}}{\eta \alpha_1^6}) \leq \widetilde{O}(\alpha_1^{O(1)}/d)$$

Now we consider how the $\Gamma_{1,1}^{(t)}$ term accumulates

$$\sum_{t \in [T_1, T_2]} \eta \Gamma_{1,1}^{(t)} = \left( \sum_{t \in [T_1, T'_{2,1}]} + \sum_{t \in [T'_{2,1}, T_2]} \right) \eta C_0 \alpha_1^6 E_{2,1}^{(t)} \Phi_2^{(t)} (B_{2,1}^{(t)})^3 (B_{1,1}^{(t)})^2 H_{2,2}^{(t)}$$

$$\overset{①}{=} \widetilde{O}(\frac{\alpha_1^{12}}{d}) + \sum_{t \in [T'_{2,1}, T_2]} O\left( \eta C_0 \alpha_1^6 \Phi_2^{(t)} |B_{2,1}^{(t)}|^3 |B_{1,1}^{(t)}|^3 H_{2,2}^{(t)} \right) \text{sign}(B_{1,1}^{(t)})$$

$$= \pm o(1) + O(1)\text{sign}(B_{1,1}^{(t)})$$

where in ① we have used $|E_{2,1}^{(t)}| \leq O(1) \leq O(B_{1,1}^{(t)})$ and $\text{sign}(E_{2,1}^{(t)}) = \prod_{j \in [2]} \text{sign}(B_{j,1}^{(t)})$ when $t \in [T'_{2,1}, T_2]$. These calculations tell us $B_{1,1}^{(t)} = B_{1,1}^{(T_1)} + O(1)\text{sign}(B_{1,1}^{(T_1)}) \pm O(\frac{1}{\alpha_1}) = \Theta(1)$ for all iterations $t \in [T_1, T_2]$. Similarly from Lemma D.7b, for $B_{2,1}^{(t)}$ we can also write

$$B_{2,1}^{(T+1)} = B_{2,1}^{(t)} + \eta \widetilde{O}(\alpha_1^{O(1)}/d^{5/2}) + \widetilde{O}(\frac{\alpha_1^6}{d}) E_{2,1}^{(t)} \Phi_1^{(t)} [R_1^{(t)}]^3$$

From similar calculations, it holds $B_{2,1}^{(t)} = B_{2,1}^{(T_1)} \pm \widetilde{O}(\alpha_1^{O(1)}/d)$, which proves that $B_{2,1}^{(t)} = B_{2,1}^{(T_1)}(1 \pm o(1))$ when $t \in [T_1, T_2]$. Now we turn to feature $v_2$. By Lemma D.6 we have for $j \in [2]$:

$$|\langle -\nabla_{w_j} L(W^{(t)}, E^{(t)}), v_2 \rangle| \leq \widetilde{O}(\frac{\alpha_2^6 \alpha_1^6}{d^{5/2}}) \left( \Phi_j^{(t)}(|E_{j,3-j}^{(t)}| + [R_j^{(t)}]^3) + \Phi_{3-j}^{(t)}(|E_{3-j,j}^{(t)}| [R_{3-j}^{(t)}]^3 + \frac{|E_{3-j,j}^{(t)}|^2}{d^{3/2}}) \right)$$

$$\leq \widetilde{O}(\frac{\alpha_2^6 \alpha_1^6}{d^{5/2}})$$

where the last inequality is from Lemma D.3a and Induction D.1c,d. Thus when $t \leq T_2 = T_1 + \widetilde{O}(\frac{d^{3/2}\alpha_1^{12}}{\eta})$ we would have

$$B_{j,2}^{(t)} = B_{j,2}^{(T_1)} \pm \widetilde{O}(\frac{\alpha_1^{O(1)}}{d}) = B_{j,2}^{(T_1)}(1 \pm o(1)) \qquad \text{since } B_{j,2}^{(T_1)} = \widetilde{\Theta}(\frac{1}{\sqrt{d}}) \text{ by Lemma C.13c}$$

Together they proved Induction D.1a and Lemma D.8a. Moreover, we have also

**Proof of Induction D.1b:** Firstly, we write down the update of $R_{1,2}^{(t)}$ using Lemma D.5b,d as follows:

$$R_{1,2}^{(t+1)} = R_{1,2}^{(t)} - \eta \langle \nabla_{w_1} L(W^{(t)}, E^{(t)}), \Pi_{V^\perp} w_2^{(t)} \rangle - \eta \langle \nabla_{w_2} L(W^{(t)}, E^{(t)}), \Pi_{V^\perp} w_1^{(t)} \rangle$$

$$+ \eta^2 \langle \Pi_{V^\perp} \nabla_{w_1} L(W^{(t)}, E^{(t)}), \Pi_{V^\perp} \nabla_{w_2} L(W^{(t)}, E^{(t)}) \rangle$$

$$= R_{1,2}^{(t)} + \eta \Sigma_{1,1}^{(t)}((-\Theta(\overline{R}_{1,2}^{(t)}) \pm O(\varrho))[R_1^{(t)}]^{5/2}[R_2^{(t)}]^{1/2} + \widetilde{O}(|E_{1,2}^{(t)}| + \frac{|E_{2,1}^{(t)}|^2}{d^{3/2}})R_1^{(t)}[R_2^{(t)}]^2)$$

$$+ \eta \left( \Sigma_{1,1}^{(t)} \Theta((E_{1,2}^{(t)})^2) + \sum_{\ell \in [2]} \Sigma_{2,\ell}^{(t)} \right)(-\Theta(\overline{R}_{1,2}^{(t)}) \pm O(\varrho))[R_2^{(t)}]^{5/2}[R_1^{(t)}]^{1/2}$$

$$+ O\left( \sum_{j,\ell} \eta \Sigma_{j,\ell}^{(t)} E_{j,3-j}^{(t)} R_2^{(t)}[R_1^{(t)}]^2 \right) + \frac{\eta}{\text{poly}(d)}$$

where in the last inequality we have used

$$|\langle \Pi_{V^\perp} \nabla_{w_1} L(W^{(t)}, E^{(t)}), \Pi_{V^\perp} \nabla_{w_2} L(W^{(t)}, E^{(t)}) \rangle|$$

$$\leq \|\Pi_{V^\perp} \nabla_{w_1} L(W^{(t)}, E^{(t)})\|_2 \|\Pi_{V^\perp} \nabla_{w_2} L(W^{(t)}, E^{(t)})\|_2 \leq \widetilde{O}(d)$$

Now from Induction D.1c,d that $R_2^{(t)} = \Theta(1)$ and $|E_{1,2}^{(t)}| \leq \widetilde{O}(\varrho + \frac{1}{\sqrt{d}})[R_1^{(t)}]^{3/2}$, $|E_{2,1}^{(t)}| \leq O(\sqrt{\eta_E/\eta})$, we can further obtain $|\Sigma_{2,2}^{(t)}| = \widetilde{O}(\frac{\alpha_1^{O(1)}}{d^{3/2}})|\Sigma_{2,1}^{(t)}|$, and the bound

$$R_{1,2}^{(t+1)} = R_{1,2}^{(t)}\Big(1 - \Theta(\eta\Sigma_{1,1}^{(t)})[R_1^{(t)}]^2 - \Theta(\eta(\Sigma_{1,1}^{(t)}(E_{1,2}^{(t)})^2 + \Sigma_{2,1}^{(t)}))[R_2^{(t)}]^2\Big)$$

$$\pm \eta O(\varrho)[R_2^{(t)}]^{1/2}[R_1^{(t)}]^{1/2}\Big(O(\Sigma_{1,1}^{(t)})[R_1^{(t)}]^2 + \Big(\Sigma_{1,1}^{(t)}\Theta((E_{1,2}^{(t)})^2) + \Sigma_{2,1}^{(t)}\Big)[R_2^{(t)}]^2\Big)$$

Notice here that there exist a constant $C = \Theta(1)$, whenever $|R_{1,2}^{(t)}| \geq C(\varrho + \frac{1}{\sqrt{d}})[R_2^{(t)}]^{1/2}[R_1^{(t)}]^{1/2}$, it will holds

$$R_{1,2}^{(t+1)} = R_{1,2}^{(t)}\Big(1 - \Theta(\eta\Sigma_{1,1}^{(t)}[R_1^{(t)}]^2) - \Theta(\eta(\Sigma_{1,1}^{(t)}(E_{1,2}^{(t)})^2 + \Sigma_{2,1}^{(t)}))[R_2^{(t)}]^2\Big)$$

$$= R_{1,2}^{(t)}\Big(1 - \Theta(\eta\Sigma_{1,1}^{(t)}[R_1^{(t)}]^2) - \Theta(\eta(\Sigma_{1,1}^{(t)}(E_{1,2}^{(t)})^2 + \frac{\alpha_1^6}{d^{3/2}}\Sigma_{2,1}^{(t)}))[R_2^{(t)}]^2\Big)$$

Thus we can go through the same analysis as in the proof of induction for $E_{1,2}^{(t)}$ to derive that

$$|R_{1,2}^{(t)}| \leq \widetilde{O}(\varrho + \frac{1}{\sqrt{d}})[R_2^{(t)}]^{1/2}[R_1^{(t)}]^{1/2}$$

which is the desired result. Note that at the end of phase II

$$
\begin{array}{ccc}
\text{Induction D.1a} & \Longrightarrow & \text{Lemma D.8a} \\
\text{Induction D.1b,c} & \Longrightarrow & \text{Lemma D.8b} \\
\text{Induction D.1d} & \Longrightarrow & \text{Lemma D.8c}
\end{array}
$$

We now complete the proof of Lemma D.8. $\qquad\square$

## E   Phase III: The Acceleration Effect of Prediction Head

We shall prove in this section that the growth of $E_{2,1}^{(t)}$ in the previous phase creates an acceleration effect to the growth of $B_{2,2}^{(t)}$, which will finally outrun the growth of $B_{2,1}^{(t)}$ to win the lottery. We define

$$T_3 := \min\left\{t : |B_{2,2}^{(t)}| \geq \frac{1}{2}\min\{|B_{1,1}^{(t)}|, \sqrt{\frac{\eta}{\eta_E}}|E_{2,1}^{(t)}|\}\right\} \tag{E.1}$$

and we call iterations $t \in [T_2, T_3]$ as the phase III of training and $t \geq T_3$ as the end phase of training.

### E.1   Induction in Phase III

**Inductions E.1** (Phase III). *During $t \in [T_2, T_3]$, we hypothesize the following conditions holds.*

*(a)* $|B_{1,1}^{(t)}| = \Theta(1)$, $B_{2,1}^{(t)} = B_{2,1}^{(T_2)}(1 \pm o(1))$, $B_{1,2}^{(t)} = B_{1,2}^{(T_2)}(1 \pm o(1))$, $|B_{2,2}^{(t)}| \in [|B_{2,2}^{(T_2)}|, O(1)]$;

*(b)* $|E_{2,1}^{(t)}| = \Theta(\sqrt{\eta_E/\eta})$, $\text{sign}(E_{2,1}^{(t)}) = \text{sign}(E_{2,1}^{(T_2)})$ *and* $|E_{1,2}^{(t)}| \leq \widetilde{O}(\varrho + \frac{1}{\sqrt{d}})[R_1^{(t)}]^{3/2}[R_2^{(t)}]^{3/2}$;

*(c)* $R_1^{(t)} \in [\Omega(\frac{1}{d}), O(\frac{d^{o(1)}}{d^{3/4}})]$, $[R_2^{(t)}] \in [\frac{1}{\sqrt{d}}, O(\frac{1}{\log d}\sqrt{\eta_E/\eta})]$.

As usual, before we prove the induction, we need to derive some useful claims. But firstly we shall give a much cleaner form of $\nabla_{E_{j,3-j}}L(W^{(t)}, E^{(t)})$ to help us understand the learning process of phase III and the end phase.

**Fact E.2.** Let us write

$$\Xi_j^{(t)} = C_0 C_1 \alpha_1^6 \alpha_2^6 \Phi_j^{(t)}\Big((B_{1,1}^{(t)})^6(B_{2,2}^{(t)})^6 + (B_{2,1}^{(t)})^6(B_{1,2}^{(t)})^6\Big)$$

$$\Delta_{j,\ell}^{(t)} = C_0 \Phi_j^{(t)} \alpha_\ell^6 (B_{j,\ell}^{(t)})^3(B_{3-j,\ell}^{(t)})^3 C_2 \mathcal{E}_{j,3-j}^{(t)}$$

Then the gradient of $E_{j,3-j}^{(t)}$ can be written as

$$-\nabla_{E_{j,3-j}}L(W^{(t)}, E^{(t)}) = -\Xi_j^{(t)}E_{j,3-j}^{(t)} + \sum_{\ell \in [2]}\Delta_{j,\ell}^{(t)} - \sum_{\ell \in [2]}\Sigma_{j,\ell}^{(t)}\nabla_{E_{j,3-j}}\mathcal{E}_{j,3-j}^{(t)}$$

*Proof.* By expanding the gradients of $E_{j,3-j}^{(t)}$, we can verify by checking each monomial of polynomials of $B_{j,\ell}$ to obtain the first term, and leave the $\mathcal{E}_{j,3-j}^{(t)}$ part for the second term. $\qquad\square$

**Lemma E.3** (variables control at phase III). *For $t \in [T_2, T_3]$, if Induction E.1 holds at iteration $t$, then we have*

(a) $\Phi_1^{(t)} = \widetilde{\Theta}(\frac{1}{\alpha_1^{12}})$, $[Q_2^{(t)}]^{-2} = \Theta(C_2[R_2^{(t)}]^3 + C_1\alpha_2^6(B_{2,2}^{(t)})^6)$, $U_2^{(t)} = \Theta(C_1(\alpha_1^6(E_{2,1}^{(t)})^2 + \alpha_2^6(B_{2,2}^{(t)})^6))$;

(b) $H_{1,1}^{(t)} = \Theta(C_1\alpha_1^6)$, $H_{1,2}^{(t)} \leq O(C_2[R_1^{(t)}]^3) + \widetilde{O}(\frac{\alpha_2^6}{d^3})$;

(c) $H_{2,1}^{(t)} = \Theta(C_1\alpha_1^6(E_{2,1}^{(t)})^2)$, $H_{2,2}^{(t)} = \Theta(C_2[R_2^{(t)}]^3)$;

(d) $\Sigma_{1,2}^{(t)} \leq \widetilde{O}(\frac{|E_{1,2}^{(t)}|}{d^{3/2}})\Sigma_{1,1}^{(t)}$;

(e) $\mathcal{E}_{j,3-j}^{(t)} = (1 \pm o(1))\mathcal{E}_j^{(t)} = O(C_2[R_j^{(t)}]^3)$

*Proof.* Assuming Induction E.1 holds at $t \in [T_2, T_3]$, we can recall the expression of these variables and prove their bounds directly. The bounds for $\Phi_1$ and $H_{1,1}$ comes from $|B_{1,1}^{(t)}| = \Theta(1)$ and $|B_{1,2}^{(t)}|, |E_{1,2}^{(t)}| = o(1)$. The bounds for $Q_2, U_2$ comes from our definition of $T_3$ in (E.1). The rest of the claims can be derived by similar arguments using Induction E.1. $\qquad\square$

## E.2 Gradient Lemmas for Phase III

In this subsection, we would give some gradient lemmas concerning the dynamics of our network in Phase III.

**Lemma E.4** (learning feature $v_2$ in phase III). *For each $t \in [T_2, T_3]$, if Induction E.1 holds at iteration $t$, then we have:*

(a) $\langle -\nabla_{w_1} L(W^{(t)}, E^{(t)}), v_2 \rangle = \Theta(\frac{(B_{1,2}^{(t)})^2}{(B_{2,2}^{(t)})^2})E_{2,1}^{(t)}\Lambda_{2,2}^{(t)} \pm \widetilde{O}(\frac{\alpha_1^{O(1)}}{d^4})|E_{2,1}^{(t)}|^2\Phi_2^{(t)} \pm \widetilde{O}(\frac{\alpha_1^{O(1)}}{d^{5/2}})$;

(b) $\langle -\nabla_{w_2} L(W^{(t)}, E^{(t)}), v_2 \rangle = (1 \pm \widetilde{O}(\frac{1}{d}))\Lambda_{2,2}^{(t)}$

*Proof.* Since $\langle -\nabla_{w_j} L(W^{(t)}, E^{(t)}), v_2 \rangle = \Lambda_{j,2}^{(t)} + \Gamma_{j,2}^{(t)} - \Upsilon_{j,2}^{(t)}$, let us write down the definition of $\Lambda_{j,2}^{(t)}, \Gamma_{j,2}^{(t)}, \Upsilon_{j,2}^{(t)}$ respectively:

$$\Lambda_{j,2}^{(t)} = C_0\alpha_2^6\Phi_j^{(t)}H_{j,1}^{(t)}(B_{j,2}^{(t)})^5$$
$$\Gamma_{j,2}^{(t)} = C_0\alpha_2^6\Phi_{3-j}^{(t)}E_{3-j,j}^{(t)}(B_{3-j,2}^{(t)})^3(B_{j,2}^{(t)})^2H_{3-j,1}^{(t)}$$
$$\Upsilon_{j,2}^{(t)} = C_0\alpha_1^6\left(\Phi_j^{(t)}(B_{j,1}^{(t)})^3(B_{j,2}^{(t)})^2K_{j,2}^{(t)} + \Phi_{3-j}^{(t)}E_{3-j,j}^{(t)}(B_{3-j,1}^{(t)})^3(B_{j,2}^{(t)})^2K_{3-j,2}^{(t)}\right)$$

Again we decompose $\Upsilon_{j,2}^{(t)} = \Upsilon_{j,2,1}^{(t)} + \Upsilon_{j,2,2}^{(t)}$ as in the proof of Lemma D.6, where

$$\Upsilon_{j,2,1}^{(t)} = C_0\alpha_1^6\Phi_j^{(t)}(B_{j,1}^{(t)})^3(B_{j,2}^{(t)})^2K_{j,2}^{(t)}, \qquad \Upsilon_{j,2,2}^{(t)} = \Phi_{3-j}^{(t)}E_{3-j,j}^{(t)}(B_{3-j,1}^{(t)})^3(B_{3-j,2}^{(t)})^2K_{3-j,2}$$

This gives

$$\begin{aligned}
\Lambda_{j,2}^{(t)} - \Upsilon_{j,2,1}^{(t)} &= C_0\alpha_2^6\Phi_j^{(t)}(B_{j,2}^{(t)})^5H_{j,1}^{(t)} - C_0\alpha_1^6\Phi_j^{(t)}(B_{j,1}^{(t)})^3(B_{j,2}^{(t)})^2K_{j,2}^{(t)} \\
&= C_0\alpha_2^6C_1\alpha_1^6\Phi_j^{(t)}(B_{j,2}^{(t)})^5\left(E_{j,3-j}^{(t)}(B_{3-j,1}^{(t)})^3(B_{j,1}^{(t)})^3 + (E_{j,3-j}^{(t)})^2(B_{3-j,1}^{(t)})^6\right) \\
&\quad - C_0\alpha_2^6C_1\alpha_1^6\Phi_j^{(t)}(B_{j,2}^{(t)})^2(B_{3-j,2}^{(t)})^3E_{j,3-j}^{(t)}\left((B_{j,1}^{(t)})^6 + E_{j,3-j}^{(t)}(B_{3-j,1}^{(t)})^3(B_{j,1}^{(t)})^3\right) \\
&\quad + C_0\alpha_2^6\Phi_j^{(t)}(B_{j,2}^{(t)})^5C_2\mathcal{E}_{j,3-j}^{(t)}
\end{aligned}$$

When $j = 1$, from Induction E.1 and Lemma E.3a (which gives $\Phi_1^{(t)} \leq \alpha_1^{O(1)}\Phi_2^{(t)}$), we can crudely obtain

$$\left| C_0\alpha_2^6 C_1\alpha_1^6 \Phi_1^{(t)}(B_{1,2}^{(t)})^5 \left( E_{1,2}^{(t)}(B_{2,1}^{(t)})^3(B_{1,1}^{(t)})^3 + (E_{1,2}^{(t)})^2(B_{2,1}^{(t)})^6 \right) \right| \leq \widetilde{O}(\frac{\alpha_1^{O(1)}}{d^4})\Phi_1^{(t)}|E_{1,2}^{(t)}|$$

$$\left| C_0\alpha_2^6 C_1\alpha_1^6 \Phi_1^{(t)}(B_{1,2}^{(t)})^2(B_{2,2}^{(t)})^3 E_{1,2}^{(t)} \left( (B_{1,1}^{(t)})^6 + E_{1,2}^{(t)}(B_{2,1}^{(t)})^3(B_{1,1}^{(t)})^3 \right) \right| \leq \widetilde{O}(\frac{\alpha_1^{O(1)}}{d})\Lambda_{2,2}^{(t)}|E_{1,2}^{(t)}|$$

$$\left| C_0\alpha_2^6 \Phi_1^{(t)}(B_{1,2}^{(t)})^5 C_2 \mathcal{E}_{1,2}^{(t)} \right| = \widetilde{O}(\frac{\alpha_1^6}{d^{5/2}})\Sigma_{1,1}^{(t)}[R_1^{(t)}]^3$$

So we have

$$\Lambda_{1,2}^{(t)} - \Upsilon_{1,2,1}^{(t)} = \widetilde{O}(\frac{\alpha_1^6}{d^{5/2}})\Sigma_{1,1}^{(t)}[R_1^{(t)}]^3 \pm \widetilde{O}(\frac{\alpha_1^{O(1)}}{d})\Lambda_{2,2}^{(t)}|E_{1,2}^{(t)}|$$

When $j = 2$, we can also derive using Lemma E.3 about $H_{2,1}^{(t)}$ and Induction E.1 about $B_{2,1}^{(t)}$ and some rearrangement to obtain

$$C_0\alpha_2^6 \Phi_2^{(t)}(B_{2,2}^{(t)})^5 \left[ C_1\alpha_1^6 \left( E_{2,1}^{(t)}(B_{1,1}^{(t)})^3(B_{2,1}^{(t)})^3 + (E_{2,1}^{(t)})^2(B_{1,1}^{(t)})^6 \right) + C_2\mathcal{E}_{2,1}^{(t)} \right] = (1 \pm \widetilde{O}(\frac{1}{d}))\Lambda_{2,2}^{(t)}$$

$$\left| C_0\alpha_2^6 C_1\alpha_1^6 \Phi_2^{(t)}(B_{2,2}^{(t)})^2(B_{1,2}^{(t)})^3 E_{2,1}^{(t)} \left( (B_{2,1}^{(t)})^6 + E_{2,1}^{(t)}(B_{1,1}^{(t)})^3(B_{2,1}^{(t)})^3 \right) \right| \leq \widetilde{O}(\frac{\alpha_1^{O(1)}}{d^3})|E_{2,1}^{(t)}|\Phi_2^{(t)}$$

which leads to the approximation

$$\Lambda_{2,2}^{(t)} - \Upsilon_{1,2,2}^{(t)} = (1 \pm \widetilde{O}(\frac{1}{d}))\Lambda_{2,2}^{(t)} \pm \widetilde{O}(\frac{\alpha_1^{O(1)}}{d^3})|E_{2,1}^{(t)}|\Phi_2^{(t)}$$

Similarly, we can also calculate

$$\Gamma_{j,2}^{(t)} - \Upsilon_{j,2,2}^{(t)} = C_0\alpha_2^6 \Phi_{3-j}^{(t)} E_{3-j,j}^{(t)}(B_{3-j,2}^{(t)})^3(B_{j,2}^{(t)})^2 H_{3-j,1}^{(t)} - C_0\alpha_1^6 \Phi_{3-j}^{(t)} E_{3-j,j}^{(t)}(B_{3-j,1}^{(t)})^3(B_{j,2}^{(t)})^2 K_{3-j,2}^{(t)}$$

$$= C_0\alpha_2^6 C_1\alpha_1^6 \Phi_{3-j}^{(t)}(B_{3-j,2}^{(t)})^3(B_{j,2}^{(t)})^2 E_{3-j,j}^{(t)} \left( E_{3-j,j}^{(t)}(B_{j,1}^{(t)})^3(B_{3-j,1}^{(t)})^3 + (E_{3-j,j}^{(t)})^2(B_{j,1}^{(t)})^6 \right)$$

$$- C_0\alpha_2^6 C_1\alpha_1^6 \Phi_{3-j}^{(t)}(B_{j,2}^{(t)})^5(E_{3-j,j}^{(t)})^2 \left( (B_{3-j,1}^{(t)})^6 + E_{3-j,j}^{(t)}(B_{j,1}^{(t)})^3(B_{3-j,1}^{(t)})^3 \right)$$

$$+ C_0\alpha_2^6 \Phi_{3-j}^{(t)} E_{3-j,j}^{(t)}(B_{3-j,2}^{(t)})^3(B_{j,2}^{(t)})^2 C_2 \mathcal{E}_{3-j,j}^{(t)}$$

When $j = 1$, following similar procedure as above, we can apply Induction E.1 and Lemma E.3 to give

$$\Gamma_{1,2}^{(t)} - \Upsilon_{1,2,2}^{(t)} = \Theta(\frac{(B_{1,2}^{(t)})^2}{(B_{2,2}^{(t)})^2})E_{2,1}^{(t)}\Lambda_{2,2}^{(t)} \pm \widetilde{O}(\frac{\alpha_1^{O(1)}}{d^4})|E_{2,1}^{(t)}|^2\Phi_2^{(t)}$$

Note that the first term on the RHS dominates the term $\pm\widetilde{O}(\frac{\alpha_1^{O(1)}}{d})\Lambda_{2,2}^{(t)}|E_{1,2}^{(t)}|$ in the approximation for $\Lambda_{1,2}^{(t)} - \Upsilon_{1,2,1}^{(t)}$ due to Induction E.1a,b. When $j = 2$, since $\Phi_1^{(t)} \leq \widetilde{\Theta}(\frac{1}{\alpha_1^{12}}) \leq \alpha_1^{O(1)}\Phi_2^{(t)}H_{2,1}^{(t)}$ in this phase and $|B_{1,1}^{(t)}| = O(1)$, we can derive

$$|\Gamma_{2,2}^{(t)} - \Upsilon_{2,2,2}^{(t)}| \leq \widetilde{\Theta}(\frac{\alpha_1^{O(1)}}{d^3})(E_{1,2}^{(t)})^2\Phi_1^{(t)} + \alpha_1^{O(1)}(E_{1,2}^{(t)})^2\Lambda_{2,2}^{(t)}$$

It can be seen that $(E_{1,2}^{(t)})^2\Phi_1^{(t)} \leq (E_{2,1}^{(t)})^2\Phi_2^{(t)}$ by Induction E.1 and Lemma E.3. And by similar arguments we can have $(1 \pm \widetilde{O}(\frac{1}{d}))\Lambda_{2,2}^{(t)} \geq \frac{1}{d^{\Omega(1)}}\widetilde{O}(\frac{\alpha_1^{O(1)}}{d^3})|E_{2,1}^{(t)}|\Phi_2^{(t)}$. Combining all the results above, we can finish the proof. $\square$

**Lemma E.5** (learning feature $v_1$ in Phase III). *For each $t \in [T_2, T_3]$, if Induction E.1 holds at iteration $t$, then we have: (recall that $\Delta$-notation is from Fact E.2 )*

*(a)* $\langle -\nabla_{w_1}L(W^{(t)}, E^{(t)}), v_1 \rangle = \Theta(\Sigma_{1,1}^{(t)}[R_1^{(t)}]^3) \pm O(\frac{(B_{1,2}^{(t)})^3}{(B_{2,2}^{(t)})^3} + \frac{1}{\sqrt{d}})\alpha_1^{O(1)}\Lambda_{2,2}^{(t)} + \frac{E_{2,1}^{(t)}}{B_{1,1}^{(t)}}\Delta_{2,1}^{(t)} - \frac{B_{2,2}^{(t)}}{B_{1,1}^{(t)}}\Lambda_{2,2}^{(t)};$

(b) $\langle -\nabla_{w_2} L(W^{(t)}, E^{(t)}), v_1 \rangle = \widetilde{O}(\frac{\alpha_1^{O(1)}}{d^{5/2}}) \Phi_2^{(t)} [R_2^{(t)}]^3 \pm \widetilde{O}(\frac{\alpha_1^{O(1)}}{d}) \Lambda_{2,2}^{(t)} \pm \widetilde{O}(\frac{\alpha_1^{O(1)}}{d^3})$

*Proof.* Recall that $\langle -\nabla_{w_j} L(W^{(t)}, E^{(t)}), v_1 \rangle = \Lambda_{j,1}^{(t)} + \Gamma_{j,1}^{(t)} - \Upsilon_{j,1}^{(t)}$. Similar to the proof of Lemma E.4, we can decompose $\Upsilon_{j,1}^{(t)} = \Upsilon_{j,1,1}^{(t)} + \Upsilon_{j,1,2}^{(t)}$ and do similar calculations:

$$
\begin{aligned}
\Lambda_{j,1}^{(t)} - \Upsilon_{j,1,1}^{(t)} &= C_0 C_1 \alpha_1^6 \alpha_2^6 \Phi_j^{(t)} (B_{j,1}^{(t)})^5 \left( E_{j,3-j}^{(t)} (B_{3-j,2}^{(t)})^3 (B_{j,2}^{(t)})^3 + (E_{j,3-j}^{(t)})^2 (B_{3-j,2}^{(t)})^6 \right) \\
&\quad - C_0 C_1 \alpha_1^6 \alpha_2^6 \Phi_j^{(t)} (B_{j,1}^{(t)})^2 (B_{3-j,1}^{(t)})^3 E_{j,3-j}^{(t)} \left( (B_{j,2}^{(t)})^6 + E_{j,3-j}^{(t)} (B_{3-j,2}^{(t)})^3 (B_{j,2}^{(t)})^3 \right) \\
&\quad + C_0 \alpha_1^6 \Phi_j^{(t)} (B_{j,1}^{(t)})^5 C_2 \mathcal{E}_{j,3-j}^{(t)}
\end{aligned}
$$

When $j = 1$, from Induction E.1 and Lemma E.3a we know $\Phi_1^{(t)} \leq \alpha_1^{(O(1))}$ during $t \in [T_2, T_3]$, which allow us to derive

$$
\begin{aligned}
& C_0 C_1 \alpha_1^6 \alpha_2^6 \Phi_1^{(t)} (B_{1,1}^{(t)})^5 \left( E_{1,2}^{(t)} (B_{2,2}^{(t)})^3 (B_{1,2}^{(t)})^3 + (E_{1,2}^{(t)})^2 (B_{2,2}^{(t)})^6 \right) \\
&\leq \widetilde{O}(\Sigma_{1,1}^{(t)} (E_{1,2}^{(t)})^2) + C_0 C_1 \alpha_1^6 \alpha_2^6 (B_{1,1}^{(t)})^5 E_{1,2}^{(t)} (B_{2,2}^{(t)})^3 (B_{1,2}^{(t)})^3 \\
&\leq O(\frac{(B_{1,2}^{(t)})^3}{(B_{2,2}^{(t)})^3}) \alpha_1^{O(1)} \Lambda_{2,2}^{(t)} |E_{1,2}^{(t)}| + \Theta(\Sigma_{1,1}^{(t)} [R_1^{(t)}]^3)
\end{aligned}
$$

And

$$
\left| C_0 C_1 \alpha_1^6 \alpha_2^6 \Phi_1^{(t)} (B_{1,1}^{(t)})^2 (B_{2,1}^{(t)})^3 E_{1,2}^{(t)} \left( (B_{1,2}^{(t)})^6 + E_{1,2}^{(t)} (B_{2,2}^{(t)})^3 (B_{1,2}^{(t)})^3 \right) \right| \leq \widetilde{O}(\frac{1}{d^{3/2}}) |E_{1,2}^{(t)}| \Lambda_{2,2}
$$

which can be summarized as

$$
\Lambda_{1,1}^{(t)} - \Upsilon_{1,1,1}^{(t)} = \Theta(\Sigma_{1,1}^{(t)} [R_1^{(t)}]^3) \pm O(\frac{(B_{1,2}^{(t)})^3}{(B_{2,2}^{(t)})^3} + (B_{2,1}^{(t)})^3 + \frac{1}{\sqrt{d}}) |E_{1,2}^{(t)}| \alpha_1^{O(1)} \Lambda_{2,2}^{(t)}
$$

A similar calculation also gives

$$
\Lambda_{2,1}^{(t)} - \Upsilon_{2,1,1}^{(t)} = \widetilde{O}(\frac{\alpha_1^{O(1)}}{d^{5/2}}) \Phi_2^{(t)} [R_2^{(t)}]^3 \pm \widetilde{O}(\frac{\alpha_1^{O(1)}}{d^4}) \Phi_2^{(t)} |E_{2,1}^{(t)}| \pm \widetilde{O}(\frac{\alpha_1^{O(1)}}{d}) \Lambda_{2,2}^{(t)} B_{2,2}^{(t)}
$$

Now we turn to the other terms in the gradient, from similar calculations in the proof of Lemma D.6, we have

$$
\begin{aligned}
\Gamma_{j,1}^{(t)} - \Upsilon_{j,1,2}^{(t)} &= C_0 \alpha_2^6 C_1 \alpha_1^6 \Phi_{3-j}^{(t)} (B_{3-j,1}^{(t)})^3 (B_{j,1}^{(t)})^2 E_{3-j,j}^{(t)} \left( E_{3-j,j}^{(t)} (B_{j,2}^{(t)})^3 (B_{3-j,2}^{(t)})^3 + (E_{3-j,j}^{(t)})^2 (B_{j,2}^{(t)})^6 \right) \\
&\quad - C_0 \alpha_2^6 C_1 \alpha_1^6 \Phi_{3-j}^{(t)} (B_{j,1}^{(t)})^5 (E_{3-j,j}^{(t)})^2 \left( (B_{3-j,2}^{(t)})^6 + E_{3-j,j}^{(t)} (B_{j,2}^{(t)})^3 (B_{3-j,2}^{(t)})^3 \right) \\
&\quad + C_0 \alpha_2^6 \Phi_{3-j}^{(t)} E_{3-j,j}^{(t)} (B_{3-j,1}^{(t)})^3 (B_{j,1}^{(t)})^2 C_2 \mathcal{E}_{3-j,j}^{(t)}
\end{aligned}
$$

which also similarly gives

$$
\Gamma_{1,1}^{(t)} - \Upsilon_{1,1,2}^{(t)} = \frac{E_{2,1}^{(t)}}{B_{1,1}^{(t)}} \Delta_{2,1}^{(t)} - \frac{B_{2,2}^{(t)}}{B_{1,1}^{(t)}} \Lambda_{2,2}^{(t)} \pm \widetilde{O}(\frac{\alpha_1^{O(1)}}{d^{3/2}}) \Lambda_{2,2}^{(t)}
$$

and

$$
|\Gamma_{2,1}^{(t)} - \Upsilon_{2,1,2}^{(t)}| \leq \widetilde{O}(\frac{\alpha_1^{O(1)}}{d}) \Phi_1^{(t)} ((E_{1,2}^{(t)})^2 + |E_{1,2}^{(t)}| [R_1^{(t)}]^3) \leq \widetilde{O}(\frac{\alpha_1^{O(1)}}{d^3})
$$

which finishes the proof. $\qquad\square$

**Lemma E.6** (reducing noise in phase III). *Suppose Induction E.1 holds at $t \in [T_2, T_3]$, then we have*

(a) $\quad \langle -\nabla_{w_1} L(W^{(t)}, E^{(t)}), \Pi_{V^\perp} w_1^{(t)} \rangle = -\Theta([R_1^{(t)}]^3) \Big( \Sigma_{1,1}^{(t)} + \sum_{\ell \in [2]} \Sigma_{2,\ell}^{(t)} (E_{2,1}^{(t)})^2 \Big)$

$$\pm O\Big( \sum_{j,\ell} \Sigma_{j,\ell}^{(t)} E_{j,3-j}^{(t)} (\overline{R}_{1,2}^{(t)} + \varrho)[R_1^{(t)}]^{3/2} [R_2^{(t)}]^{3/2} \Big);$$

(b) $\quad \langle -\nabla_{w_1} L(W^{(t)}, E^{(t)}), \Pi_{V^\perp} w_2^{(t)} \rangle = \Big( \Sigma_{1,1}^{(t)} + \sum_{\ell \in [2]} \Sigma_{2,\ell}^{(t)} (E_{2,1}^{(t)})^2 \Big) (-\Theta(\overline{R}_{1,2}^{(t)}) \pm O(\varrho))[R_1^{(t)}]^{5/2} [R_2^{(t)}]^{1/2}$

$$+ O\Big( \sum_{(j,\ell) \neq (1,2)} \Sigma_{j,\ell}^{(t)} E_{j,3-j}^{(t)} R_1^{(t)} [R_2^{(t)}]^2 \Big)$$

(c) $\quad \langle -\nabla_{w_2} L(W^{(t)}, E^{(t)}), \Pi_{V^\perp} w_2^{(t)} \rangle = -\Theta([R_2^{(t)}]^3) \Big( \sum_{\ell \in [2]} \Sigma_{1,1}^{(t)} \Theta((E_{1,2}^{(t)})^2) + \sum_{\ell \in [2]} \Sigma_{2,\ell}^{(t)} \Big)$

$$\pm O\Big( \sum_{j,\ell} \Sigma_{j,\ell}^{(t)} E_{j,3-j}^{(t)} (\overline{R}_{1,2}^{(t)} + \varrho)[R_1^{(t)}]^{3/2} [R_2^{(t)}]^{3/2} \Big);$$

(d) $\quad \langle -\nabla_{w_2} L(W^{(t)}, E^{(t)}), \Pi_{V^\perp} w_1^{(t)} \rangle = \Big( \Sigma_{1,1}^{(t)} \Theta((E_{1,2}^{(t)})^2) + \sum_{\ell \in [2]} \Sigma_{2,\ell}^{(t)} \Big) (-\Theta(\overline{R}_{1,2}^{(t)}) \pm O(\varrho))[R_2^{(t)}]^{5/2} [R_1^{(t)}]^{1/2}$

$$+ O\Big( \sum_{(j,\ell) \neq (1,2)} \Sigma_{j,\ell}^{(t)} E_{j,3-j}^{(t)} R_2^{(t)} [R_1^{(t)}]^2 \Big)$$

*Proof.* The proof of Lemma E.6 is very similar to Lemma D.5, but we write it down to stress some minor differences. As in (B.2), we first write down

$$\langle -\nabla_{w_1} L(W^{(t)}, E^{(t)}), \Pi_{V^\perp} w_1^{(t)} \rangle = -\sum_{j,\ell} \Sigma_{j,\ell}^{(t)} \langle \nabla_{w_1} \mathcal{E}_{j,3-j}^{(t)}, w_1^{(t)} \rangle$$

**Proof of (a):** Combine the bounds above, we can obtain for each $j \in [2]$: $\Sigma_{1,2}^{(t)} = \widetilde{O}(E_{1,2}^{(t)}/d^{3/2}) \Sigma_{1,1}^{(t)}$. We can then directly apply Claim B.1 to prove Lemma E.6a as follows

$$\langle -\nabla_{w_1} L(W^{(t)}, E^{(t)}), \Pi_{V^\perp} w_1^{(t)} \rangle$$
$$= (1 \pm \widetilde{O}(E_{1,2}^{(t)}/d^{3/2})) \Sigma_{1,1}^{(t)} \Big( -\Theta([R_1^{(t)}]^3) \pm O(E_{1,2}^{(t)})(\overline{R}_{1,2}^{(t)} + \varrho)[R_1^{(t)}]^{3/2} [R_2^{(t)}]^{3/2} \Big)$$
$$+ (\Sigma_{2,1}^{(t)} + \Sigma_{2,2}^{(t)}) \Big( -\Theta((E_{2,1}^{(t)})^2)[R_1^{(t)}]^3 \pm O(E_{2,1}^{(t)})(\overline{R}_{1,2}^{(t)} + \varrho)[R_1^{(t)}]^{3/2} [R_2^{(t)}]^{3/2} \Big)$$
$$= -\Theta(\Sigma_{1,1}^{(t)} + \Sigma_{2,1}^{(t)} + \Sigma_{2,2}^{(t)})[R_1^{(t)}]^3 \pm O(\sum_{j,\ell} \Sigma_{j,\ell}^{(t)} E_{j,3-j}^{(t)} (\overline{R}_{1,2}^{(t)} + \varrho)[R_1^{(t)}]^{3/2} [R_2^{(t)}]^{3/2})$$
$$\text{(Since } |E_{1,2}^{(t)}| \leq d^{-\Omega(1)} \text{ by Induction E.1)}$$

**Proof of (b):** For Lemma D.5b, we can use the same analysis for $\Sigma_{1,1}^{(t)}$ above and Claim B.1d,e to get (again we have used $\Sigma_{1,2}^{(t)} = \widetilde{O}(E_{1,2}^{(t)}/d^{3/2}) \Sigma_{1,1}^{(t)}$)

$$\langle -\nabla_{w_1} L(W^{(t)}, E^{(t)}), \Pi_{V^\perp} w_2^{(t)} \rangle$$
$$= (1 \pm \widetilde{O}(E_{1,2}^{(t)}/d^{3/2})) \Sigma_{1,1}^{(t)} \Big( (-\Theta(\overline{R}_{1,2}^{(t)}) \pm O(\varrho))[R_1^{(t)}]^{5/2} [R_2^{(t)}]^{1/2} + E_{1,2}^{(t)} R_1^{(t)} [R_2^{(t)}]^2 \Big)$$
$$+ \Theta(\Sigma_{2,1}^{(t)} + \Sigma_{2,2}^{(t)}) \Big( (-\Theta(\overline{R}_{1,2}^{(t)}) + O(\varrho))(E_{2,1}^{(t)})^2 [R_1^{(t)}]^{5/2} [R_2^{(t)}]^{1/2} + E_{2,1}^{(t)} R_1^{(t)} [R_2^{(t)}]^2 \Big)$$
$$= \Big( \Sigma_{1,1}^{(t)} + \sum_{\ell \in [2]} \Sigma_{2,\ell}^{(t)} (E_{2,1}^{(t)})^2 \Big) ((-\Theta(\overline{R}_{1,2}^{(t)}) + O(\varrho))[R_1^{(t)}]^{5/2} [R_2^{(t)}]^{1/2})$$
$$+ O\Big( \sum_{(j,\ell) \neq (2,1)} \Sigma_{j,\ell}^{(t)} E_{j,3-j}^{(t)} R_1^{(t)} [R_2^{(t)}]^2 \Big)$$

**Proof of (c):** Similarly to the proof of (a), we can also expand as follows

$$\langle -\nabla_{w_2} L(W^{(t)}, E^{(t)}), \Pi_{V^\perp} w_2^{(t)} \rangle$$
$$= (1 \pm \widetilde{O}(E_{1,2}^{(t)}/d^{3/2})) \Sigma_{1,1}^{(t)} \Big( -[R_2^{(t)}]^3 \Theta((E_{1,2}^{(t)})^2) \pm O(E_{1,2}^{(t)})(\overline{R}_{1,2}^{(t)} + \varrho)[R_1^{(t)}]^{3/2}[R_2^{(t)}]^{3/2} \Big)$$
$$- \sum_{\ell \in [2]} \Sigma_{2,\ell}^{(t)} \Big( [R_2^{(t)}]^3 \pm O(E_{2,1}^{(t)})(\overline{R}_{1,2}^{(t)} + \varrho)[R_1^{(t)}]^{3/2}[R_2^{(t)}]^{3/2} \Big)$$
$$= -\Theta([R_2^{(t)}]^3)\Big(\Sigma_{1,1}^{(t)}\Theta((E_{1,2}^{(t)})^2) + \sum_{\ell \in [2]} \Sigma_{2,\ell}^{(t)}\Big) \pm O\Big(\sum_{j,\ell} \Sigma_{j,\ell}^{(t)} E_{j,3-j}^{(t)}(\overline{R}_{1,2}^{(t)} + \varrho)[R_1^{(t)}]^{3/2}[R_2^{(t)}]^{3/2}\Big)$$

**Proof of (d):** Similarly, we can calculate

$$\langle -\nabla_{w_2} L(W^{(t)}, E^{(t)}), \Pi_{V^\perp} w_1^{(t)} \rangle$$
$$= (1 \pm \widetilde{O}(E_{1,2}^{(t)}/d^{3/2})) \Sigma_{1,1}^{(t)} \Big( (-\Theta(\overline{R}_{1,2}^{(t)}) \pm O(\varrho))(E_{1,2}^{(t)})^2 [R_2^{(t)}]^{5/2}[R_1^{(t)}]^{1/2} + E_{1,2}^{(t)} R_2^{(t)}[R_1^{(t)}]^2 \Big)$$
$$+ \sum_{\ell \in [2]} \Sigma_{2,\ell}^{(t)} \Big( (-\Theta(\overline{R}_{1,2}^{(t)}) \pm O(\varrho))[R_2^{(t)}]^{5/2}[R_1^{(t)}]^{1/2} + E_{1,2}^{(t)} R_2^{(t)}[R_1^{(t)}]^2 \Big)$$
$$= \Big( \Sigma_{1,1}^{(t)} \Theta((E_{1,2}^{(t)})^2) + \sum_{\ell \in [2]} \Sigma_{2,\ell}^{(t)} \Big)(-\Theta(\overline{R}_{1,2}^{(t)}) \pm O(\varrho))[R_2^{(t)}]^{5/2}[R_1^{(t)}]^{1/2}$$
$$+ O\Big( \sum_{(j,\ell) \neq (2,1)} \Sigma_{j,\ell}^{(t)} E_{j,3-j}^{(t)} R_2^{(t)}[R_1^{(t)}]^2 \Big)$$

which completes the proof. $\qquad\square$

**Lemma E.7** (learning the prediction head in phase III). *If Induction E.1 holds at iteration $t \in [T_2, T_3]$, then using the notations from Fact E.2, we have*

$$-\nabla_{E_{j,3-j}} L(W^{(t)}, E^{(t)}) = \Theta(\sum_{\ell \in [2]} \Sigma_{j,\ell}^{(t)})(-E_{j,3-j}^{(t)}[R_{3-j}^{(t)}]^3 \pm O(\overline{R}_{1,2}^{(t)} + \varrho)[R_1^{(t)}]^{3/2}[R_2^{(t)}]^{3/2})$$
$$- \Xi_j^{(t)} E_{j,3-j}^{(t)} + \sum_{\ell \in [2]} \Delta_{j,\ell}^{(t)}$$

*Proof.* By Fact E.2, we only need to bound the last term $\sum_{\ell \in [2]} \Sigma_{j,\ell}^{(t)} \nabla_{E_{1,2}} \mathcal{E}_{j,3-j}^{(t)}$, which can be directly obtained from applying Claim B.1. $\qquad\square$

### E.3 At the End of Phase III

In order to argue that $B_{2,2}^{(T_2)} = \Omega(1)$ at the end of phase III, we need to define some auxiliary notions. Recall that $T_3$ is defined in (E.1), and now we further define

$$T_{3,1} := \min\{t : C_1 \alpha_2^6 (B_{2,2}^{(t)})^6 \geq C_2 [R_2^{(t)}]^3\}, \qquad T_{3,2}^{(t)} = \min\big\{t : |B_{2,2}^{(t)}| \geq \frac{1}{3}\min\{|E_{2,1}^{(t)}|, |B_{1,1}^{(t)}|\}\big\} \tag{E.2}$$

It can be observed that if Induction E.1 holds for $t \in [T_2, T_3]$ and our learning rate $\eta$ is small enough, we shall have $T_2 < T_{3,1} \leq T_{3,2} < T_3$. Now we are ready to present the main lemma we want to prove in this phase.

**Lemma E.8** (Phase III). *Let $T_3$ be defined as in (E.1). Suppose $\eta = \frac{1}{\text{poly}(d)}$ is sufficiently small, then Induction E.1 holds for all iteration $t \in [T_2, T_3]$, and at iteration $t = T_3$, the followings holds:*

*(a)* $|B_{1,1}^{(T_3)}| = \Theta(1)$, $|B_{2,2}^{(T_3)}| = \Theta(1)$, $B_{j,\ell}^{(T_3)} = B_{j,\ell}^{(T_2)}(1 \pm o(1))$ *for $j \neq \ell$;*

*(b)* $R_1^{(T_3)} = \widetilde{O}(\frac{1}{d^{3/4}})$, $R_2^{(T_3)} \in [\widetilde{O}(\frac{1}{d^{1/2}}), \widetilde{O}(\frac{1}{d^{1/4}})]$, *and* $\overline{R}_{1,2}^{(T_3)} \leq \widetilde{O}(\varrho + \frac{1}{\sqrt{d}})$;

*(c)* $|E_{2,1}^{(T_2)}| = \Theta(\sqrt{\eta_E/\eta})$ *and* $|E_{1,2}^{(T_2)}| = \widetilde{O}(\varrho + \frac{1}{\sqrt{d}})[R_1^{(t)}]^{3/2}[R_2^{(t)}]^{3/2} = \widetilde{O}(\frac{1}{d})$.

*Moreover,* $|B_{2,2}^{(t)}|$ *is increasing and* $R_2^{(t)}$ *is decreasing. The part of learning* $|B_{2,2}^{(t)}|$ *till* $\Omega(1)$ *and keeping* $B_{2,1}^{(t)}$ *close to its initialization is what's been accelerated by the prediction head* $E_{2,1}^{(t)}$.

The proof of Lemma E.8 will be proven after we have proven Induction E.1, which will again be proven after some intermediate results are proven.

**Lemma E.9** (The growth of $B_{2,2}^{(t)}$ before $T_{3,1}$). *Let $T_{3,1}$ be defined as in* (E.2). *If Induction E.1 holds for $t \in [T_2, T_{3,1}]$, then we have $R_2^{(T_{3,1})} \leq \frac{\alpha_1^{12}}{d^{1/4}}$ and $B_{2,2}^{(T_{3,1})} \in [\frac{1}{d^{1/4}}, O(\frac{\alpha_1^{O(1)}}{d^{1/4}})]$ and $T_{3,1} \leq T_2 + \widetilde{O}(\frac{d^{1.625}\alpha_1^{O(1)}}{\eta})$.*

*Proof.* Firstly by Lemma E.6b , we can write down the update of $R_2^{(t)}$: (as in Lemma D.8)

$$R_2^{(t+1)} = R_2^{(t)} - \eta\Theta([R_2^{(t)}]^3)\Big(\Sigma_{1,1}^{(t)}\Theta((E_{1,2}^{(t)})^2) + \sum_{\ell\in[2]}\Sigma_{2,\ell}^{(t)}\Big)$$
$$\pm O\Big(\sum_{j,\ell}\eta\Sigma_{j,\ell}^{(t)}E_{j,3-j}^{(t)}(\overline{R}_{1,2}^{(t)} + \varrho)[R_1^{(t)}]^{3/2}[R_2^{(t)}]^{3/2}\Big) \pm \frac{\eta}{\mathsf{poly}(d)}$$

Next, by Claim B.1 and Lemma E.3a combined with Induction E.1a,b, we have $\widetilde{O}(\frac{|E_{2,1}^{(t)}|}{d^{3/2}})\Sigma_{1,1}^{(t)}\frac{\Phi_1^{(t)}}{\Phi_2^{(t)}} \leq \widetilde{O}(\Sigma_{2,1}^{(t)})$, which leads to the bound

$$\eta\Sigma_{1,1}^{(t)}\Theta((E_{1,2}^{(t)})^2) \leq \widetilde{O}(\varrho^2 + \frac{1}{d})\alpha_1^{O(1)}\eta\Sigma_{1,1}^{(t)}[R_1^{(t)}]^3[R_2^{(t)}]^3 \leq O(\frac{1}{d^{9/4}})\eta\Sigma_{1,1}^{(t)}[R_2^{(t)}]^3 \leq O(\frac{\alpha_1^{O(1)}}{d^{3/4}})\eta\Sigma_{2,1}^{(t)}[R_2^{(t)}]^3$$

Similarly, we can bound the following term

$$\sum_{\ell\in[2]}\eta\Sigma_{1,\ell}^{(t)}|E_{1,2}^{(t)}|(|\overline{R}_{1,2}^{(t)}| + \varrho)[R_1^{(t)}]^{3/2}[R_2^{(t)}]^{3/2} \leq \widetilde{O}(\varrho^2 + \frac{1}{d})\alpha_1^{O(1)}\sum_{\ell\in[2]}\eta\Sigma_{1,\ell}^{(t)}[R_1^{(t)}]^3[R_2^{(t)}]^3$$
$$\leq \widetilde{O}(\varrho^2 + \frac{1}{d})\alpha_1^{O(1)}\frac{1}{d^{9/4}}\sum_{\ell\in[2]}\eta\Sigma_{1,\ell}^{(t)}[R_2^{(t)}]^3$$
$$\leq \widetilde{O}(\frac{\alpha_1^{O(1)}}{d^{3/4}})\eta\Sigma_{2,1}^{(t)}[R_2^{(t)}]^3$$

Moreover, from Induction E.1c that $R_2^{(t)} \geq R_1^{(t)}$, we can also calculate for each $t \in [T_2, T_{3,1}]$:

$$\eta\Sigma_{2,\ell}^{(s)}|E_{2,1}^{(t)}|(|\overline{R}_{1,2}^{(t)}| + \varrho)[R_1^{(t)}]^{3/2}[R_2^{(t)}]^{3/2} \leq \widetilde{O}(\varrho + \frac{1}{\sqrt{d}})\alpha_1^{O(1)}\eta\Sigma_{2,\ell}^{(t)}[R_2^{(t)}]^3$$

Thus by combining the results above, we have the update of $R_2^{(t)}$ at $t \in [T_2, T_3]$ as follows:

$$R_2^{(t+1)} = R_2^{(t)} - \eta\Theta([R_2^{(t)}]^3)\Big(\Sigma_{1,1}^{(t)}\Theta((E_{1,2}^{(t)})^2) + \sum_{\ell\in[2]}\Sigma_{2,\ell}^{(t)}\Big)$$
$$= R_2^{(t)} - \eta(\Sigma_{2,1}^{(t)} + \Sigma_{2,2}^{(t)})[R_2^{(t)}]^3 \tag{E.3}$$

which implies that $R_2^{(t)}$ is decreasing throughout phase III. From Lemma E.3a and Induction E.1b, we know that for $t \in [T_2, T_{3,1}]$:

$$\Phi_2^{(t)} = Q_2^{(t)}/[U_2^{(t)}]^{3/2} = \Theta\Big(\frac{1}{\sqrt{C_2[R_2^{(t)}]^3(C_1\alpha_1^6(E_{2,1}^{(t)})^2)^{3/2}}}\Big)$$

which implies (also using a bit of Claim B.1 and Induction E.1a)

$$\Sigma_{2,1}^{(t)}[R_2^{(t)}]^3 = (1 \pm \widetilde{O}(\frac{1}{d^{3/2}}))E_{2,1}^{(t)}\Delta_{2,1}^{(t)}$$

$$= (1 \pm \widetilde{O}(\frac{1}{d^{3/2}}))(1 \pm \widetilde{O}(\frac{1}{d^{3/2}}))C_0 C_2 \alpha_1^6 \Phi_2^{(t)} E_{2,1}^{(t)}(B_{1,1}^{(t)})^3(B_{2,1}^{(t)})^3[R_2^{(t)}]^3$$

$$= \Theta(\frac{C_2^{1/2}[R_2^{(t)}]^{3/2}}{(U_2^{(t)})^{3/2}})C_0 \alpha_1^6 E_{2,1}^{(t)}(B_{1,1}^{(t)})^3(B_{2,1}^{(t)})^3$$

$$= \Theta(\frac{C_0 C_2^{1/2}|B_{2,1}^{(T_2)}|^3}{C_1^{3/2}\alpha_1^3|E_{2,1}^{(T_2)}|})[R_2^{(t)}]^{3/2}$$

(because $B_{2,1}^{(t)} = B_{2,1}^{(T_2)}(1 \pm o(1))$, $B_{1,1}^{(t)} = \Theta(B_{1,1}^{(T_2)})$ and $E_{2,1}^{(t)} = \Theta(E_{2,1}^{(T_2)})\text{sign}(B_{1,1}^{(T_2)}B_{2,1}^{(T_2)})$)

And for $\Sigma_{2,2}^{(t)}$, from some simple calcualtions (using Claim B.1), we have

- when $|B_{2,2}^{(t)}| \leq \frac{\alpha_1}{\alpha_2}\sqrt{|B_{2,1}^{(T_2)}|}$, we would have $\Sigma_{2,2}^{(t)} \leq O(\Sigma_{2,1}^{(t)})$;

- otherwise, we have $\Sigma_{2,1}^{(t)} + \Sigma_{2,2}^{(t)} = \Theta(\Sigma_{2,2}^{(t)})$.

So by (E.3), we know $R_2$ is decreasing for $t \in [T_2, T_{3,1}]$ by at least

$$R_2^{(t+1)} \leq R_2^{(t)} - \eta\Theta(\frac{C_0 C_2^{1/2}|B_{2,1}^{(T_2)}|^3}{C_1^{3/2}\alpha_1^3|E_{2,1}^{(T_2)}|})[R_2^{(t)}]^{3/2} \leq R_2^{(t)}(1 - \eta\zeta[R_2^{(t)}]^{1/2}) \qquad \text{(E.4)}$$

where $\zeta := \Theta(\frac{C_0 C_2^{1/2}|B_{2,1}^{(T_2)}|^3}{C_1^{3/2}\alpha_1^3|E_{2,1}^{(T_2)}|}) = \widetilde{\Theta}(\frac{\sqrt{\eta/\eta_E}}{d^{3/2}\alpha_1^3})$. By this update, we can prove $T_{3,1} \leq T_2 + O(\frac{d^{3/2+1/8}\alpha_1^{O(1)}}{\eta})$. In order to do that, we can first see that for some $t_{3,1}' \in [T_2 + \widetilde{\Theta}(\frac{d^{3/2}\alpha_1^2\sqrt{\eta_E/\eta}}{\eta}), T_2 + \widetilde{\Theta}(\frac{d^{3/2}\alpha_1^4\sqrt{\eta_E/\eta}}{\eta})]$, we shall have $R_2^{(t_{3,1}')} \leq d^{-1/4}$. Indeed, suppose otherwise $R_2^{(t_{3,1}'-1)} \geq d^{-1/4}$, then (E.4) implies

$$R_2^{(t_{3,1}')} \leq R_2^{(t_{3,1}'-1)}(1 - \eta\zeta[R_2^{(t_{3,1}'-1)}]^{1/2}) \leq R_2^{(t_{3,1}'-1)}(1 - \eta\zeta\frac{1}{d^{1/8}})$$

$$\leq R_2^{(T_2)}\left(1 - \Theta(\frac{C_0 C_2^{1/2}\sqrt{\eta/\eta_E}}{C_1^{3/2}d^{3/2}\alpha_1^3})\frac{\eta}{d^{1/8}}\right)^{t_{3,1}'-T_2-1}$$

$$\leq O(\sqrt{\eta_E/\eta})\left(1 - \Theta(\frac{C_0 C_2^{1/2}\sqrt{\eta/\eta_E}}{C_1^{3/2}d^{3/2}\alpha_1^3})\frac{\eta}{d^{1/8}}\right)^{t_{3,1}'-T_2-1}$$

which means there must exist an iteration $t_{3,1}' \in [T_2 + \widetilde{\Theta}(\frac{d^{3/2}\alpha_1^2\sqrt{\eta_E/\eta}}{\eta}), T_2 + \widetilde{\Theta}(\frac{d^{3/2}\alpha_1^4\sqrt{\eta_E/\eta}}{\eta})]$ such that $R_2^{(t_{3,1}'-1)} \geq d^{-1/4}$ (so the above update bound is still valid when the RHS is for $t \leq t_{3,1}' - 1$) and $R_2^{(t_{3,1}')} < d^{-1/4}$. Next we need to prove that at $t = t_{3,1}'$, it holds $C_1\alpha_2^6(B_{2,2}^{(t)})^6 \geq C_2[R_2^{(t)}]^3$. Let us discuss several possible cases:

1. Suppose $|B_{2,2}^{(t_{3,1}')}| \geq \frac{\alpha_1}{\alpha_2}|B_{2,1}^{(T_1)}|^{1/2} \geq \Theta(\frac{1}{d^{1/4}})$ (by Induction E.1a and Lemma E.8), then we already have $C_1\alpha_2^6(B_{2,2}^{(t_{3,1}')})^6 \geq C_2[R_2^{(t_{3,1}')}]^3$ and $T_{3,1} \leq t_{3,1}'$;

2. Suppose otherwise $|B_{2,2}^{(t_{3,1}')}| \leq \frac{\alpha_1}{\alpha_2}|B_{2,1}^{(T_1)}|^{1/2}$, then we shall have $\Sigma_{2,2}^{(t)} \leq O(\Sigma_{2,1}^{(t)})$. So the update of $R_2^{(t)}$ during $t \in [T_2, T_{3,1}]$ can be written as

$$R_2^{(t+1)} = R_2^{(t)} - \Theta(\eta\Sigma_{2,1}^{(t)})[R_2^{(t)}]^3 = R_2^{(t)}(1 - \Theta(\eta\zeta)[R_2^{(t)}]^{1/2})$$

Let $t'_{3,2} = \min\{t : R_2^{(t)} \leq 2d^{-1/4}\}$ be an iteration between $T_2$ and $t'_{3,1}$, we shall have

$$\sum_{t \in [t'_{3,2}, t'_{3,1}]} \eta\zeta[R_2^{(t)}]^{3/2} = \Theta(R_2^{(t'_{3,2})} - R_2^{(t'_{3,1})}) = \Theta(\frac{1}{d^{1/4}}) \quad \text{and} \quad R_2^{(t)} \in [0.99\frac{1}{d^{1/4}}, 2.01\frac{1}{d^{1/4}}]$$

which also implies $t'_{3,1} - t'_{3,2} = \Theta(\frac{d^{1/8}}{\eta\zeta}) = \widetilde{\Theta}(\frac{d^{3/2+1/8}\alpha_1^3\sqrt{\eta_E/\eta}}{\eta})$. In this case, let us look at the update of $B_{2,2}^{(t)}$ at $t \in [T_2, T_3]$. By Lemma E.42, we have

$$B_{2,2}^{(t+1)} = B_{2,2}^{(t)} + \eta(1 \pm \widetilde{O}(\frac{1}{d}))\Lambda_{2,2}^{(t)}$$

It is not hard to see $|B_{2,2}^{(t)}|$ is monotonically increasing. Also by Induction E.1a and Lemma E.3a, if we sum together the update between $t'_{3,2}$ and $t'_{3,1}$ as follows: (suppose the sign of $B_{2,2}^{(t'_{3,2})}$ is positive for now, the negative case can be similarly dealt with)

$$
\begin{aligned}
B_{2,2}^{(t'_{3,2})} + \sum_{t \in [t'_{3,2}, t'_{3,1}]} \eta(1 \pm \widetilde{O}(\frac{1}{d}))\Lambda_{2,2}^{(t)} &= \sum_{t \in [t'_{3,2}, t'_{3,1}]} \Theta(\frac{\eta C_0 C_1 \alpha_1^6 \alpha_2^6 (E_{2,1}^{(T_2)})^2}{\sqrt{C_2[R_2^{(t)}]^3}(C_1\alpha_1^6(E_{2,1}^{(T_2)})^2)^{3/2}})(B_{2,2}^{(t)})^5 \\
&\geq B_{2,2}^{(t'_{3,2})} + (B_{2,2}^{(T_2)})^4 \sum_{t \in [t'_{3,2}, t'_{3,1}]} \Theta(\frac{\eta C_0 \alpha_1^3 \alpha_2^6 B_{2,2}^{(t)}}{C_1^{1/2}C_2^{1/2}[R_2^{(t)}]^{3/2}|E_{2,1}^{(T_2)}|}) \\
&\geq B_{2,2}^{(t'_{3,2})} \prod_{t=t'_{3,2}}^{t'_{3,1}} \left(1 + \eta\widetilde{\Theta}(\frac{\alpha_1^3\alpha_2^6}{d^{3/2+1/8}\sqrt{\eta_E/\eta}})\right) \\
&\geq \widetilde{\Theta}(\frac{1}{\sqrt{d}})\left(1 + \eta\widetilde{\Theta}(\frac{\alpha_1^3\alpha_2^6}{d^{3/2+1/8}\sqrt{\eta_E/\eta}})\right)^{\widetilde{\Theta}(d^{3/2}\alpha_1^3\sqrt{\eta_E/\eta}/\eta)} \\
&\geq \Omega(e^{\alpha_1})
\end{aligned}
$$

which is a contradiction to our assumption $|B_{2,2}^{(t'_{3,1})}| \leq \frac{\alpha_1}{\alpha_2}|B_{2,1}^{(T_1)}|^{1/2}$. Since $|B_{2,2}^{(t)}|$ is monotonically increasing, we know there must exist some iteration $t \leq t'_{3,1}$ such that $|B_{2,2}^{(t)}| \geq \frac{\alpha_1}{\alpha_2}|B_{2,1}^{(T_1)}|^{1/2}$, which means $T_{3,1} \leq t'_{3,1}$.

Thus we proved the bound of $T_{3,1} \leq T_2 + \widetilde{\Theta}(\frac{d^{3/2}\alpha_1^{O(1)}}{\eta})$.

Using similar arguments, we can prove that $R_2^{(T_{3,1})} \leq \frac{\alpha_1^{O(1)}}{d^{1/4}}$. Indeed, we can set $T_{3,3} := \min\{t : |B_{2,2}^{(t'_{3,1})}| \geq \frac{\alpha_1}{\alpha_2}|B_{2,1}^{(T_1)}|^{1/2}\}$. From our arguments in this proof, we know $\Sigma_{2,2}^{(t)} \leq O(\Sigma_{2,2}^{(t)})$ for $t \leq T_{3,3}$. Now we can further choose $t'_{3,3} = \min\{t : R_2^{(t)} \leq a\}$ for some $a = \frac{\alpha_1^{12}}{d^{1/4}}$ to be some iteration with $R_2^{(t)} \geq a$ for $t \in [T_2, t'_{3,3}]$ and $t'_{3,3} - T_2 = \Theta(\frac{\sqrt{a}\log d}{\eta\zeta})$. Now we can work out the update of $B_{2,2}^{(t)}$ during $t \in [T_2, t'_{3,3}]$ again to see that $|B_{2,2}^{(t'_{3,3})}| \leq B_{2,2}^{(T_2)}\left(1 + \eta\widetilde{\Theta}(\frac{\alpha_1^3\alpha_2^6}{d^2 a^{3/2}\sqrt{\eta_E/\eta}})\right)^{\frac{\sqrt{a}}{\eta\zeta}} \leq \widetilde{O}(\frac{1}{\sqrt{d}})$. This would prove that $t'_{3,3} \leq T_{3,3}$ and $R_2^{(T_{3,3})} \leq \frac{\alpha_1^{O(1)}}{d^{1/4}}$. So we also have $|B_{2,2}^{(T_{3,3})}| \leq \frac{\alpha_1^{O(1)}}{d^{1/4}}$ because of the definition of $T_{3,1}$. But since $T_{3,3} \geq T_{3,1}$ by our arguments above and the fact that $|B_{2,2}^{(t)}|$ is increasing, we shall have $|B_{2,2}^{(T_{3,1})}| \in [\frac{1}{d^{1/4}}, \frac{\alpha_1^{O(1)}}{d^{1/4}}]$. $\qquad\square$

Now we proceed to characterize the learning of $B_{2,2}^{(t)}$ during $t \in [T_{3,1}, T_{3,2}]$.

**Lemma E.10** (The growth of $B_{2,2}^{(t)}$ until $T_3$). *Let $T_{3,1}, T_{3,2}$ be defined as in* (E.2). *If Induction E.1 holds true for all $t \in [T_2, T_3]$, then we have $T_{3,2} = T_{3,1} + \widetilde{O}(\frac{d^{1/4}\alpha_1^{O(1)}}{\eta})$ and $T_3 \leq T_{3,2} + \widetilde{O}(\frac{\alpha_1^{O(1)}}{\eta})$.*

*Proof.* We first calculate the bound for $T_{3,2}$. After $T_{3,1}$, since $|B_{2,2}^{(t)}|$ is increasing while $R_2^{(t)}$ is decreasing by Induction E.1. So by Lemma E.3a, we have

$$[Q_2^{(t)}]^{-2} = \Theta(C_1 \alpha_2^6 (B_{2,2}^{(t)})^6), \quad \Phi_2^{(t)} = Q_2^{(t)}/[U_2^{(t)}]^{3/2} = \Theta((C_1^{3/2} \alpha_2^3 \alpha_1^9 |B_{2,2}^{(t)}|)^3 |E_{2,1}^{(t)}|^3)^{-1})$$

So according to Lemma E.4, we would have for all $t \in [T_{3,1}, T_{3,2}]$:

$$\langle -\nabla_{w_2} L(W^{(t)}, E^{(t)}), v_2 \rangle = (1 \pm o(1))\Lambda_{2,2}^{(t)} = \Theta(\frac{1}{C_1^{3/2}\alpha_1^9 |E_{2,1}^{(T_2)}|^3})(B_{2,2}^{(t)})^2 \mathrm{sign}(B_{2,2}^{(t)})$$

where we have used $(E_{2,1}^{(t)})^3 = \Theta((E_{2,1}^{(T_2)})^3)$ from Induction E.1a. So when $t \in [T_{3,1}, T_{3,2}]$, we can write down the explicit form of $\Lambda_{2,2}^{(t)}$ and use Lemma E.3d to derive

$$|B_{2,2}^{(t+1)}| = |B_{2,2}^{(t)}| + \eta\Theta(\frac{C_1 \alpha_1^6 |E_{2,1}^{(T_2)}|^2}{C_1^{3/2}\alpha_1^9 |E_{2,1}^{(T_2)}|^3})(B_{2,2}^{(t)})^2$$

$$\geq |B_{2,2}^{(t)}| \left(1 + \Theta(\frac{1}{C_1 \alpha_1^{O(1)}})|B_{2,2}^{(T_{3,1})}|\right)$$

$$\geq |B_{2,2}^{(t)}| \left(1 + \Theta(\frac{1}{C_1 \alpha_1^{O(1)}})\frac{1}{d^{1/4}}\right)$$

Thus after $\widetilde{O}(\frac{d^{1/4}\alpha^{O(1)}}{\eta})$ many iterations, we would have $|B_{2,2}^{(t)}| \geq \frac{1}{3}\min\{|E_{2,1}^{(t)}|, |B_{1,1}^{(t)}|\}$. Now let us deal with the growth of $|B_{2,2}^{(t)}|$ at $t \in [T_{3,2}, T_{3,3}]$. During this stage, since $B_{2,2}^{(t)}$ is still increasing and $|E_{2,1}^{(t)}| = |E_{2,1}^{(T_2)}|$ by Induction E.1, we have from Lemma E.3a that

$$\Phi_2^{(t)} = Q_2^{(t)}/[U_2^{(t)}]^{3/2} = \Theta(\frac{1}{C_1^2 \alpha_2^{12}(B_{2,2}^{(t)})^{12}}) \geq \Theta(\frac{1}{C_1^2 \alpha_1^{O(1)}})$$

And we can redo the calcualtions as above to get $T_3 \leq T_{3,2} + \widetilde{O}(\frac{\alpha_1^{O(1)}}{\eta})$ since $\sqrt{\eta/\eta_E}|E_{2,1}^{(t)}|$ and $|B_{1,1}^{(t)}|$ are both $\Theta(1)$ according to Induction E.1a,b $\qquad\square$

**Proving The Main Lemma.** Now we finally begin to prove Lemma E.8.

*Proof of Lemma E.8.* We start with proving Induction E.1.

**Proof of Induction E.1a:** From Lemma E.5, we know the update of $B_{1,1}^{(t)}$ can be written as

$$B_{1,1}^{(t+1)} = B_{1,1}^{(t)} + \Theta(\eta\Sigma_{1,1}^{(t)}[R_1^{(t)}]^3) \pm \eta O(\frac{(B_{1,2}^{(t)})^3}{(B_{2,2}^{(t)})^3} + \frac{1}{\sqrt{d}})\alpha_1^{O(1)}\Lambda_{2,2}^{(t)} + \frac{E_{2,1}^{(t)}}{B_{1,1}^{(t)}}\eta\Delta_{2,1}^{(t)} - \frac{B_{2,2}^{(t)}}{B_{1,1}^{(t)}}\eta\Lambda_{2,2}^{(t)}$$

Since from Lemma E.9 and Lemma E.10, we know $T_3 \leq \widetilde{O}(\frac{d^{1.625}\alpha_1^{O(1)}}{\eta})$ and from Claim B.1 and Induction E.1a,c we have $\Sigma_{1,1}^{(t)}[R_1^{(t)}]^3 \leq \widetilde{O}(\frac{\alpha_1^{O(1)}}{d^{2.25}})$, we shall have

$$\sum_{s\in[T_2,t]} \Theta(\eta\Sigma_{1,1}^{(s)}[R_1^{(s)}]^3) \leq \widetilde{O}(\frac{d^{1.625}\alpha_1^{O(1)}}{\eta})\widetilde{O}(\frac{\eta\alpha_1^{O(1)}}{d^{2.25}}) \leq \frac{1}{\sqrt{d}} = o(1)$$

Further more, by applying Lemma H.3 to $x_t = B_{2,2}^{(t)}$ with $q' = q - 2$, and notice that $\mathrm{sign}(B_{j,2}^{(t)}) = \mathrm{sign}(B_{j,2}^{(T_2)})$ for all $t \in [T_2, T_3]$, we also have

$$\left|\sum_{s\in[T_2,t]} O(\frac{(B_{1,2}^{(s)})^3}{(B_{2,2}^{(s)})^3})\alpha_1^{O(1)}\eta\Lambda_{2,2}^{(s)}\right| \leq \widetilde{O}(\frac{\alpha_1^{O(1)}}{\sqrt{d}})$$

Now we turn to the last two terms. We first see that from the expression (E.3) of $R_2^{(t)}$'s update, we have that (note that $\text{sign}(E_{2,1}^{(t)}\Delta_{2,1}^{(t)}) = 1$)

$$\sum_{s\in[T_2,t)}\frac{E_{2,1}^{(s)}}{|B_{1,1}^{(s)}|}\eta\Delta_{2,1}^{(s)} = \sum_{s\in[T_2,t)}\frac{1}{|B_{1,1}^{(s)}|}\Theta(\eta\Sigma_{2,1}^{(s)}[R_2^{(s)}]^3) = \Theta(\frac{\sqrt{\eta_E/\eta}}{|B_{1,1}^{(T_2)}|}) = \Theta(\sqrt{\eta_E/\eta})$$

where we have used the fact that $\Sigma_{2,1}^{(t)}[R_2^{(t)}]^3 = (1\pm O(\frac{1}{d}))E_{2,1}^{(t)}\Delta_{2,1}^{(t)}$ and $\sum_{s\in[T_2,t)}\eta\Sigma_{2,1}^{(s)}[R_2^{(s)}]^3 \lesssim R_2^{(T_2)}$ from (E.3) (which holds for all $t\in[T_2,T_3]$). And also, the analysis above shows that

$$|B_{1,1}^{(t)}| = |B_{1,1}^{(T_2)}| + O(\sqrt{\eta_E/\eta}) - \sum_{s\in[T_2,t]}\frac{B_{2,2}^{(s)}}{B_{1,1}^{(s)}}\eta\Lambda_{2,2}^{(s)}$$

for all $t\in[T_2,T_3]$, which means that either $\sum_{s\in[T_2,t]}\frac{B_{2,2}^{(s)}}{|B_{1,1}^{(s)}|}\eta\Lambda_{2,2}^{(s)} \leq \sum_{s\in[T_2,t]}\frac{E_{2,1}^{(s)}}{|B_{1,1}^{(s)}|}\eta\Delta_{2,1}^{(s)}$ and we have $|B_{1,1}^{(t)}| \geq |B_{1,1}^{(T_2)}|$ holds throughout $t\in[T_2,T_3]$, or that $\sum_{s\in[T_2,t]}\frac{B_{2,2}^{(s)}}{|B_{1,1}^{(s)}|}\eta\Lambda_{2,2}^{(s)} \geq \Omega(\sqrt{\eta_E/\eta})$, in which case we would have $|B_{1,1}^{(t)}|$ to be actually decreasing (as $B_{2,2}^{(t)}$ is increasing). Now that since $B_{1,1}^{(T_2)} = \Theta(1)$, we can easily see by our definition of $T_3$ and the monotonicity of $B_{1,1}^{(t)}$ after going below $B_{1,1}^{(T_2)} - \Omega(\sqrt{\eta_E/\eta})$ that $B_{1,1}^{(t)} \geq 0.49B_{1,1}^{(T_2)} = \Omega(1)$ for all $t\in[T_2,T_3]$.

Next let us look at the change of $B_{2,1}^{(t)}$. From Lemma E.5, we can write down the update of $B_{2,1}^{(t)}$:

$$B_{2,1}^{(t+1)} = B_{2,1}^{(t)} + \widetilde{O}(\frac{\alpha_1^{O(1)}}{d^{5/2}})\eta\Phi_2^{(t)}[R_2^{(t)}]^3 \pm \widetilde{O}(\frac{\alpha_1^{O(1)}}{d})\eta\Lambda_{2,2}^{(t)} \pm \widetilde{O}(\frac{\eta\alpha_1^{O(1)}}{d^3})$$

For the first term, according to Lemma E.9 and Lemma E.10 and $R_2^{(t)} \leq O(\sqrt{\eta_E/\eta}) = o(1)$ for all $t\in[T_2,T_3]$ by Induction E.1c, we have $\Phi_2^{(t)}[R_2^{(t)}]^3 \leq \alpha_1^{O(1)}$ for all $t\in[T_2,T_3]$ and

$$\sum_{s\in[T_2,t]}\widetilde{O}(\frac{\alpha_1^{O(1)}}{d^{5/2}})\eta\Phi_2^{(s)}[R_2^{(s)}]^3 \leq \widetilde{O}(\frac{d^{1.625}\alpha_1^{O(1)}}{\eta})\eta\widetilde{O}(\frac{\alpha_1^{O(1)}}{d^{5/2}}) \leq \widetilde{O}(\frac{\alpha_1^{O(1)}}{d^{7/8}})$$

And similarly as in the proof of induction for $B_{1,1}^{(t)}$, we have

$$\sum_{s\in[T_2,t]}\widetilde{O}(\frac{\alpha_1^{O(1)}}{d})\eta\Lambda_{2,2}^{(s)} \leq \widetilde{O}(\frac{\alpha_1^{O(1)}}{d}), \quad \sum_{s\in[T_2,t]}\widetilde{O}(\frac{\eta\alpha_1^{O(1)}}{d^3}) \leq \widetilde{O}(\frac{\alpha_1^{O(1)}}{d})$$

which proved the induction for $B_{2,1}^{(t)}$ since $|B_{2,1}^{(T_2)}| = \widetilde{\Theta}(\frac{1}{\sqrt{d}})$.

Next we go on for the induction of $B_{1,2}^{(t)}$, we write down its update:

$$B_{1,2}^{(t+1)} = B_{1,2}^{(t)} + \Theta(\frac{(B_{1,2}^{(t)})^2}{(B_{2,2}^{(t)})^2})E_{2,1}^{(t)}\eta\Lambda_{2,2}^{(t)} \pm \eta\widetilde{O}(\frac{\alpha_1^{O(1)}}{d^4})|E_{2,1}^{(t)}|^2\Phi_2^{(t)} \pm \eta\widetilde{O}(\frac{\alpha_1^{O(1)}}{d^{5/2}})$$

By Lemma E.9 and Lemma E.10, we have for any $t\in[T_2,T_3]$

$$\sum_{s\in[T_2,t]}\eta\widetilde{O}(\frac{\alpha_1^{O(1)}}{d^{5/2}}) \leq \frac{1}{\sqrt{d}\mathsf{polylog}(d)}$$

and also

$$\sum_{s\in[T_2,t]}\eta\widetilde{O}(\frac{\alpha_1^{O(1)}}{d^4})|E_{2,1}^{(t)}|^2\Phi_2^{(t)} \leq \left(\sum_{s\in[T_2,T_{3,1}]} + \sum_{s\in[T_{3,1},T_3]}\right)\eta\widetilde{O}(\frac{\alpha_1^{O(1)}}{d^4})|E_{2,1}^{(t)}|^2\Phi_2^{(t)}$$

$$\leq \eta\widetilde{O}(\frac{\alpha_1^{O(1)}}{d^4})\cdot(T_{3,1} - T_2)\cdot O(\alpha_1^{O(1)}d^{3/8}) + \eta\widetilde{O}(\frac{\alpha_1^{O(1)}}{d^4})(T_3 - T_{3,1})$$

$$\leq \widetilde{O}(\frac{\alpha_1^{O(1)}}{d^2})$$

Now we consider the term $\Theta(\frac{(B_{1,2}^{(t)})^2}{(B_{2,2}^{(t)})^2})E_{2,1}^{(t)}\eta\Lambda_{2,2}^{(t)}$, we have by Induction E.1a that

$$\left|\sum_{s\in[T_2,t]}\Theta(\frac{(B_{1,2}^{(t)})^2}{(B_{2,2}^{(t)})^2})E_{2,1}^{(t)}\eta\Lambda_{2,2}^{(t)}\right| \leq O(\sqrt{\eta_E/\eta}(B_{1,2}^{(T_2)})^2)\sum_{s\in[T_2,t]}\eta\frac{|\Lambda_{2,2}^{(t)}|}{(B_{2,2}^{(t)})^2}$$

where we have used our induction hypothesis that $B_{1,2}^{(t)} = B_{1,2}^{(T_2)}(1\pm o(1))$. Using Lemma H.3 by setting $x_t = B_{2,2}^{(t)}$, $q' = 3$, and $A = \Theta(1) \geq d^{\Omega(1)}B_{2,2}^{(T_2)}$, it holds that

$$\left|\sum_{s\in[T_2,t]}\Theta(\frac{(B_{1,2}^{(t)})^2}{(B_{2,2}^{(t)})^2})E_{2,1}^{(t)}\eta\Lambda_{2,2}^{(t)}\right| \leq O(\sqrt{\eta_E/\eta})\frac{(B_{1,2}^{(T_2)})^2}{|B_{2,2}^{(T_2)}|} \leq O(\sqrt{\eta_E/\eta})\frac{(B_{1,2}^{(0)})^2}{|B_{2,2}^{(0)}|} \leq \frac{1}{\sqrt{d}\mathsf{polylog}(d)}$$

where in the second inequality we have used Lemma C.13c, Lemma D.8a and Lemma C.1, and in the last our choice of $\eta_E/\eta \leq \frac{1}{\mathsf{polylog}(d)}$. This ensures the induction can go on until $t = T_3$. And we finished our proof of Induction E.1a.

**Proof of Induction E.1b:** Let us write down the update of $E_{1,2}^{(t)}$ using Lemma E.7:

$$E_{1,2}^{(t+1)} = E_{1,2}^{(t)}(1 - \eta_E\Xi_1^{(t)}) + \sum_{\ell\in[2]}\Theta(\eta_E\Sigma_{1,\ell}^{(t)})(-E_{1,2}^{(t)}[R_2^{(t)}]^3 \pm O(\overline{R}_{1,2}^{(t)} + \varrho)[R_1^{(t)}]^{3/2}[R_2^{(t)}]^{3/2}) + \sum_{\ell\in[2]}\eta_E\Delta_{1,\ell}^{(t)}$$

$$= E_{1,2}^{(t)}(1 - \eta_E\Xi_1^{(t)} - \sum_{\ell\in[2]}\Theta(\eta_E\Sigma_{1,\ell}^{(t)})[R_2^{(t)}]^3) + \widetilde{O}(\frac{\eta_E}{d^{3/2}})\Phi_1^{(t)}[R_1^{(t)}]^3$$

$$\pm \widetilde{O}(\varrho + \frac{1}{\sqrt{d}})\sum_{\ell\in[2]}\eta_E\Sigma_{1,\ell}^{(t)}[R_1^{(t)}]^{3/2}[R_2^{(t)}]^{3/2}$$

$$= E_{1,2}^{(t)}(1 - \eta_E\Xi_1^{(t)} - \Theta(\eta_E\Sigma_{1,1}^{(t)})[R_2^{(t)}]^3) \pm \widetilde{O}(\varrho + \frac{1}{\sqrt{d}})\eta_E\Sigma_{1,1}^{(t)}[R_1^{(t)}]^{3/2}[R_2^{(t)}]^{3/2}$$

where in the last inequality we have used $R_2^{(t)} \geq R_1^{(t)}$ from Induction E.1c and $\Sigma_{1,1}^{(t)} \geq \Omega(\Phi_1^{(t)})$, $\Sigma_{2,1}^{(t)} \leq \widetilde{O}(\frac{1}{d^{3/2}})\Sigma_{1,1}^{(t)}$ from Claim B.1 and Induction E.1a. Now we can use the same analysis in the proof of Lemma D.8 on $E_{1,2}^{(t)}$ to prove the desired claim, which we do not repeat here.

As for $E_{2,1}^{(t)}$, we can obtain similar expressions:

$$E_{2,1}^{(t+1)} = E_{2,1}^{(t)}(1 - \eta_E\Xi_2^{(t)} - \sum_{\ell\in[2]}\Theta(\eta_E\Sigma_{2,\ell}^{(t)})[R_1^{(t)}]^3)$$

$$\pm \widetilde{O}(\varrho + \frac{1}{\sqrt{d}})\sum_{\ell\in[2]}\Theta(\eta_E\Sigma_{2,\ell}^{(t)})[R_1^{(t)}]^{3/2}[R_2^{(t)}]^{3/2} + \sum_{\ell\in[2]}\eta_E\Delta_{2,\ell}^{(t)}$$

Now we can obtain bounds for each terms as

$$\sum_{s\in[T_2,t]}\sum_{\ell\in[2]}\Theta(\eta_E\Sigma_{2,\ell}^{(s)})[R_1^{(s)}]^3 \leq \widetilde{O}(\frac{\eta_E\alpha_1^{O(1)}}{d^2})\cdot\widetilde{O}(\frac{d^{1.625}\alpha_1^{O(1)}}{\eta}) \leq \frac{1}{d^{3/4}}$$

and by (E.3) in Lemma E.9, we also have for any $t\in[T_2,T_3]$

$$\sum_{s\in[T_2,t]}\widetilde{O}(\varrho + \frac{1}{\sqrt{d}})\sum_{\ell\in[2]}\Theta(\eta_E\Sigma_{2,\ell}^{(s)})[R_1^{(s)}]^{3/2}[R_2^{(s)}]^{3/2} \leq \widetilde{O}(\varrho + \frac{1}{\sqrt{d}})\sum_{s\in[T_2,t]}\sum_{\ell\in[2]}\Theta(\eta_E\Sigma_{2,\ell}^{(s)})[R_2^{(s)}]^3$$

$$\leq \widetilde{O}(\varrho + \frac{1}{\sqrt{d}})R_2^{(T_2)}$$

$$\leq \widetilde{O}(\varrho + \frac{1}{\sqrt{d}})$$

And also by using our induction and by (E.3) in Lemma E.9:

$$\sum_{s\in[T_2,t]}\sum_{\ell\in[2]}\eta_E\Delta_{2,\ell}^{(s)} \leq \sum_{s\in[T_2,t]}\frac{\eta_E/\eta}{|E_{2,1}^{(t)}|}\Theta(\eta\Sigma_{2,1}^{(s)} + \eta\Sigma_{2,2}^{(s)})[R_2^{(s)}]^3 \leq \frac{\eta_E/\eta}{|E_{2,1}^{(T_2)}|}R_2^{(T_2)} \leq O(\frac{\eta_E/\eta}{\log d}) = o(\sqrt{\eta_E/\eta})$$

Finally, we can calculate

$$\sum_{s\in[T_2,t]} \eta_E \Xi_2^{(t)} E_{2,1}^{(t)} = \sum_{s\in[T_2,t]} \frac{\eta_E}{\eta}\frac{B_{2,2}^{(t)}}{E_{2,1}^{(t)}}\eta\Lambda_{2,2}^{(t)}$$

By resorting to the defintion of $T_3$ and go through similar analysis as for the induction of $B_{1,1}^{(t)}$, we can obtain that $|E_{2,1}^{(t)}|$ is either above $|E_{2,1}^{(T_2)}|(1+o(1))$ or is decreasing and always above $\frac{1}{2}|E_{2,1}^{(T_2)}|$. This proves Induction E.1b.

**Proof of Induction E.1c:** The proof of induction of $R_2^{(t)}$ is half done in Lemma E.9, we only need to complete the part when $t \in [T_{3,1}, T_3]$, since by (E.3), we always have $R_2^{(t)}$ to be decreasing by

$$R_2^{(t+1)} = R_2^{(t)}(1 - \sum_{\ell\in[2]} \Theta(\eta\Sigma_{2,\ell}^{(s)})[R_2^{(t)}]^2)$$

And when $t \in [T_{3,1}, T_3]$, we have

$$\sum_{\ell\in[2]} \Theta(\eta\Sigma_{2,\ell}^{(s)} \leq \widetilde{O}(\eta d^{3/8+o(1)})$$

So if we suppose $R_2^{(T_3)} \leq \frac{1}{\sqrt{d}}$, we shall have for $T_3 - T_{3,1} = O(d^{1/4+o(1)}/\eta)$ many iterations that

$$R_2^{(t+1)} \geq R_2^{(T_{3,1})}(1 - \frac{\eta}{d^{5/8}})^{T_3-T_{3,1}} \geq \Omega(R_2^{(T_{3,1})}) \geq \frac{1}{d^{1/4}} \qquad \text{(by Lemma E.9)}$$

So it negates our supposition, which completes the proof of the induction for $R_2^{(t)}$ in $t \in [T_2, T_3]$.

Now we turn to the proof of induction for $R_1^{(t)}$, we write down its update: (as in Lemma D.8)

$$R_1^{(t+1)} = R_1^{(t)} - \Theta(\eta[R_1^{(t)}]^3)\Big(\Sigma_{1,1}^{(t)} + \sum_{\ell\in[2]}\Sigma_{2,\ell}^{(t)}(E_{2,1}^{(t)})^2\Big)$$

$$\pm O\Big(\sum_{j,\ell}\eta\Sigma_{j,\ell}^{(t)}E_{j,3-j}^{(t)}(\overline{R}_{1,2}^{(t)} + \varrho)[R_1^{(t)}]^{3/2}[R_2^{(t)}]^{3/2}\Big) \pm \frac{\eta}{\mathsf{poly}(d)}$$

It is straightforward to derive

$$\sum_{\ell\in[2]}\Sigma_{1,\ell}^{(t)}|E_{1,2}^{(t)}|(\overline{R}_{1,2}^{(t)} + \varrho)[R_1^{(t)}]^{3/2}[R_2^{(t)}]^{3/2} \leq \widetilde{O}(\varrho + \frac{1}{\sqrt{d}})^2\sum_{\ell\in[2]}\Sigma_{1,\ell}^{(t)}[R_1^{(t)}]^3[R_2^{(t)}]^3$$

and when $t \in [T_2, T_{3,1}]$:

$$\sum_{s\in[T_2,t]}\sum_{\ell\in[2]}\eta\Sigma_{2,\ell}^{(s)}|E_{2,1}^{(s)}|(\overline{R}_{1,2}^{(s)} + \varrho)[R_1^{(s)}]^{3/2}[R_2^{(s)}]^{3/2} \leq \widetilde{O}(\varrho + \frac{1}{\sqrt{d}})\frac{d^{o(1)}d^{3/8}}{d^{9/8}}\sum_{s\in[T_2,t]}\sum_{\ell\in[2]}\eta\Sigma_{2,\ell}^{(s)}[R_2^{(s)}]^3$$

$$\leq o(\frac{d^{o(1)}}{d^{3/4}})$$

and when $t \in [T_{3,1}, T_3]$:

$$\sum_{s\in[T_2,t]}\sum_{\ell\in[2]}\eta\Sigma_{2,\ell}^{(s)}|E_{2,1}^{(s)}|(\overline{R}_{1,2}^{(s)} + \varrho)[R_1^{(s)}]^{3/2}[R_2^{(s)}]^{3/2} \leq \widetilde{O}(\varrho + \frac{1}{\sqrt{d}})\frac{d^{o(1)}d^{3/8}}{d^{9/8}}\eta\widetilde{O}(\frac{d^{1/4+o(1)}}{\eta}) \leq O(\frac{1}{d})$$

So these combined with Lemma D.8 proved that $R_1^{(t)} \leq O(\frac{d^{o(1)}}{d^{3/4}})$ for all $t \in [T_2, T_3]$. We can go through some similar analysis about $R_2^{(t)}$ to get that $R_1^{(t)} \geq \frac{1}{d}$ for all $t \in [T_2, T_3]$.

Finally we begin to prove the induction of $\overline{R}_{1,2}^{(t)}$. Similarly as in the proof of Lemma D.8, we first write down

$$
\begin{aligned}
R_{1,2}^{(t+1)} &= R_{1,2}^{(t)} - \eta\langle\nabla_{w_1}L(W^{(t)}, E^{(t)}), \Pi_{V^\perp}w_2^{(t)}\rangle - \eta\langle\nabla_{w_2}L(W^{(t)}, E^{(t)}), \Pi_{V^\perp}w_1^{(t)}\rangle \\
&\quad + \eta^2\langle\Pi_{V^\perp}\nabla_{w_1}L(W^{(t)}, E^{(t)}), \Pi_{V^\perp}\nabla_{w_2}L(W^{(t)}, E^{(t)})\rangle \\
&= R_{1,2}^{(t)} + \eta\Big(\Sigma_{1,1}^{(t)} + \sum_{\ell\in[2]}\Sigma_{2,\ell}^{(t)}(E_{2,1}^{(t)})^2\Big)(-\Theta(\overline{R}_{1,2}^{(t)}) \pm O(\varrho))[R_1^{(t)}]^{5/2}[R_2^{(t)}]^{1/2} \\
&\quad + \eta\Big(\Sigma_{1,1}^{(t)}\Theta(E_{1,2}^{(t)})^2 + \sum_{\ell\in[2]}\Sigma_{2,\ell}^{(t)}\Big)(-\Theta(\overline{R}_{1,2}^{(t)}) \pm O(\varrho))[R_2^{(t)}]^{5/2}[R_1^{(t)}]^{1/2} \\
&\quad + O\Big(\sum_{(j,\ell)\neq(1,2)}\eta\Sigma_{j,\ell}^{(t)}E_{j,3-j}^{(t)}(R_1^{(t)}[R_2^{(t)}]^2 + R_2^{(t)}[R_1^{(t)}]^2)\Big) \pm \frac{\eta}{\mathsf{poly}(d)}
\end{aligned}
$$

Note that since $|E_{1,2}^{(t)}| \leq \widetilde{O}(\varrho + \frac{1}{\sqrt{d}})[R_2^{(t)}]^{3/2}[R_1^{(t)}]^{3/2}$ and $R_1^{(t)} \leq O(\frac{1}{d^{3/4}})$, it holds

$$
\sum_{(j,\ell)\neq(1,2)}\eta\Sigma_{j,\ell}^{(t)}|E_{j,3-j}^{(t)}|R_2^{(t)}[R_1^{(t)}]^2 \leq \sum_{(j,\ell)\neq(1,2)}\eta\Sigma_{j,\ell}^{(t)}|E_{j,3-j}^{(t)}|R_1^{(t)}[R_2^{(t)}]^2
$$

$$
\leq o\left(\Sigma_{1,1}^{(t)}[R_1^{(t)}]^2 + \sum_{\ell\in[2]}\Sigma_{2,\ell}^{(t)}[R_2^{(t)}]^2\right)\widetilde{O}(\varrho + \frac{1}{\sqrt{d}})[R_2^{(t)}]^{1/2}[R_1^{(t)}]^{1/2}
$$

so the update becomes

$$
\begin{aligned}
R_{1,2}^{(t+1)} &= R_{1,2}^{(t)}\left(1 - \eta\Theta\Big(\Sigma_{1,1}^{(t)} + \sum_{\ell\in[2]}\Sigma_{2,\ell}^{(t)}(E_{2,1}^{(t)})^2\Big)[R_1^{(t)}]^2 - \eta\Theta\Big(\Sigma_{1,1}^{(t)}(E_{1,2}^{(t)})^2 + \sum_{\ell\in[2]}\Sigma_{2,\ell}^{(t)}\Big)[R_2^{(t)}]^2\right) \\
&\quad \pm \eta\widetilde{O}(\varrho + \frac{1}{\sqrt{d}})[R_1^{(t)}]^{1/2}[R_2^{(t)}]^{1/2}\Theta\Big(\Sigma_{1,1}^{(t)} + \sum_{\ell\in[2]}\Sigma_{2,\ell}^{(t)}(E_{2,1}^{(t)})^2\Big)[R_1^{(t)}]^2 \\
&\quad \pm \eta\widetilde{O}(\varrho + \frac{1}{\sqrt{d}})[R_1^{(t)}]^{1/2}[R_2^{(t)}]^{1/2}\Theta\Big(\Sigma_{1,1}^{(t)}(E_{1,2}^{(t)})^2 + \sum_{\ell\in[2]}\Sigma_{2,\ell}^{(t)}\Big)[R_2^{(t)}]^2
\end{aligned}
$$

Now we can use the same arguments as in the proof of $\overline{R}_{1,2}^{(t)}$ in Lemma D.8 to conclude.

**Proof of Lemma E.8a,b,c:** Indeed, at the end of phase III:

$$
\begin{array}{lll}
\text{Induction E.1a} & \implies & \text{Lemma E.8a} \\
\text{Induction E.1b} & \implies & \text{Lemma E.8c} \\
\text{Induction E.1c} & \implies & \text{Lemma E.8b}
\end{array}
$$

Now we have completed the whole proof. $\qquad\square$

# F The End Phase: Convergence

When we arrive at $t = T_3$, we have already obtained the representation we want for the encoder network $f(X)$, where $v_1$ and $v_2$ are satisfactorily learned by different neurons. In the last phase, we prove that such features are the solutions that the algorithm are converging to, which gives a stronger guarantee than just accidentally finding the solution at some intermediate steps.

To prove the convergence, we need to ensure all the good properties that we got through the training still holds. Fortunately, mosts of Induction E.1 still hold, as we summarized below:

**Inductions F.1.** *At the end phase, i.e. when $t \in [T_3, T]$, Induction E.1a continues to hold except that $|B_{2,2}^{(t)}| = \Theta(1)$, Induction E.1b will hold except that for $|E_{2,1}^{(t)}|$ only the upper bound still holds, and the upper bounds in Induction E.1c still hold while the lower bounds for $R_1^{(t)}, R_2^{(t)}$ is $1/\mathsf{poly}(d)$. Moreover, there is a constant $C = O(1)$ such that when $t \geq T_3 + \frac{\alpha_1^C}{\eta}$, we would have $|E_{2,1}^{(t)}| \leq \widetilde{O}(\varrho + \frac{1}{\sqrt{d}})[R_1^{(t)}]^{3/2}[R_2^{(t)}]^{3/2}$.*

Now we present the main theorem of the paper, which we shall prove in this section.

**Theorem F.2** (End phase: convergence). *For some $T_4 = T_3 + \frac{d^{2+o(1)}}{\eta}$ and $T = \mathsf{poly}(d)/\eta$, we have for all $t \in [T_4, T]$ that Induction F.1 holds true and:*

*(a) Successful learning of both $v_1, v_2$: $|B_{1,1}^{(t)}|, |B_{2,2}^{(t)}| = \Theta(1)$ while $|B_{2,1}^{(t)}|, |B_{1,2}^{(t)}| = \widetilde{O}(\frac{1}{\sqrt{d}})$.*

*(b) Successful denoising at the end: $R_j^{(t)} \leq R_j^{(T_3)}(1 - \widetilde{\Theta}(\frac{1}{\alpha_j^6})[R_j^{(t)}]^2)$ for all $j \in [2]$.*

*(c) Prediction head is close to identity: $|E_{j,3-j}^{(t)}| \leq \widetilde{O}(\varrho + \frac{1}{\sqrt{d}})[R_1^{(t)}]^{3/2}[R_1^{(t)}]^{3/2}$ for all $j \in [2]$;*

*In fact, (b) and (c) also imply for some sufficiently large $t = \mathsf{poly}(d)/\eta$, it holds $R_j^{(t)} \leq \frac{1}{\mathsf{poly}(d)}$ and $|E_{j,3-j}^{(t)}| \leq \frac{1}{\mathsf{poly}(d)}$ for all $j \in [2]$.*

And we have a simple corollary for the objective convergence.

**Corollary F.3** (objective convergence, with prediction head). *Let $\mathsf{OPT}$ denote the global minimum of the population objective* (B.1). *It is easy to derive that $\mathsf{OPT} = 2 - 2\frac{C_0}{C_1} = \Theta(\frac{1}{\log d})$. We have for some sufficiently large $t \geq \mathsf{poly}(d)/\eta$:*

$$L(W^{(t)}, E^{(t)}) \leq \mathsf{OPT} + \frac{1}{\mathsf{poly}(d)}$$

Now we need to establish some auxiliary lemmas:

**Lemma F.4.** *For some $t \in [T_3, \mathsf{poly}(d)/\eta]$, if Induction F.1 holds from $T_3$ to $t$, we have Lemma E.6 holds at $t$.*

*Proof.* Simple from similar calculations in the proof of Lemma E.6 . $\qquad\square$

**Lemma F.5.** *For some $t \in [T_3, \mathsf{poly}(d)/\eta]$, if Induction F.1 holds from $T_3$ to $t$, we have for each $j \in [2]$ that*

$$\sum_{s \in [T_3, t]} \sum_{\ell \in [2]} \eta \Sigma_{j,\ell}^{(s)} [R_j^{(s)}]^3 \leq O(R_j^{(T_3)}), \quad \forall j \in [2]$$

*Proof.* Notice that when Induction F.1 holds, we always have

$$\sum_{\ell \in [2]} (\Sigma_{j,\ell}^{(t)} + \Sigma_{3-j,\ell}^{(t)}(E_{3-j,j}^{(t)})^2) = (1 \pm o(1)) \sum_{\ell \in [2]} \Sigma_{j,\ell}^{(t)}$$

we can use Lemma F.4 to obtain the update of $R_2^{(t)}$ as in the calculations when we obtained (E.3):

$$R_2^{(t)} = R_2^{(T_3)} - \sum_{s \in [T_3, t)} \sum_{\ell \in [2]} \Theta(\eta \Sigma_{2,\ell}^{(s)})[R_2^{(s)}]^3$$

which means that $R_2^{(t)}$ is decreasing from $T_3$ to $t$. Summing up the update, the part of $R_2^{(t)}$ is solved. For the part of $R_1^{(t)}$, we separately discuss when $|E_{2,1}^{(t)}|$ is larger than or smaller than $\widetilde{O}(\varrho + \frac{1}{\sqrt{d}})[R_1^{(t)}]^{3/2}[R_2^{(t)}]^{3/2}$. When the former happens, which we know from Induction F.1 that it cannot last until some $t_4' = T_3 + \frac{\alpha_1^{O(1)}}{\eta}$ many iterations, we have for $t \in [T_3, t_4']$

$$\sum_{s \in [T_3, t)} \sum_{(j,\ell) \in [2]^2} \eta \Sigma_{j,\ell}^{(s)} |E_{j,3-j}^{(s)}| (\overline{R}_{1,2}^{(s)} + \varrho)[R_1^{(s)}]^{3/2}[R_2^{(s)}]^{3/2} \leq \widetilde{O}(\varrho + \frac{1}{\sqrt{d}}) \frac{\alpha_1^{O(1)}}{d} R_1^{(T_3)} \leq \frac{1}{d} R_1^{(T_3)}$$

Now for $t \geq t_4'$ we can simply go through similar calculations as in the proof of Induction E.1c to obtain

$$\sum_{s\in[t_4',t)}\sum_{(j,\ell)\in[2]^2}\eta\Sigma_{j,\ell}^{(s)}|E_{j,3-j}^{(s)}|(\overline{R}_{1,2}^{(s)}+\varrho)[R_1^{(s)}]^{3/2}[R_2^{(s)}]^{3/2} \leq \sum_{s\in[t_4',t)}\widetilde{O}(\varrho+\frac{1}{\sqrt{d}})^2\sum_{(j,\ell)\in[2]^2}\eta\Sigma_{j,\ell}^{(s)}[R_1^{(s)}]^3[R_2^{(s)}]^3$$

$$\leq \widetilde{O}(\varrho+\frac{1}{\sqrt{d}})^2 R_2^{(T_3)}\max_{s\in[t_4',t)}[R_1^{(s)}]^3$$

$$\leq \frac{1}{d}R_1^{(T_3)}$$

So by applying Lemma F.4a and Lemma E.6, we have

$$R_1^{(t)} = (1\pm o(1))R_1^{(T_3)} - \sum_{s\in[T_3,t)}\sum_{\ell\in[2]}\Theta(\eta\Sigma_{j,\ell}^{(s)})[R_1^{(s)}]^3$$

which proves the claim. $\qquad\qquad\square$

**Lemma F.6.** *For some $t \in [T_3, \mathrm{poly}(d)/\eta]$, if Induction F.1 holds from $T_3$ to $t$. Then we have $|E_{j,3-j}^{(t)}|$ is decreasing until $|E_{j,3-j}^{(t)}| \leq O(\overline{R}_{1,2}^{(t)}+\varrho)[R_1^{(t)}]^{3/2}[R_2^{(t)}]^{3/2} + \widetilde{O}(\frac{1}{d^{3/2}})[R_j^{(t)}]^3$. Moreover, we have for each $t\in[T_3,T]$ that*

$$\left|\sum_{s\in[T_3,t]}\eta_E\Xi_j^{(t)}E_{j,3-j}^{(s)}\right| \leq |E_{j,3-j}^{(T_3)}| + \widetilde{O}(\varrho+\frac{1}{\sqrt{d}}) \leq O(\sqrt{\eta_E/\eta})$$

*Proof.* We can go through the same calculations in the proof of Induction E.1b (using Fact E.2) to obtain

$$E_{j,3-j}^{(t+1)} = E_{j,3-j}^{(t)}(1-\eta_E\Xi_j^{(t)}) + \sum_{\ell\in[2]}\eta_E\Delta_{j,\ell}^{(t)}$$

$$+ \sum_{\ell\in[2]}\Theta(\eta_E\Sigma_{j,\ell}^{(t)})(-E_{j,3-j}^{(t)}[R_{3-j}^{(t)}]^3 \pm O(\overline{R}_{1,2}^{(t)}+\varrho)[R_1^{(t)}]^{3/2}[R_2^{(t)}]^{3/2})$$

$$= E_{j,3-j}^{(t)}(1-\eta_E\Xi_j^{(t)}-\eta_E\Theta(\Sigma_{j,j}^{(t)}[R_{3-j}^{(t)}]^3)) + \widetilde{O}(\frac{1}{d^{3/2}})\sum_{\ell\in[2]}\eta_E\Sigma_{j,\ell}^{(t)}[R_j^{(t)}]^3$$

$$\pm O(\eta_E\Sigma_{j,j}^{(t)})(\overline{R}_{1,2}^{(t)}+\varrho)[R_1^{(t)}]^{3/2}[R_2^{(t)}]^{3/2}$$

where we have used in the second equality that $\sum_{\ell\in[2]}\Delta_{j,\ell}^{(t)} \leq \widetilde{O}(\frac{1}{d^{3/2}})\sum_{\ell\in[2]}\Sigma_{j,\ell}^{(t)}[R_j^{(t)}]^3$ and also $\Sigma_{j,3-j}^{(t)} \leq O(\frac{1}{d^{3/2}})\Sigma_{j,j}^{(t)}$ for both $j\in[2]$ when Induction F.1 holds. Note that from above calculations, there exist a constant C such that if $|E_{j,3-j}^{(t)}| \geq C(\overline{R}_{1,2}^{(t)}+\varrho)[R_1^{(t)}]^{3/2}[R_2^{(t)}]^{3/2} + \sum_{\ell\in[2]}\eta_E\Delta_{j,\ell}^{(t)}$, we have $|E_{2,1}^{(t)}|$ to be decreasing. Now it suffices to observe that:

$$\sum_{s\in[T_3,t]}O(\eta_E\Sigma_{j,j}^{(t)})(\overline{R}_{1,2}^{(t)}+\varrho)[R_1^{(t)}]^{3/2}[R_2^{(t)}]^{3/2} \leq \sum_{s\in[T_3,t]}O(\eta_E\Sigma_{1,1}^{(t)}+\eta_E\Sigma_{2,2}^{(t)})(\overline{R}_{1,2}^{(t)}+\varrho)([R_1^{(t)}]^3+[R_2^{(t)}]^3)$$

$$\leq \widetilde{O}(\varrho+\frac{1}{\sqrt{d}})$$

which is from Induction F.1, Induction E.1c and Lemma F.4. Also note that $\Sigma_{j,j}^{(t)}[R_{3-j}^{(t)}]^3 \leq O(\frac{d^{o(1)}}{d^{3/4}})\Xi_j^{(t)}$ at this stage, we have

$$E_{3-j,j}^{(t)} = E_{j,3-j}^{(T_3)} - \sum_{s\in[T_3,t)}\Xi_j^{(s)}E_{j,3-j}^{(s)} + \widetilde{O}(\varrho+\frac{1}{\sqrt{d}})$$

Recalling the expression of $\Xi_j^{(t)}$ finishes the proof. $\qquad\qquad\square$

**Lemma F.7.** *Recall $T_2$ defined in (D.1) and $T_3$ defined in (E.1), we have*

$$\sqrt{\eta/\eta_E} \max_{t \leq T_3} |E_{2,1}^{(t)}| \leq \sum_{t \leq T_2} \frac{\eta \Sigma_{1,1}^{(t)}}{|B_{1,1}^{(t)}|} \mathcal{E}_{1,2}^{(t)} + \frac{1}{\alpha_1^{\Omega(1)}}$$

To prove this lemma, we need a simple claim.

**Claim F.8.** *If $\{x_t\}_{t<T}, x_t \geq 0$ is an increasing sequence and $C = \Theta(1)$ is a constant such that $x_{t+1} - x_t \leq O(\eta)$ and $\sum_{t<T} x_t(x_{t+1} - x_t) = C$, then for each $\delta \in (\frac{1}{d}, 1)$ it holds $|x_T - \sqrt{C}| \leq O(\delta^2 + x_0^2 + O(\frac{\log d}{d}))$.*

*Proof.* Indeed, for every $g \in 0, 1, \ldots$, we define $\mathcal{T}_g := \min\{t : x_t \geq (1+\delta)^g x_0\}$. and define $b := \min\{g : ((1+\delta)^g x_0)^2 \geq C - \delta^2\}$. Now for any $g < b$, we have

$$\sum_{t \in [\mathcal{T}_g, \mathcal{T}_{g+1}]} x_t(x_{t+1} - x_t) \geq x_{\mathcal{T}_g}(x_{\mathcal{T}_{g+1}} - x_{\mathcal{T}_g}) \geq (1+\delta)^g \delta(1+\delta)^{g-1} x_0^2 - \frac{1}{d} = \delta(1+\delta)^{2g-1} x_0^2 - \frac{1}{d}$$

By our definition of $\mathcal{T}_g$, we can further get

$$C = \sum_{t<T} x_t(x_{t+1} - x_t) = \sum_{g=1}^{b} \sum_{t \in [\mathcal{T}_g, \mathcal{T}_{g+1}]} x_t(x_{t+1} - x_t) \geq (1+\delta)^{2b} x_0^2 - x_0^2 - \frac{b}{d} \geq C - \delta^2 - x_0^2 - \frac{b}{d}$$

And also we have $C \leq (\max_{t \leq T} x_t) \sum_{t<T}(x_{t+1} - x_t) = x_T^2$, so we have $|x_T^2 - C| \leq \delta^2 + x_0^2 + \frac{b}{d}$, where $b = O(\log(C)/\log(1+\delta)) \leq O(\log d)$, which proves the claim. $\square$

*Proof of Lemma F.7.* From the proof of Lemma D.8 and Lemma E.8 we know that

$$\max_{t \leq T_3} |E_{2,1}^{(t)}| \leq \sum_{t \leq T_3} (1 \pm \frac{1}{\alpha_1^{\Omega(1)}}) \eta_E |\Delta_{2,1}^{(t)}| + \widetilde{O}(\varrho + \frac{1}{\sqrt{d}})$$

And since from the proof of Lemma D.8 we know that

$$R_2^{(T_3)} = R_2^{(0)} - \sum_{t \leq T_3} (1 \pm \widetilde{O}(\frac{1}{d^{3/2}})) \eta \Sigma_{2,1}^{(t)} \mathcal{E}_{2,1}^{(t)} \pm \widetilde{O}(\varrho + \frac{1}{\sqrt{d}})$$

$$= (1 \pm \widetilde{O}(\frac{1}{d^{3/2}})) \sum_{t \leq T_3} E_{2,1}^{(t)} \Delta_{2,1}^{(t)} \pm \widetilde{O}(\varrho + \frac{1}{\sqrt{d}})$$

We can define some alternative variables $\widetilde{E}_{2,1}^{(t)}$ updated as $\widetilde{E}_{2,1}^{(t+1)} = \widetilde{E}_{2,1}^{(t)} + \eta_E \Delta_{2,1}^{(t)}$ and $\widetilde{R}_2^{(t+1)} = \widetilde{R}_2^{(t)} - \widetilde{E}_{2,1}^{(t)} \Delta_{2,1}^{(t)}$. It is easy to see that $|E_{2,1}^{(t)} - \widetilde{E}_{2,1}^{(t)}| \leq \frac{1}{\alpha_1^{\Omega(1)}} \max_{t \leq T_3} |E_{2,1}^{(t)}|$. From above calculations, we know $\frac{\eta}{\eta_E} \sum_{t \in [T_1, T_3]} \widetilde{E}_{2,1}^{(t)}(\widetilde{E}_{2,1}^{(t+1)} - \widetilde{E}_{2,1}^{(t)}) = \widetilde{R}_2^{(T_1)} \pm \widetilde{O}(\varrho + \frac{1}{\sqrt{d}}) + O(\frac{1}{d^{1/4}})$, which by Claim F.8 implies that

$$\sqrt{\eta/\eta_E} |\widetilde{E}_{2,1}^{(T_3)}| = \sqrt{\widetilde{R}_2^{(T_1)}} \pm O(\frac{1}{d^{1/4}}) = \sqrt{2} \pm \widetilde{O}(\varrho + \frac{1}{\sqrt{d}}) \pm O(\frac{1}{d^{1/4}})$$

And when we turn back, we shall have $\sqrt{\eta/\eta_E} \max_{t \leq T_3} |E_{2,1}^{(t)}| \leq \sqrt{2} + \frac{1}{\alpha_1^{\Omega(1)}}$. Now we can use similar techniques on $B_{1,1}^{(t)}$ and $R_1^{(t)}$. Indeed, from (D.4) and similar arguments in phase I, we know for all $t \in [T_1, T_2]$

$$R_1^{(t+1)} = R_1^{(0)} - \sum_{s \leq t} (1 \pm \widetilde{O}(\frac{1}{d^{3/2}})) \eta \Sigma_{1,1}^{(s)} \mathcal{E}_{1,2}^{(s)} \pm \widetilde{O}(\varrho + \frac{1}{\sqrt{d}}) \tag{F.1}$$

$$R_1^{(t+1)} \leq R_1^{(t)}(1 - \widetilde{O}(\frac{\eta}{\alpha_1^6})[R_1^{(t)}]^2)$$

So one can obtain that at some iteration $t' = T_1 + O(\frac{d\alpha_1^{O(1)}}{\eta})$, we shall have $R_1^{(t)} \leq O(\frac{1}{\sqrt{d}})$ for all $t \geq t'$. Now let us consider the growth of $B_{1,1}^{(t)}$ before $t'$, which clearly constitutes of

$$B_{1,1}^{(t')} = B_{1,1}^{(T_1)} + \sum_{t \in [T_1, t')} (\Lambda_{1,1}^{(t)} + \Gamma_{1,1}^{(t)} - \Upsilon_{1,1}^{(t)})$$

$$= B_{1,1}^{(T_1)} + \sum_{t \in [T_1, t')} \left( \frac{\eta \Sigma_{1,1}^{(t)}}{|B_{1,1}^{(t)}|} \mathcal{E}_{1,2}^{(t)} \mathrm{sign}(B_{1,1}^{(t)}) + \eta \Gamma_{1,1}^{(t)} - \eta \Upsilon_{1,1}^{(t)} \right)$$

$$= B_{1,1}^{(0)} + \sum_{t < t'} \frac{\eta \Sigma_{1,1}^{(t)}}{|B_{1,1}^{(t)}|} \mathcal{E}_{1,2}^{(t)} \mathrm{sign}(B_{1,1}^{(t)}) + \sum_{t \in [T_1, t')} \eta \left( \Gamma_{1,1}^{(t)} - \Upsilon_{1,1}^{(t)} \right) + \widetilde{O}(\frac{1}{\sqrt{d}})$$

where the last one comes from the proof of Lemma C.13. Moreover by using the same arguments in the proof of Lemma D.8 we can easily prove that

$$\left| \sum_{t \in [T_1, t')} (\Gamma_{1,1}^{(t)} - \Upsilon_{1,1}^{(t)}) \right| \leq \widetilde{O}(\frac{1}{\sqrt{d}}) \quad \Longrightarrow \quad \sum_{t < t'} \frac{\eta \Sigma_{1,1}^{(t)}}{|B_{1,1}^{(t)}|} \mathcal{E}_{1,2}^{(t)} \geq |B_{1,1}^{(t')}| - |B_{1,1}^{(0)}| - \widetilde{O}(\frac{1}{\sqrt{d}})$$

And for $t \in [t', T_2]$, we also have by (F.1) that

$$\sum_{t \in [t', T_2]} \frac{\eta \Sigma_{1,1}^{(t)}}{|B_{1,1}^{(t)}|} \mathcal{E}_{1,2}^{(t)} \leq \sum_{t \in [t', T_2)} \eta \Sigma_{1,1}^{(t)} \mathcal{E}_{1,2}^{(t)} \leq O(\frac{1}{\sqrt{d}})$$

Recall $R_1^{(0)} = \sum_{t \in [0, t')} (1 \pm \widetilde{O}(\frac{1}{d^{3/2}})) \eta \Sigma_{1,1}^{(t)} \mathcal{E}_{1,2}^{(t)} \pm \widetilde{O}(\varrho + \frac{1}{\sqrt{d}})$ by (F.1) and $R_1^{(t)} \leq O(\frac{1}{\sqrt{d}})$ for $t \geq t'$. Now we can finally go through the same analysis using Claim F.8 on $B_{1,1}^{(t)}$ and $R_1^{(t)}$ during $t \in [0, t']$ as above to obtain that

$$\sum_{t \leq T_2} \frac{\eta \Sigma_{1,1}^{(t)}}{|B_{1,1}^{(t)}|} \mathcal{E}_{1,2}^{(t)} \geq (1 - \widetilde{O}(\frac{1}{d^{3/2}})) \sqrt{R_1^{(0)}} - \widetilde{O}(\frac{1}{\sqrt{d}}) = 1 - \widetilde{O}(\varrho + \frac{1}{\sqrt{d}})$$

Combining the results, we finishes the proof. $\qquad \square$

Now we are prepared to prove Theorem F.2.

### F.1 Proof of Convergence

*Proof of Theorem F.2.* First we start with the $B_{j,\ell}^{(t)}$s. Indeed, we can go through similar calculations to see that all gradients $\langle -\nabla_{w_j} L(W^{(t)}, E^{(t)}), v_\ell \rangle$ can be decomposed into

$$\langle -\nabla_{w_j} L(W^{(t)}, E^{(t)}), v_\ell \rangle = (\Lambda_{j,\ell}^{(t)} - \Upsilon_{j,\ell,1}^{(t)}) + (\Gamma_{j,\ell}^{(t)} - \Upsilon_{j,\ell,2}^{(t)})$$

where $\Lambda_{j,\ell}^{(t)} - \Upsilon_{j,\ell,1}^{(t)}$ and $\Gamma_{j,\ell}^{(t)} - \Upsilon_{j,\ell,2}^{(t)}$ can be expressed as

$$\Lambda_{j,\ell}^{(t)} - \Upsilon_{j,\ell,1}^{(t)} = C_0 \alpha_2^6 C_1 \alpha_1^6 \Phi_j^{(t)} (B_{j,\ell}^{(t)})^5 \left( E_{j,3-j}^{(t)} (B_{3-j,3-\ell}^{(t)})^3 (B_{j,3-\ell}^{(t)})^3 + (E_{j,3-j}^{(t)})^2 (B_{3-j,3-\ell}^{(t)})^6 \right)$$

$$- C_0 \alpha_2^6 C_1 \alpha_1^6 \Phi_j^{(t)} (B_{j,\ell}^{(t)})^2 (B_{3-j,\ell}^{(t)})^3 E_{j,3-j}^{(t)} \left( (B_{j,3-\ell}^{(t)})^6 + E_{j,3-j}^{(t)} (B_{3-j,3-\ell}^{(t)})^3 (B_{j,3-\ell}^{(t)})^3 \right)$$

$$+ C_0 \alpha_2^6 \Phi_j^{(t)} (B_{j,\ell}^{(t)})^5 C_2 \mathcal{E}_{j,3-j}^{(t)}$$

$$\Gamma_{j,\ell}^{(t)} - \Upsilon_{j,\ell,2}^{(t)} = C_0 \alpha_2^6 C_1 \alpha_1^6 \Phi_{3-j}^{(t)} (B_{3-j,\ell}^{(t)})^3 (B_{j,\ell}^{(t)})^2 E_{3-j,j}^{(t)} \left( E_{3-j,j}^{(t)} (B_{j,3-\ell}^{(t)})^3 (B_{3-j,3-\ell}^{(t)})^3 + (E_{3-j,j}^{(t)})^2 (B_{j,3-\ell}^{(t)})^6 \right)$$

$$- C_0 \alpha_2^6 C_1 \alpha_1^6 \Phi_{3-j}^{(t)} (B_{j,\ell}^{(t)})^5 (E_{3-j,j}^{(t)})^2 \left( (B_{3-j,3-\ell}^{(t)})^6 + E_{3-j,j}^{(t)} (B_{j,3-\ell}^{(t)})^3 (B_{3-j,3-\ell}^{(t)})^3 \right)$$

$$+ C_0 \alpha_2^6 \Phi_{3-j}^{(t)} E_{3-j,j}^{(t)} (B_{3-j,\ell}^{(t)})^3 (B_{j,\ell}^{(t)})^2 C_2 \mathcal{E}_{3-j,j}^{(t)}$$

Firstly, for all the terms that contain factors of $(B_{j,\ell}^{(t)})^2 (B_{3-j,\ell}^{(t)})^2$ (or $(B_{j,\ell}^{(t)})^2 (B_{j,3-\ell}^{(t)})^2$), we can apply Lemma F.6, our Induction F.1 assumption and $|E_{j,3-j}^{(t)}| \leq O(1), \forall t \in [T_3, T]$ to obtain that their

(multiplicated by $\eta$) summation over $t \in [T_3, T]$ is absolutely bounded by $\widetilde{O}(\frac{1}{d})$. So we can move on to deal with all other terms. When $j = \ell$, Using Lemma F.6, we have

$$\sum_{t \in [T_3, T]} \eta C_0 \alpha_2^6 C_1 \alpha_1^6 \Phi_j^{(t)} |B_{j,\ell}^{(t)}|^5 (E_{j,3-j}^{(t)})^2 (B_{3-j,3-\ell}^{(t)})^6 = \sum_{t \in [T_3, T]} \frac{\eta \Xi_j^{(t)}}{|B_{j,\ell}^{(t)}|} (E_{j,3-j}^{(t)})^2$$
$$\leq \sqrt{\frac{\eta}{\eta_E}} |E_{j,3-j}^{(T_3)}| + \widetilde{O}(\varrho + \frac{1}{\sqrt{d}}) = O(1)$$

And the sign of LHS is $\mathrm{sign}(B_{j,\ell}^{(t)})$. Moreover, for $j = \ell = 1$, from Lemma F.7 and Lemma F.6 we also have

$$\sum_{t \in [T_3, T]} \eta C_0 \alpha_2^6 C_1 \alpha_1^6 \Phi_2^{(t)} |B_{1,1}^{(t)}|^5 (E_{2,1}^{(t)})^2 (B_{2,2}^{(t)})^6 \leq \sqrt{\frac{\eta}{\eta_E}} \left| \sum_{t \in [T_3, T]} \eta_E \Xi_j^{(t)} E_{j,3-j}^{(t)} \right|$$
$$\leq \sqrt{\frac{\eta}{\eta_E}} |E_{2,1}^{(T_3)}| + \widetilde{O}(\varrho + \frac{1}{\sqrt{d}})$$
$$\leq \sum_{t \leq T_2} \frac{\eta \Sigma_{1,1}^{(t)}}{|B_{1,1}^{(t)}|} \mathcal{E}_{1,2}^{(t)} + \frac{1}{\alpha_1^{\Omega(1)}}$$

Since we have

$$B_{1,1}^{(T_2)} = \sum_{s \leq T_2} \frac{\eta \Sigma_{1,1}^{(t)}}{|B_{1,1}^{(t)}|} \mathcal{E}_{1,2}^{(t)} + \sum_{s \leq T_2} \frac{\eta \Sigma_{2,1}^{(t)}}{|B_{1,1}^{(t)}|} \mathcal{E}_{2,1}^{(t)} - \sum_{t \in [T_3, T]} \frac{\eta \Xi_j^{(t)}}{|B_{j,\ell}^{(t)}|} (E_{j,3-j}^{(t)})^2$$

And since by Induction D.1 we have $|B_{1,1}^{(t)}| = \Theta(1)$ during $t \in [T_1, T_2]$ and $\sum_{t \in [T_1, T_2]} \eta \Sigma_{2,1}^{(t)} \geq R^{(T_1)} - o(1) = \sqrt{2} - o(1)$. For all the other terms in the gradient , we can apply Lemma F.6, our Induction F.1 assumption and $|E_{j,3-j}^{(t)}| \leq O(1)$ so we have for $t \in [T_3, T]$

$$|B_{1,1}^{(t)}| = \sum_{s \leq T_2} \frac{\eta \Sigma_{1,1}^{(t)}}{|B_{1,1}^{(t)}|} \mathcal{E}_{1,2}^{(t)} + \sum_{s \leq T_2} \frac{\eta \Sigma_{2,1}^{(t)}}{|B_{1,1}^{(t)}|} \mathcal{E}_{2,1}^{(t)} - \sum_{t \in [T_3, T]} \frac{\eta \Xi_j^{(t)}}{|B_{j,\ell}^{(t)}|} (E_{j,3-j}^{(t)})^2 - o(1)$$
$$\geq \sqrt{\eta/\eta_E} \max_{t \leq T_3} |E_{2,1}^{(t)}| + \sum_{s \leq T_2} \frac{\eta \Sigma_{2,1}^{(t)}}{|B_{1,1}^{(t)}|} \mathcal{E}_{2,1}^{(t)} - \sqrt{\frac{\eta}{\eta_E}} |E_{j,3-j}^{(T_3)}| + \widetilde{O}(\varrho + \frac{1}{\sqrt{d}}) - o(1)$$
$$\geq \sum_{s \leq T_2} \frac{\eta \Sigma_{2,1}^{(t)}}{|B_{1,1}^{(t)}|} \mathcal{E}_{2,1}^{(t)} - o(1) \geq \Omega(1)$$

which also proved $|B_{1,1}^{(t)}| = O(1)$ since all the terms on the RHS are absolutely $O(1)$ bounded. Since one can see from Lemma F.6 that $|E_{2,1}^{(t)}|$ is decreasing before it reaches $\frac{1}{d}$). Moreover this proves $\sqrt{\eta/\eta_E} |E_{2,1}^{(t)}| \leq B_{1,1}^{(t)}$ for all $t \in [T_3, T]$, and also the fact that

$$B_{1,1}^{(t)} \geq \Omega(1), \quad \forall t \in [T_3, T]$$

The case of $B_{2,2}^{(t)}$ is much more simple as $E_{1,2}^{(t)} \leq \widetilde{O}(\frac{1}{d})$ throughout $t \in [T_3, T]$ by Lemma F.6 and Lemma E.8c, Now we can go through the similar calculations again to obtain that $B_{2,2}^{(t)} = \Theta(1)$ for all $t \in [T_3, T]$. When $j \neq \ell$, all the terms calculated in the expansion of $\Lambda_{j,\ell}^{(t)} - \Upsilon_{j,\ell,1}^{(t)}$ and $\Gamma_{j,\ell}^{(t)} - \Upsilon_{j,\ell,2}^{(t)}$ contain factors of $(B_{2,1}^{(t)})^2 = \widetilde{O}(\frac{1}{d})$ or $(B_{1,2}^{(t)})^2 = \widetilde{O}(\frac{1}{d})$. So we can similarly use Lemma F.6 as before to derive that $B_{j,3-j}^{(t)} = B_{j,3-j}^{(T_3)}(1 \pm \widetilde{O}(\frac{\alpha_1^{O(1)}}{\sqrt{d}}))$ for all $t \in [T_3, T]$ and $j \in [2]$.

As for the prediction head, the induction of $E_{1,2}^{(t)}$ follows from exactly the same proof in Lemma E.8. The part of $E_{2,1}^{(t)}$ is half done in Lemma F.6. It suffices to notice that $\Xi_2^{(t)} = \widetilde{\Theta}(\frac{\alpha_1^6}{\alpha_2^6})$ and if

$|E_{2,1}^{(t)}| \geq C(\overline{R}_{1,2}^{(t)} + \varrho)[R_1^{(t)}]^{3/2}[R_2^{(t)}]^{3/2}$ for some $C = O(1)$, then

$$E_{2,1}^{(t+1)} = E_{2,1}^{(t)}(1 - \eta_E \Xi_2^{(t)} - \eta_E \Theta(\Sigma_{2,2}^{(t)}[R_1^{(t)}]^3)) + \widetilde{O}(\frac{1}{d^{3/2}}) \sum_{\ell \in [2]} \eta_E \Sigma_{2,\ell}^{(t)}[R_2^{(t)}]^3$$

$$\pm O(\eta_E \Sigma_{2,2}^{(t)})(\overline{R}_{1,2}^{(t)} + \varrho)[R_1^{(t)}]^{3/2}[R_2^{(t)}]^{3/2}$$

$$\leq E_{2,1}^{(t)}(1 - \widetilde{\Theta}(\frac{\eta \alpha_1^6}{\alpha_2^6}))$$

So after $\frac{\alpha_1^{O(1)}}{\eta}$ many epochs will we have

$$|E_{2,1}^{(t)}| \leq (\log d)|\overline{R}_{1,2}^{(t)} + \varrho|[R_1^{(t)}]^{3/2}[R_2^{(t)}]^{3/2} \leq \widetilde{O}(\varrho + \frac{1}{\sqrt{d}})[R_1^{(t)}]^{3/2}[R_2^{(t)}]^{3/2}$$

as desired. And the rest of the induction of $E_{2,1}^{(t)}$ is the same as in the induction arguments of $E_{1,2}^{(t)}$ in Lemma E.8.

The induction of $R_1^{(t)}, R_2^{(t)}$ and $R_{1,2}^{(t)}$ is exactly the same as those in the proof of Lemma E.8 except here we only need $R_1^{(t)}/R_2^{(t)} \in [\frac{1}{\alpha_1^{O(1)}}, \alpha_1^{O(1)}]$ after $T_4$. Indeed, from the update of $R_j^{(t)}$ (which can be easily worked out), we have

$$R_j^{(t+1)} = R_j^{(t)}(1 - \Theta(\eta \Sigma_{j,j}^{(t)})[R_j^{(t)}]^2) = R_j^{(t)}(1 - \widetilde{\Theta}(\frac{\eta}{\alpha_j^6})[R_j^{(t)}]^2)$$

Now after $\frac{d^2 \alpha_1^{O(1)}}{\eta}$ many epochs, we can obtain from similar arguments in Lemma E.8 that $R_1^{(t)}/R_2^{(t)} \in [\frac{1}{\alpha_1^{O(1)}}, \alpha_1^{O(1)}]$ and $R_j^{(t)} \leq \frac{1}{d}$. The induction can go on untill $t = \mathsf{poly}(d)/\eta$.

For the convergence of $B_{1,1}^{(t)}$ and $B_{2,2}^{(t)}$ after $t = T_4$, notice that their change depends on $\sum_{t \geq T_4} \frac{E_{j,3-j}^{(t)}}{B_{j,j}^{(t)}} \Xi_j^{(t)}$, which stays very small after $T_4$, we have that $|B_{j,j}^{(t)} - B_{j,j}^{(T_4)}| \leq o(1)$ for all $j \in [2]$. This finishes the whole proof. $\qquad\square$

## G    Learning Without Prediction Head

When we do not use prediction head in the network architecture, the analysis is much simpler. We can reuse most of the gradient calculations in previous sections as long as we set $E^{(t)}$ to the identity. Note that here we allow $m \geq 1$ to be any positive integer.

**Theorem G.1** (learning without the prediction head). *Let $m$ be any positive integer. If we keep $E^{(t)} \equiv I_m$ during the whole training process, then for all $t \in [\widetilde{\Omega}(\frac{d^2}{\eta}), \mathsf{poly}(d)/\eta]$, we shall have $|B_{j,1}^{(t)}| = \Theta(1)$, $|B_{j,2}^{(t)}| = \widetilde{O}(\frac{1}{\sqrt{d}})$ and $R_j^{(t)} = O(\frac{1}{d^{1-o(1)}})$ for all $j \in [m]$ with probability $1 - o(1)$. Moreover, for a longer training time $t = \mathsf{poly}(d)/\eta$, we would have $R_j^{(t)} \leq \frac{1}{\mathsf{poly}(d)}$ for all $j \in [m]$.*

Moreover, it is direct to obtain a objective convergence result similar to Corollary F.3.

**Corollary G.2** (objective convergence, without prediction head). *Let $\mathsf{OPT}$ denote the global minimum of the population objective* (B.1). *When trained with $E^{(t)} \equiv I_m$, we have for some sufficiently large $t \geq \mathsf{poly}(d)/\eta$:*

$$L(W^{(t)}, I_m) \leq \mathsf{OPT} + \frac{1}{\mathsf{poly}(d)}$$

*Proof of Theorem G.1.* The proof is easy to obtain since it is very similar to some proofs in previous sections, and we only sketch it here. Indeed, using the calculations in Lemma E.5 and Lemma E.4 and set $E_{i,j}^{(t)}, i \neq j \in [m]$ to zero. We shall have (note that here $\mathcal{E}_{j,r}^{(t)} \equiv \mathcal{E}_j^{(t)}$ for any $r \neq j$)

$$\langle -\nabla_{w_j} L(W^{(t)}, E^{(t)}), v_\ell \rangle = C_0 C_2 \alpha_\ell^6 (B_{j,\ell}^{(t)})^5 \Phi_j^{(t)} \mathcal{E}_j^{(t)} = \Theta(C_0 C_2 \alpha_\ell^6 \Phi_j^{(t)} (B_{j,\ell}^{(t)})^5 [R_j^{(t)}]^3)$$

Now we can go through the similar induction arguments as in the proof of Lemma C.13 (with TPM lemma to distinguish the learning speed) to obtain that for each $j \in [m]$:

$$|B_{j,1}^{(t)}| = \Theta(1), \quad |B_{j,2}^{(t)}| = |B_{j,2}^{(0)}|(1 \pm o(1)), \quad \forall j \in [m] \qquad \text{(when } t \geq \tfrac{d^2}{\eta})$$

When this is proven, we can also reuse the calculations as in the proof of Lemma D.5 to obtain that

$$R_j^{(t+1)} = R_j^{(t)}(1 - \Theta(\eta \Sigma_{j,1}^{(t)})[R_j^{(t)}]^2) = R_j^{(t)}(1 - \Theta(\eta C_0 C_2 \alpha_1^6 \Phi_j^{(t)}(B_{j,1}^{(t)})^6 [R_j^{(t)}]^2), \quad \forall j \in [m]$$

So again after some $t = \widetilde{O}(\tfrac{d^2}{\eta})$, we shall have $R_j^{(t)} \leq O(\tfrac{d^{o(1)}}{d})$. While the decrease of $R_j^{(t)}$ is happening, we can make induction that $|B_{j,2}^{(t)}| = |B_{j,2}^{(0)}|(1 \pm o(1))$, since if it holds for all previous iterations before $t$, then

$$\sum_{s \leq t-1} \eta |\langle -\nabla_{w_j} L(W^{(s)}, E^{(s)}), v_2 \rangle| = \sum_{s \leq t-1} \eta C_0 \alpha_2^6 \Phi_j^{(s)} |B_{j,2}^{(s)}|^5 C_2 \mathcal{E}_j^{(s)}$$

$$\overset{\textcircled{1}}{\leq} \frac{1}{\mathsf{polylog}(d)} |B_{j,2}^{(0)}|$$

where $\textcircled{1}$ is due to Corollary H.2, where $x_t = |B_{j,1}^{(t)}|$ and $y_t = |B_{j,2}^{(t)}|$ and $S_t \leq \frac{1}{\mathsf{polylog}(d)}$, $y_0 \leq O(\log d)x_0$. which finishes the proof. $\qquad \square$

## H   Tensor Power Method Bounds

In this section, we give two lemmas related to the tensor power method that can help us in previous sections' proofs.

**Lemma H.1** (TPM, adapted from [3]). *Consider an increasing sequence $x_t \geq 0$ defined by $x_{t+1} = x_t + \eta C_t x_t^q$ for some integer $q \geq 3$ and $C_t > 0$, and suppose for some $A > 0$ there exist $t' \geq 0$ such that $x_{t'} \geq A$. Then for every $\delta > 0$, and every $\eta \in (0,1)$:*

$$\sum_{t \geq 0, x_t \leq A} \eta C_t \geq \left( \frac{\delta(1+\delta)^{-1}}{(1+\delta)^{q-1} - 1} \left( 1 - \left( \frac{(1+\delta)x_0}{A} \right)^{q-1} \right) - \frac{O(\eta A^q)}{x_0} \frac{\log(A/x_0)}{\log(1+\delta)} \right) \cdot \frac{1}{x_0^{q-1}}$$

$$\sum_{t \geq 0, x_t \leq A} \eta C_t \leq \left( \frac{(1+\delta)^{q-1}}{q-1} + \frac{O(\eta A^q)}{x_0} \frac{\log(A/x_0)}{\log(1+\delta)} \right) \cdot \frac{1}{x_0^{q-1}}$$

This lemma has a corollary:

**Corollary H.2** (TPM, from [3]). *Let $q \geq 3$ be a constant and $x_0, y_0 = o(1)$ and $A = O(1)$. Let $\{x_t, y_t\}_{t \geq 0}$ be two positive sequences updated as*

- *$x_{t+1} = x_t + \eta C_t x_t^q$ for some $C_t > 0$;*

- *$y_{t+1} = y_t + \eta S_t C_t y_t^q$ for some $S_t > 0$.*

*Suppose $x_0 \geq y_0 (\max_{t: x_t \leq A} S_t)^{\frac{1}{q-1}} (1 + \frac{1}{\mathsf{polylog}(d)})$, then $y_t \leq \widetilde{O}(y_0)$ for all $t$ such that $x_t \leq A$. Moreover, if $x_0 \geq y_0 (\max_{t: x_t \leq A} S_t)^{\frac{1}{q-1}} \log(d)$, we would have $|y_t - y_0| \lesssim \frac{|y_0|}{\mathsf{polylog}(d)}$.*

Moreover, we prove the following lemma for comparing the updates of different variables.

**Lemma H.3** (TPM of different degrees). *Consider an increasing sequences $x_t \geq 0$ defined by $x_{t+1} = x_t + \eta C_t x_t^q$, for some integer $q > q' \geq 3$ and $q' \leq q - 2$, and $C_t > 0$, and further suppose given $A = O(1)$, there exists $t' \geq 0, x_{t'} \geq A$. Then for every $\delta > 0$ and every $\eta \in (0,1)$:*

$$\sum_{t \geq 0, x_t \leq A} \eta C_t x_t^{q'} \leq (1+\delta)^{q'} \left( O(1) + \eta b A^q \right) \frac{1}{x_0^{q-q'-1}}$$

$$\sum_{t \geq 0, x_t \leq A} \eta C_t x_t^{q'} \geq (1+\delta)^{-q'} \left( \delta(1+\delta)^{-1} \frac{1 - (1+\delta)^{-b(q-q'-1)}}{1 - (1+\delta)^{-(q-q'-1)}} - \eta b A^q \right) \frac{1}{x_0^{q-q'-1}}$$

*where $b = \Theta(\log(A/x_0)/\log(1+\delta))$. When $A = x_0 d^{\Theta(1)}$, $\eta = o(\frac{1}{A^q \delta})$ and $q = O(1)$, then*

$$\sum_{t \geq 0, x_t \leq A} \eta C_t x_t^{q'} = \Theta\left(\frac{1}{x_0^{q-q'-1}}\right)$$

*Proof.* For every $g \in 0, 1, \ldots$, we define $\mathcal{T}_g := \min\{t : x_t \geq (1+\delta)^g x_0\}$. and define $b := \min\{g : (1+\delta)^g \geq A\}$, we can write down the following two inequalities according to the update of $x_t$:

$$\sum_{t \in [\mathcal{T}_g, \mathcal{T}_{g+1}]} \eta C_t [(1+\delta)^g x_0]^q \leq (1+\delta)x_{\mathcal{T}_g} - x_{\mathcal{T}_g} + \eta A^q \leq \delta(1+\delta)^g x_0 + \eta A^q$$

$$\sum_{t \in [\mathcal{T}_g, \mathcal{T}_{g+1}]} \eta C_t [(1+\delta)^{g+1} x_0]^q \geq (1+\delta)x_{\mathcal{T}_g} - x_{\mathcal{T}_g} - \eta A^q \geq \delta(1+\delta)^g x_0 - \eta A^q$$

where $g+1 \leq b$. Dividing both sides by $[(1+\delta)^g x_0]^{q-q'}$ in the first inequality and $[(1+\delta)^{g+1} x_0]^{q-q'}$ in the second, we have

$$\sum_{t \in [\mathcal{T}_g, \mathcal{T}_{g+1}]} \eta C_t [(1+\delta)^g x_0]^{q'} \leq \frac{\delta}{(1+\delta)^{g(q-q'-1)}} \frac{1}{x_0^{q-q'-1}} + \frac{\eta A^q}{x_0^{q-q'-1}}$$

$$\sum_{t \in [\mathcal{T}_g, \mathcal{T}_{g+1}]} \eta C_t [(1+\delta)^{g+1} x_0]^{q'} \geq \frac{\delta(1+\delta)^{-1}}{(1+\delta)^{(g+1)(q-q'-1)}} \frac{1}{x_0^{q-q'-1}} - \frac{\eta A^q}{x_0^{q-q'-1}}$$

Therefore if we sum over $g = 0, \ldots, b$, then

$$\sum_{t \geq 0, x_t \leq A} \eta C_t x_t^{q'} \leq \sum_{t \geq 0, x_t \leq A} \eta C_t [(1+\delta)^{g+1} x_0]^{q'}$$

$$= (1+\delta)^{q'} \sum_{t \geq 0, x_t \leq A} \eta C_t [(1+\delta)^g x_0]^{q'}$$

$$\leq (1+\delta)^{q'} \sum_{0 \leq g \leq b} \left( \frac{\delta}{(1+\delta)^{g(q-q'-1)}} \frac{1}{x_0^{q-q'-1}} + \frac{\eta A^q}{x_0^{q-q'-1}} \right)$$

$$= (1+\delta)^{q'} O\left( \frac{\delta}{(1+\delta)^{q-q'-1} - 1} + \eta b A^q \right) \frac{1}{x_0^{q-q'-1}}$$

$$\leq (1+\delta)^{q'} O\left( \frac{1}{q-q'-1} + \eta b A^q \right) \frac{1}{x_0^{q-q'-1}}$$

For the lower bound, we also have

$$\sum_{t \geq 0, x_t \leq A} \eta C_t x_t^{q'} \geq (1+\delta)^{-q'} \sum_{t \geq 0, x_t \leq A} \eta C_t [(1+\delta)^{g+1} x_0]^{q'}$$

$$\geq (1+\delta)^{-q'} \sum_{0 \leq g \leq b} \left( \frac{\delta(1+\delta)^{-1}}{(1+\delta)^{(g+1)(q-q'-1)}} - \eta A^q \right) \frac{1}{x_0^{q-q'-1}}$$

$$= (1+\delta)^{-q'} \left( \delta(1+\delta)^{-1} \frac{1 - (1+\delta)^{-b(q-q'-1)}}{1 - (1+\delta)^{-(q-q'-1)}} - \eta b A^q \right) \frac{1}{x_0^{q-q'-1}}$$

$$= (1+\delta)^{-q'} \left( \delta(1+\delta)^{-1} \frac{1 - (1+\delta)^{-b(q-q'-1)}}{1 - (1+\delta)^{-(q-q'-1)}} - \eta b A^q \right) \frac{1}{x_0^{q-q'-1}}$$

Inserting $b = \Theta(\log(A/x_0)/\log(1+\delta))$ proves the lower bound. For the last one we can choose $\delta = \frac{1}{\sqrt{\log d}}$ to get:

$$b = \Theta(\mathsf{polylog}(d)), \quad \frac{\delta(1 - (1+\delta)^{-b(q-q'-1)})}{1 - (1+\delta)^{-(q-q'-1)}} = \Omega(1), \quad (1+\delta)^{-q'} = \Omega(1),$$

which proves the claim. $\qquad \square$