# OpenReview forum: "The Mechanism of Prediction Head in Non-contrastive Self-supervised Learning"
_NeurIPS.cc/2022/Conference — NeurIPS 2022 Accept_

### Official Review · Reviewer_X3Gx · 2022-07-04

**Rating:** 7
**Confidence:** 4
**Soundness:** 3 good
**Presentation:** 4 excellent
**Contribution:** 4 excellent

**Summary:**

This paper provides original theorems to explain the convergence of self-supervised learning methods based on positive views only (i.e., BYOL-like). Under reasonable assumptions, the authors demonstrate that the predictor module prevents the appearance of dimensional collapse. Interestingly, the authors could mathematically and empirically highlight different training regimes. They also introduce the concept of weak vs. strong features to intuit the learning dynamics.

DISCLAIMER: I have not checked the Appendix yet, for they are 40 pages long. I *may* have additional comments during the discussion as the Appendix requires quite some extra time to parse. Therefore, this review did not check the mathematical details. Besides, I cannot put high positive/negative scores for intellectual ethic reasons.


**Questions:**

I have many questions listed below:

A) It is a recurrent complaint about the description of BYOL-like methods: BYOL does not optimize a loss nor have a global minimum. It is a dynamic system (or a game) that has equilibria; as mentioned in the original paper, similar to GAN, BYOL has no optimum too (because of the stop gradient). As far as I could tell from the proof, the authors work at the gradient level, i.e., training updates, and not loss minimization, so the proof still holds. I would thus recommend that the authors replace the mention of loss with gradient updates or equilibria (depending on the context); especially l49 and 248.

B) As mentioned by the authors in Section 1.1., this paper does not study the convergence from an eigenvalue perspective. However, the concept of weak/strong features could be related to high/low eigenvalues. Do you have any intuition about it? From an empirical perspective, I would try to plot the eigenvalues (linear layer only), and see whether they may relate to the training phase you mention.

C) I may be missing something... but I do not understand how dotted lines may match the magnitude of non-dotted lines in Figure 2. Indeed, I would expect $E^t - (E^t)^T$ to be around zero (or around 1). I do understand that the off-diagonal is non-symmetric, but it here means that there exist a few leading values where the transpose is null (which sounds weird). Can you explain to me why the difference is so tight?

D)  In corollary 4.2, the authors demonstrate that dimensional collapses are avoided. However, do the authors have a similar results that show that without the predictor, the dimensional collapse will happen with high probability? Here, I am afraid of a potential confirmation bias.

E) In the footnote, 1 page 7, the authors mentioned that the same proof might be feasible with $\eta_E = \eta$. On a personal note, I do not think that it is possible in the BYOL setting. Indeed, BYOL requires the predictor to be close to optimal. Therefore, $\eta_E$ must be smaller than $\eta$, cf Appendix I.1 of the original BYOL paper. Note that this remark may be wrong in the SimSiam setting because of the inner optimization loop. Would it be possible to discuss this point and potentially update the paper to link it to the optimal predictor observation?

F) In the paper, the authors explored different normalizations, showing that a non-trainable BN may be enough to avoid the collapse (even if it only returns mediocre results). It is indeed quite interesting and pushes further the limit of non-negative SSL methods. Differently in Appendix F.6 - Figure 7 of BYOL, the authors also noticed that the predictor works without normalization if and only if it is combined with weight decay. In such a case, BYOL diverged, but it is then stabilized because of the WD. In the absence of WD, the representation just diverged to infinity. More generally, the encoder WD is one of the most crucial parameters in BYOL. From your analysis, can you explain/extend this empirical (and reproducible) phenomenon?

G) It is quite an open remark. One of the strengths of a hypothesis is to be able to be predictive and thus... be contradicted. If you had infinite resources, what would be the experiment you would do (and the observation that you would get) that could refute your paper? Please, note that I am not trying to trap you here but to understand the limit cases better.

Remarks:
  - When mentioning BN, I would recommend mentioning that you are using non-trainable BN, and only the centering/normalizing components
 - The size/depth of the predictor may have a bigger/smaller impact given the task difficulty. Therefore, I am not surprised that you empirically observed little difference between method accuracy in Figure 1. Differently, did you gather some intuition regarding the predictor of inner/outer dimensions during your work? (i.e., some quantities that may depend on $|z|$ or generalization requirement)?
 - While the mathematical notation is very well explained, it may take time to apprehend them for some people. The authors mentioned that they give an illustration in Figure 3 (Appendix). Yet, the figure barely ground the notation, and it is simply the classic non-negative SSL pipeline. I would strongly recommend that the authors create a new large figure to illustrate all the mathematical variables. In other words, the paper impact could be even more significant if they use sketches to explain the results.
 - Please add all your training hyperparameters in the Appendix for your experiments.


Nit:
  - non-contrastive methods may be too much of an approximation for BYOL-like methods. Indeed, pretext task methods would fall under this umbrella. Yet, I do not have a better naming, and BYOL-like is awkward too.
 - Put the OPT definition in the main paper instead of the footnote. I looked for it for a few minutes!
 - sigma is used both for activations and std; please use another notation for activation
 - In equation 2.1, I would add j != i in the sum of the negative

In summary, I am quite positive about the paper. Even though I still have many questions, most of them only deal with opening directions.

**Strengths And Weaknesses:**

This paper is impressive for multiple reasons.
First, the theoretical contribution is partially resolving a recent open problem in self-supervised learning: why non-contrastive methods such as BYOL or SimSiam do not collapse while they have trivial poor minima. Furthermore, the authors even describe learning dynamics, which is quite remarkable and even unique to my knowledge.
Second, the authors did a fantastic job explaining their theoretical results, making it extraordinarily accessible for most self-supervised learning practitioners. The structure with a formal definition, the lemme, and the intuition combine rigor and pedagogy.
Third, the authors complete their theoretical insights with experiments. While less impressive and may still be completed, they perfectly complete the papers.
Overall, this paper and [1] now provide the basics to understand the impressive and surprising results of non-contrastive methods

The main paper's weakness also results from its strength: the level of granularity of the learning dynamics may be too precise and, therefore, more open to debate. For instance, the time complexity may overlap between the lemma (if I am not mistaken, they rely on different parameters). Therefore, the authors may have wanted to make the story a bit too much compelling: while they describe it as an iterative process, the training stage phenomenon may occur jointly.

[1] Tian, Yuandong, Xinlei Chen, and Surya Ganguli. "Understanding self-supervised learning dynamics without contrastive pairs." International Conference on Machine Learning. PMLR, 2021.

---

> ### Author Response · Authors · 2022-08-03
> **Response to reviewer X3Gx**
>
> We are extremely thankful for your appreciation of our work. We will address the concerns below and discuss further possible ways to improve the paper.
>
> We quite agree with the weaknesses pointed out by the reviewer. Our theory is based on a special case data distribution, and are derived with many simplifications of the real scenarios to make the message clean and easily understandable. It is very likely that this theory cannot describe many practical phenomena in real experiments, and that the clear phase transitions is actually mixed together in real data experiments. We actually hope practitioners can take away the intuitive message derived from our statements rather than the technical description of the training phases, which are intended as an introduction to the rigorous derivation of our statements.
>
> ### Answers to the questions
> - A) To us the mentioning of loss minimization is mainly for people working on the theory side to understand our theoretical challenges, and more importantly, how our challenges differ from those encountered in the traditional learning theory papers (and variants of those frameworks). Without mentioning the loss minimization aspects and providing the loss convergence result for our theory, it might be hard for people working in these area to appreciate and understand our challenges and results. It would also be good to mention the equilibrium and gradient trajectory though.
> - B) Visualizing the eigenvalues can be a possible way of looking for the phenomenon our theory and what we saw in synthetic data experiments. However, due to the non-linearity involved in the deep encoder network and the projection head, it might be harder to see an eigenvalue trajectory similar to our theoretical predictions.
> - C) Thank you so much for pointing out the closeness between the trajectories of $E$ and $E - E^{\top}$. We previously thought it to be a coincidence, but now we might have a new understanding that is also predicted in our theory. Indeed, suppose the off-diag prediction head really serves to substitute the strong feature of slower neurons, then for each pair of neurons, **the substitution can only happen uni-directionally**: meaning that for each pair $E_{i,j}$ and $E_{j,i}$, only one of them can be large, or otherwise there may be other factors involved that cannot be explained (like in the STL-10 experiments).
> - D) Yes, and the result is shown in Corollary 4.4.
> - E) It is worth noting the observation in the BYOL paper. However, there might be issues connecting those observations. As SimSiam is actually using the same setup in the optimal predictor section but with $\lambda$ set to $1$, it still has comparable performance to BYOL, which is much better than in the optimal predictor section. It is a bit strange to see such a discrepancy between experiments. I personally believe that the experiments in the BYOL paper might have some implementation details that affects their results.
> - F) It is possible that using WD without output-normalization can make BYOL work, as weight regularizers are often close to weight-normalization, which is also similar to the neuron-wise regularization. It is of interest to explore the difference between different normalization scheme and the differences in their mechanisms.
> - G) A possible experiment is to train with transformer-based model on larger datasets (e.g. ImageNet 22k). Currently DINO is the standard choice for non-contrastive method with transformers, we can see if a transformer-based model can also benefit from using a prediction head structure.
>
> We also thank the reviewer for the valuable remarks and comments, many of them will be incorporated in the next updated version of our paper.

---

> > ### Comment · Reviewer_X3Gx · 2022-08-08
> > **Reviewer X3Gx**
> >
> > A) This is my point, there is no loss, and the actual theory should think about BYOL-like approach, not as a loss minimization problem, but as a dynamic problem. Yet, people keep wrongly refering it as a loss. It is simalar to think a multi-agant problem as a RL problem. Following this analogy, there is no optimal policy, but equilibria between different policies. Here, there is no minimal loss, but a stable equilibria. Therefore, if "To us the mentioning of loss minimization is mainly for people working on the theory side to understand our theoretical challenge", they would start with the wrong a priori, and would miss the point.
> >
> > B) It is possible if you use you a linear layer, and compute the SVD on small dataset (or ImageNet Subet). Cf Figure 2 from DirectPred
> >
> > C) I would then strongly recommend to use the extra page to comment on this point
> >
> > E) I would encourage the authors to keep thinking about the link with the optimal predictor. I implemented BYOL three to four tiems already, and this observation is definitly not some implementation detail :). The SimSiam inner loop is simply a way to keep the predictor to optimality without using an ema, (The ema network makes the target distribution moves slower, allowing the predictor to remain close to optimal). In any case, this optimal predictor is the key for stability. If you manage to somehow incorparate the predictor error, this would be another core contribution.
> >
> > F) Interesting point. Thx!
> >
> > My apologize as I could not dig into the long mathematical details further. Therefore, I would stick to Accept for ethical reasons.

---

### Official Review · Reviewer_hrov · 2022-07-10

**Rating:** 4
**Confidence:** 2
**Soundness:** 2 fair
**Presentation:** 2 fair
**Contribution:** 3 good

**Summary:**

This paper aims to theoretically analyze why non-contrastive self-supervised learning methods (e.g., BYOL and SimSiam) can avoid trivial solutions even without negative pairs. To this end, the authors assume a simple setting: (i) data contains only two feature vectors, (ii) the backbone encoder architecture is a single-layer CNN with a non-linear activation, and (iii) the prediction head architecture is a linear layer. Under these assumptions, this paper proves that the trainable prediction head can learn CNN to avoid trivial solutions. Furthermore, this paper provides experimental results that support the theoretical results under more standard (i.e., real-world) settings.


**Questions:**

**Q1.** Why $|E_{2,1}|$ is increasing in Phase II?

At the ends of Phase I and II (Lemma C.4 and D.8, resp.), $B=(B_{ij})=(w_i^\top v_j)$ does not change with $B_{1,1}=\Theta(1)$ and $B_{ij}=\Theta(1/\sqrt{d})$ for $(i,j)\neq(1,1)$, but $E_{2,1}$ increases from $\tilde{O}(\eta_E / \eta d)$ to $\Theta(\sqrt{\eta_E / \eta})$. It means that $F_2$ chagnes but $G_2$ does not in Phase II. This does not make sense because our objective (3.3) minimizes the distance between $F$ and $G$.

Also, the intuition of the substitution effect (L276-L279) is also hard to understand. Why does borrowing the features in $f_1$ minimize the distance between $F_2$ and $G_2$?

---

**Q2.** After Phase II, what is $w_2$?

This question is closely related to Q1. If Lemma 5.2 holds, then $F_2\approx E_{2,1} f_1$. Due to the objective (3.3), $G_2=f_2 \approx F_2$ anytime. It means that $w_2$ should be aligned with $w_1$ which is also aligned with $v_1$ by Lemma 5.1. Namely, $w_2\approx v_1$ at the end of Phase II. Is it right? If so, why does $w_2$ change from $v_1$ to $v_2$ in Phase III? I think $w_2$ will remain as $v_1$ after Phase II.

---

**Q3.** Why use Batch Normalization (BN) instead of l2 normalization?

BN is an unnecessary component in non-contrastive SSL methods, as shown in [68]. Also, BYOL and SimSiam apply l2 normalization without BN at their output layers. However, $\tilde{F}$ and $\tilde{G}$ defined in L198-200 use BN, and the objective (3.3) does not use l2 normalization. This gap may undermine this paper's motivation.


**Limitations:**

This paper does not address the limitations and potential negative societal impact.

**Limitations.** This paper uses strong assumptions on data distribution (i.e., only two feature vectors) and network architectures (i.e., 1-layer CNNs). Also, this paper does not focus on l2 normalization, but it is widely utilized in various SSL methods.

**Negative societal impact.** This paper is a theory paper. I think there is no negative impact.


**Strengths And Weaknesses:**

I think the main **strength** of this paper is the theoretical analysis of non-contrastive SSL methods. I agree that why the methods are working is less studied, so such a theoretical study is important in the field of SSL. Furthermore, the empirical observations with linear prediction heads support the validity of the theoretical assumptions well.

My main concern with this paper is the presentation quality. In general, it is hard to understand statements and their intuitions. First of all, Definition 3.1 is hard to understand:
- There is no Figure 3 about Definition 3.1 (e.g., see L173 and L213).
- Confused notations: $\mathcal{S}(X)$ and $S(X)$ (e.g., see L170, L172, L180).
- Does all $X$ in $\mathcal{D}$ share the same feature vectors $\{v_\ell\}_{\ell\in[2]}$? If so, Definition 3.1 seems to assume that the data distribution $\mathcal{D}$ has only two data points (with some noises). Is this reasonable?
- I think the constraint in L171, i.e., "all feature patches have the same direction of $v_\ell$", is also too strong.
- In L171, it would be better to use $z_p(X)=z_{p'}(X)\in\{\pm\alpha_\ell\}$ rather than $\{0,\pm\alpha_\ell\}$ because for any $p\in\mathcal{S}$, $z_p\neq 0$ by definition.
- You should describe what is $\xi_p$ first. Until p5, it is hard to know whether $\xi_p$ is a random variable, a constant vector, or something else.

And, what are the intuitions of theorems in Sections 4 and 5? I can understand what the theorems state, but it is hard to know how they can be derived. For example, how does borrowing the features in $f_1$ by increasing $E_{2,1}$ minimize the distance between $F_2$ and $G_2$? I feel that the authors should allow more space for explanations about the substitution and acceleration effects since they are key components of this paper. I also feel it would be better to provide empirical results based on the theoretical setup described in Section 3 to understand the changes in Phase I ~ IV.

To sum up, I think this paper's contributions are meaningful, but the presentation is unsatisfactory, and some results are still hard to understand. Hence, I vote for borderline rejection now, but I'm willing to increase my score after a strong rebuttal.

---

> ### Author Response · Authors · 2022-08-03
> **Response to reviewer hrov**
>
> We are extremely grateful for the acknowledgement of our contributions toward the subject. We admit that our exposition needs to be improved, and we shall correct the problematic parts in the main paper (especially on the definition of data distribution) in the next version. Below we mainly address your concerns on the correctness of the theoretical claims.
>
> ### On the Questions
> It is a little bit involved to answer the technical questions, I hope the following explanation is both simple enough to understand, and also informative enough to grasp the technical ingredient.
>
> We observe that the confusion mostly comes from the conflicts between the minimization of objective and the growth of $E_{2,1}$ and the misalignment of features in $F_2, G_2$. So we shall first clarify these issues here.
>
> **On the minimization of objective:** Recall that our objective is defined as
> $$
> L(F) = \mathbb{E}_{X^{(1)}, X^{(2)} } [ || \widetilde{F}(X^{(1)}) -  \widetilde{G}(X^{(2)}) ||^2 ]  =  2 - 2 \mathbb{E}[ \langle \widetilde{F}(X^{(1)}), \widetilde{G}(X^{(2)}) \rangle ]
> $$
> It can be seen that this objective aligns $\widetilde{F}(X^{(1)})  := \mathsf{BN}(F(X^{(1)})) $ and $\widetilde{G}(X^{(2)}) = \mathsf{BN}(G(X^{(2)}))$, which are normalized version of $F$ and $G$ (batch-wise centered and normalized).
> There are multiple implications of this form:
> 1. As long as $F_i$ and $G_i$ are **aligned in direction over the population**, the objective can be minimized. So it is not needed to have $F-G \approx 0$ to minimize the objective. For example, $F_i \propto G_i$ (and have the same sign) can be an optimal solution as well, and thus if $E_{i,3-i}f_{3-i}$ is aligned with the direction of $G_i$, it can help reduce the objective.
> 2. Since $X^{(1)}, X^{(2)}$ is a augmented pair that contains non-overlapping noise patches, and also since output-normalization is used, the alignment problem is not just about the aligning the features in $F$ and $G$, but also about reducing noise proportion in $F$ and $G$, because it helps reduce the normalizing constants. This also holds true in the case of using $\ell_2$ norm as output normalization.
> 3. Again because of our data augmentation to get the positive pair, the standard Gaussian initialization of weights $ w_{i}^{(0)} \sim \mathcal{N}(0,I_d/d) $ makes the network focus much more on the noise patches at init. The objective is minimized only when at least one of the features is learned and the noise are all removed.
>
> *Remark:* Our experiments in Figure 1 shows that *loss may rise during the intermediate stage*, so objective minimization could be misleading sometimes for understanding the dynamics.
>
> Based on the above, we can answer the questions:
>
> **Answer to the first question:**
> To answer why $E_{2,1}$ can grow while $G_2$ remains largely the same, first notice that the neurons at (or close to) initialization focus more on the noise than the features, which makes learning any feature (in $F$ or in $G$) beneficial to minimizing the objective.  In this case, if $ F_2 $ can obtain a high signal-to-noise ratio feature using $E_{2,1} f_1$ (where $E_{2,1}$ controls the feature sign to be the same with $G_2$'s), then it can at least help align $\widetilde{F}_2$ and $\widetilde{G}_2$ over data points that contains feature $v_1$, thereby helps reducing the objective.
>
> **Answer to the second question:**
> Indeed $F_2 \approx E_{2,1}f_1$ at the end of phase II is correct, but notice that at this moment $G_2 = f_2$ contains a lot of noise here, so the objective are not sufficiently minimized. Any improvement of signal-to-noise ratio of $F_2\times G_2$ (even only on one side) can help reduce the objective. The technical ingredient behind phase II is the update speed of $E_{2,1}$ is faster than the feature learning of $G_2$, which is due to the high signal-to-noise ratio of $f_1$ that boosts the gradient of $E_{2,1}$ after phase I.
>
> **Answer to the third question:**
> 1. Using BN help us separate the cases of **dimensional collapse** and successful learning. As we emphasize on **feature diversity**, we prove that BN alone cannot diversify the features (actually it is simple to obtain a complete collapse result for using $\ell_2$-norm without prediction head), but training with prediction head can.
> 2. The mechanism behind $\ell_2$-norm is even simpler--as $\ell_2$ norm is a more intuitive way to balance gradients according to the feature magnitude: larger and already learned feature/data has smaller gradient, and smaller and yet to learn feature/data has larger gradient. However it's technically much harder to use $\ell_2$ norm, because using $\ell_2$ norm requires working at the sample level (not expectation level as we do), which needs much more complicated anti-concentration arguments, and does not provide additional insights.
>
> Lastly on the experiment side, a synthetic experiment supporting our theory will be added in the next version.

---

> > ### Comment · Reviewer_hrov · 2022-08-08
> > **Thank you for the response.**
> >
> > Thank you very much for your time and effort in writing this response.
> >
> > **Additional response & Q3.** I got your points. I agree that the observations of this paper are meaningful in the field of non-contrastive self-supervised learning, although the assumptions seem to be somewhat artificial.
> >
> > **Q1-Q2.** Now I understand what happens in Phase II intuitively. Thank you for your explanation.
> >
> > However, I am still concerned about the presentation quality, and my concerns are not resolved in the revision yet. So I would like to keep my initial rating due to this reason. Other reviewers seem to think differently on this point, so I'll discuss this in the reviewer discussion period.

---

> ### Author Response · Authors · 2022-08-04
> **Additional response**
>
> **Question:**
> Does all $X$ in $\mathcal{D}$ in share the same feature vectors? If so, Definition 3.1 seems to assume that the data distribution has only two data points (with some noises). Is this reasonable?
>
> **Our explanation:**
> Yes, all $X$ in $\mathcal{D}$ in share the same feature vectors. However, we are only having two neurons in the learner to learn the two features, so the capacity is almost exactly enough. Moreover, it is not inaccurate to view our data distribution as having only two data points with noises (I am assuming the reviewer think of this as a two-mixture of Gaussian). The features in the data shows up in the different places/patches across different data points, which also contributes to the intra-cluster variance, and thus the data points (or cluster centers) scale at least linearly to the number of patches.
>
> Beyond technical sophistication, what is more important is that we want to capture the spirit of our observation in the simplest way to facilitate understanding. it is unclear whether more complicated data setting can provide any additional insight into the problem. Actually even in this simple setup, due to the multiple phases in the training process, our proof still took more than 50 pages to finish, thus we believe it is better to convey our message under the simplest setting as possible.

---

### Official Review · Reviewer_a9Hp · 2022-07-11

**Rating:** 6
**Confidence:** 2
**Soundness:** 3 good
**Presentation:** 3 good
**Contribution:** 3 good

**Summary:**

The authors explore why BYOL-style self-supervised learners do not collapse to trivial solutions from a theoretical perspective, focusing on the prediction head component of the architecture. They use a simplified setting with a strong feature and a weak feature that compete. They define various phases of training and describe what their theoretical results suggest should happen in those phases. Some empirical results are presented to support this.

**Questions:**

What limitations does using the strong/weak feature setup introduce?
Could you elaborate more on how batch normalization interacts with the stop-gradient to contribute to the acceleration effect?
As someone outside of deep learning theory, I would like to know more about how the assumptions posed here differ from those in other works, as well as some justification of these difference.

**Limitations:**

See questions. I am mostly interested in the limitations posed by the assumptions utilized in this work.

**Strengths And Weaknesses:**

Strengths:
- The authors consider an important problem in a way that is simplified but feels justified.
- Simple steps are provided to qualitatively understand what the theory reveals about the training process.
- The previously-introduced dimensional collapse problem and reasons behind it are made conceptually clear in this work.
- I like the organization of the paper and the intuition provided at different steps.

Weaknesses:
- I found the acceleration effect section relatively harder to follow.
- The writing could use some revision. A review with an eye for grammar in particular would be helpful.
- The strong/weak feature assumption could be better motivated.

---

> ### Author Response · Authors · 2022-08-04
> **Response to reviewer a9Hp**
>
> We thank the reviewer for the valuable comments. We admit that the strong/weak feature setting could be better motivated. Below we shall address your questions.
>
> **On the strong/weak features:**
> The strong/weak feature setup is motivated by the dimensional collapse phenomenon. The dimensional collapse phenomenon essentially means that, even with BN to ensure non-zero variance of the representation, the neural network tends to learn 1-dimensional representation without the prediction head. In light of this, it is very natural to think that the 1-dimensional representation only captures one feature that has the largest variance (and thus having the largest signal strength and are easier to be picked up), which leads to the strong feature in our setting. And then the weak feature is just a natural construction to explain why the neural net could possibly fail to learn the more diverse feature.
>
> The limitations of the strong/weak feature setup is mostly its simplification of the reality. In real cases such as image pretraining, there are multiple possible causes for the features to be "strong" or "weak". For example, maybe certain features are easier to be learned by vision transformers than by convolutional neural networks, or maybe certain features are more easily picked up by Adam algorithm. Moreover, real image features may be hierarchical and much more complicated, which is also hard to be incorporated into our setting unless more advanced theory tools are developed.
>
> **On the output-normalization and stop-grad:**
> Without stop-gradient, the algorithm align $F_i = f_i + E_{i,3-i}f_{3-i} $ and $G_i = f_i$ together. But with stop-grad on $G_i$, when you compute the gradient of $f_i$, you only align $f_i$ with $f_i$ together (ignoring smaller terms). This remove the influence from the $E_{i,3-i}f_{3-i}$ to the learning of $f_i$.
>
> The BatchNorm, or what we view as simply an output-normlaization, serves to balance the learning of different features at later stages. It cannot work alone without the help of prediction head and the stop-grad, but since its gradient calculation is a bit involved, we suggest one only take away the message that it can help enlarge the gradient of weaker features, and suppress the gradient of stronger features (when the substitution effect happened).
>
> **On the difference of assumptions:**
> Our setup is mostly unique, as there is few work in this area. But we highlight that the most important difference between our work with similar works is that we **analyze the SGD training process of prediction head, rather than manually setting it to a special matrix**. This allows us to obtain a complete training dynamic analysis that conforms with our empirical observations.

---

### Official Review · Reviewer_6azf · 2022-07-14

**Rating:** 6
**Confidence:** 4
**Soundness:** 3 good
**Presentation:** 3 good
**Contribution:** 4 excellent

**Summary:**

This paper studies how the prediction head in non-contrastive learning avoids trivial collapsed solutions (full collapse or dimension collapse). The empirical observations made in the paper are: (1) initializing the prediction head to identity matrix and training only the off-diagonal entries can lead to competitive performance, (2) the off-diagonal entries display a rise and fall behavior. The paper then provides theory for a simple setting to explain this phenomenon. It analyzes the dynamics of a two-layer non-linear convolutional style  neural network for a setting where data contains a strong (higher weight) and a weak feature (lower weight). Here the following can be shown: (1) without a prediction head the network will demonstrate dimensional collapse, even with batch norm (BN), (2) with a trainable prediction head + BN.+ stop-gradient can avoid such a collapse. The analysis identifies two effects that help with this, (1) substitution effect where once the stronger feature is learned in a neuron, the prediction head helps decrease the learning speed of the strong feature in other neurons and (2) acceleration effect where the strong features, via the prediction head, can further accelerate the learning of weaker features in the substituted neurons. These two steps also benefit from BN and stop-gradient.

**Questions:**

- Is there a way to check for substitution + acceleration in real networks? For instance, noticing that some features are trained faster and others.
- What is the distribution of $z_p$ in Definition 3.1?
- L173: Figure 3 is missing or wrongly referenced? Figure 3 in the appendix is just a simple framework.
- L205: [84] is cited twice
- OPT is undefined
- L241: will creates -> will create
- L51: “no well-established statistical framework” [50] connects non-contrastive methods like SimSiam with a identity covariance constraint to non-linear CCA. Although not the case where there are collapsed solutions

**Limitations:**

Some limitations discussed, The contributions being theoretical, there is no immediate negative societal impact.

**Strengths And Weaknesses:**

**Strengths**
- Studies a relevant problem of understanding why non-contrastive methods do not collapse and makes concrete progress towards it. To my knowledge this is the first paper that studies this for a non-linear neural network. I believe this paper would help subsequent work on non-contrastive methods.
- The use of linear predictor + identity init + trainable off-diagonal entries is new and interesting. The empirical findings about rising and falling off-diagonals and theoretical results capturing this behavior seem non-trivial.
- The paper is well written and mostly easy to follow

**Weaknesses**

The main “weakness” in my opinion is lack of discussions about certain aspects:

- Role of non-linearity: Analyzing non-linear models is key to opening up the deep learning black-box. That said, the paper currently lacks a discussion about whether and why non-linearity is crucial for the example being studies. In particular, what would happen with a linear model in this case? Will there be dimension collapse? Will substitution and acceleration kind of phenomena not happen? Including a discussion about this could be insightful either way.
- Proof sketch: The discussion about the various phases in Section 5 is much appreciated, as it also helps understand the role of various components like trainable predictor, BN and stop-gradient. However I felt that the paper could benefit from from some more details in the proof sketch, especially to describe the substitution and acceleration effects. In particular, it might help to spell out (in terms of gradient computations say) what happens in phase III when there is no normalization and/or stop-grad.

Given these (relatively minor) concerns I am currently assigning a score of weak accept. I was unable to read the proofs in the appendix and so the above discussion will also help with increasing confidence in the results. I would be happy to raise the score if the above points are addressed in the author response, because I believe that the overall contribution of the paper is positive.

---

> ### Author Response · Authors · 2022-08-02
> **Response to reviewer 6azf**
>
> We thank the reviewer for the acknowledgement of our conceptual and technical contributions to understand the subject. We shall address your concerns of the weaknesses and your questions below, in the hope of clarifying the unclear parts not sufficiently addressed in the paper.
>
> ### About the weaknesses
>
> **Regarding the role of non-linearity:** The use of non-linear encoder network here is based on multiple reasons, as we list here:
> 1. Using linear network as the encoder creates plenty of invertible transformations between different solutions of encoder network, which mixed the update of encoder weights and prediction head matrix together, and complicates their interactions. It would be unconvincing to speak about the role of prediction head in this case.
> 2. Under linear network, current technique can barely describe the difference of learning speed of different neurons (not different features), thus can hardly be used to disentangle the different phases in the optimization. It is also very likely that similar results can hardly be obtained under linear network setting. As previous works [1] which relies on the linearity of network can only analyze the case of freezing the prediction head.
>
> **Regarding proof sketch:** Thank you so much for the appreciation of our explanation about the training phases. Due to the page limit of the conference and the extremely long forms of the gradient of each trainable parameter (which can be seen in the Appendix section B), we are unfortunately unable to incorporate a more detailed explanation of the proof in the main paper. However, your suggestion is still very much appreciated. It is likely that we will add a section in the appendix explaining the role of each components in the gradient at different phases, which can hopefully benefit readers who are, like us, amazed by the interesting mechanism behind this method.
>
> **As for the mechanism of stop-grad and output normalization**, it absolutely deserve more (and clearer) explanation as you suggest. But the more intuitive version wrote in section 5 is actually what we had in mind when deriving the result. In a more intuitive way, the stop-grad mainly serves to isolate the learning the strong feature in neuron 1 with learning the weak feature in neuron 2. And without the output normalization, there will not be a re-balance of learning speed of different features and lead to collapse, as in this case the stronger feature will always have a larger gradient. The proof is implicitly incorporated in our induction arguments of phase 3.
>
> ### About the questions
> 1. The clear separation of phases of our theory is actually observed in our synthetic experiment (which uses similar data distributions as our definition), and will be added to the appendix in the next version. However, we do not know how to observe that over real data such as image data.
> 2. Thank you for pointing out! Figure 3 should be a illustrative picture about the data distribution, we will fix this issue in the next version.
> 3. Thank you for pointing out the citation issue.
> 4. Thanks for pointing out! OPT is the global optimal (minimal) objective value, and we will put the definition to the main paper instead of in the footnote in the next version.
> 5. Thank you for pointing out the typo issue.
> 6. Thank you for mentioning the Jason's work. We have the same opinion about their work on this subject, but since we focus on the case where collapsed solutions exist (as we wrote in L51, before "no well-established statistical framework"), we do not think their work is relevant enough in this case.
>
>
> [1] Wang et al. Towards Demystifying Representation Learning with Non-contrastive Self-supervision

---

> > ### Comment · Reviewer_6azf · 2022-08-09
> > **Thank you**
> >
> > I thank the authors for their responses about the role of non-linearity and intuitions for the different stages. It would be interesting to check whether a linear network succeeds or fails under the setting being analyzed, since the author response suggests that the outcome is unclear (with failure being more likely).
> >
> > I am likely to increase my score to "Accept" after taking another look at the paper. Just to confirm, was there an updated "Figure 3" included in the revision? I could not find it in the main paper of the supplementary material.

---

### Meta-Review · Area_Chair_g7UA · 2022-08-23

**Recommendation:** Accept
**Confidence:** Certain

**Metareview:**

This paper proposed a theoretical explanation to the open question of why BYOL-style (non-contrastive) self-supervised learning does not collapse to trivial solutions. This paper could be an important contribution to better understanding this widely observed but poorly understood phenomenon. Although there are still some issues raised by the reviewers that are not addressed during the rebuttal, it is generally agreed that the contribution of this paper still overweigh the issues. Therefore, I decide to accept this paper, but urge the authors to fully address the reviewers' comments in the final revision (e.g. missing Fig.3, clarification of Def. 3.1, etc.).

**Award:**

No

---

### Decision · Program_Chairs · 2022-09-14

Accept